# Navigating Neural Space: Revisiting Concept Activation Vectors to Overcome Directional Divergence

Frederik Pahde[1]     Maximilian Dreyer[1]     Moritz Weckbecker[1]     Leander Weber[1]
Christopher J. Anders[2]     Thomas Wiegand[1,2,3]     Wojciech Samek[1,2,3,†]
Sebastian Lapuschkin[1,†]

[1]Department of Artificial Intelligence, Fraunhofer Heinrich Hertz Institute
[2]Department of Electrical Engineering and Computer Science, Technische Universität Berlin
[3]BIFOLD – Berlin Institute for the Foundations of Learning and Data
[†]corresponding authors:
{wojciech.samek,sebastian.lapuschkin}@hhi.fraunhofer.de

## Abstract

With a growing interest in understanding neural network prediction strategies, Concept Activation Vectors (CAVs) have emerged as a popular tool for modeling human-understandable concepts in the latent space. Commonly, CAVs are computed by leveraging linear classifiers optimizing the *separability* of latent representations of samples with and without a given concept. However, in this paper we show that such a separability-oriented computation leads to solutions, which may diverge from the actual goal of precisely modeling the concept direction. This discrepancy can be attributed to the significant influence of distractor directions, *i.e.*, signals unrelated to the concept, which are picked up by filters (*i.e.*, weights) of linear models to optimize class-separability. To address this, we introduce *pattern-based CAVs*, solely focussing on concept signals, thereby providing more accurate concept directions. We evaluate various CAV methods in terms of their alignment with the true concept direction and their impact on CAV applications, including concept sensitivity testing and model correction for shortcut behavior caused by data artifacts. We demonstrate the benefits of pattern-based CAVs using the Pediatric Bone Age, ISIC2019, and FunnyBirds datasets with VGG, ResNet, ReXNet, EfficientNet, and Vision Transformer as model architectures.[1].

## 1 Introduction

In recent years, eXplainable Artificial Intelligence (XAI) has gained increased interest, as Deep Neural Networks (DNNs) are ubiquitous in high-stake decision processes, such as medicine (Brinker et al., 2019), finance (Rouf et al., 2021), and criminal justice (Završnik, 2021; Travaini et al., 2022), with black-box predictions being unacceptable. Whereas local explainability methods compute the relevance of input features for individual predictions, global XAI approaches aim at identifying global prediction strategies employed by the model, often to be represented as human-understandable concepts. Backed by recent research, suggesting that DNNs encode concepts as superpositions in latent space (Alain & Bengio, 2017; Elhage et al., 2022; Nanda et al., 2023; Wang et al., 2023), Concept Activation Vectors (CAVs), originally introduced for concept sensitivity testing (Kim et al., 2018), model concepts in DNNs by finding directions pointing from samples without the concept to samples with the concept. Commonly, the direction is estimated by taking the weight vector of a linear classifier (*e.g.*, a linear Support Vector Machine (SVM)), representing the normal to the decision hyperplane separating the two sample sets. However, while linear classifiers optimize the *separability* of two classes, they might fail at precisely identifying the *signal* direction encoding the

---

[1]Code is available at https://github.com/frederikpahde/pattern-cav

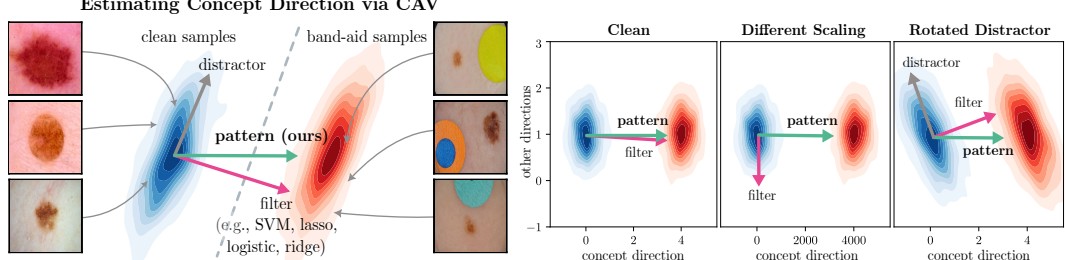

Figure 1: CAVs obtained from filters, *i.e.* weight vectors from linear classifiers, are optimized for class separability, but fail at precisely estimating concept signal directions. *Left:* Different CAV computation strategies are employed to estimate the "band-aid" concept, a confounding artifact in the ISIC2019 dataset. *Right:* Weaknesses of filter-based CAVs are apparent for simple transformations in a 2D toy experiment, where we scale concept features (x-axis) differently than other (*e.g.*, distracting) features or rotate distracting directions. Only pattern-based CAVs precisely estimate the concept signal direction, while filter-based CAVs diverge to optimize class separability. Animated visualizations for these and additional 2D experiments can be found here: https://github.com/frederikpahde/pattern-cav/tree/main/animations.

concept. This can be attributed to the significant influence of distractor (*i.e.*, non-signal) directions contained in the data, which are picked up by filters (*i.e.*, weights) of linear models to optimize class-separability (Haufe et al., 2014). This decomposition of filters into signal and distractor patterns has also been addressed in the context of local explainability methods (Kindermans et al., 2018). We follow their approach and introduce *pattern-based CAVs* for global explainability, disregarding distractors and thereby precisely estimating the concept signal direction (see Fig. 1).

Despite directional divergence from the true concept signal, CAVs have been employed for a plethora of tasks in recent years, such as concept sensitivity testing (Kim et al., 2018), model correction for shortcut removal (Anders et al., 2022; Dreyer et al., 2024), knowledge discovery by investigation of internal model states (McGrath et al., 2022), and training of post-hoc concept bottleneck models (Yuksekgonul et al., 2023). Many of these applications can be improved by more precise concept directions, as provided by pattern-CAVs, instead of optimized class-separability, as provided by filters. To demonstrate the superiority of pattern-CAVs, we run controlled and non-controlled experiments using the Pediatric Bone Age, ISIC2019, and FunnyBirds datasets with VGG, ResNet, ReXNet, EfficientNet, and Vision Transformer architectures. Our main contributions include the following:

1. We introduce pattern-CAVs, more precisely estimating the concept *signal* direction and being less influenced by distractors.
2. We measure the alignment of CAVs with the true concept direction in controlled settings, confirming that pattern-CAVs align with the true concept direction, while the widely used filter-CAVs diverge.
3. We measure the impact of directional shifts in popular CAV applications, including Testing with CAV (TCAV) and model correction with Class Artifact Compensation (ClArC) in controlled and real-world experiments, demonstrating benefits of pattern-CAV in both cases.

## 2 RELATED WORK

A variety of approaches has emerged to identify human-understandable concepts in DNNs. Some works consider single neurons as concepts (Olah et al., 2017; Achtibat et al., 2023), while others focus on identifying interesting subspaces (Vielhaben et al., 2023) or linear directions (Nanda et al., 2023). We follow the latter approach and encode concepts as linear combinations of neurons, also known as superposition (Elhage et al., 2022). These directions can be identified through unsupervised activation matrix factorization (Fel et al., 2023) or by the supervised training of CAVs, *i.e.*, vectors pointing from samples without to samples with the concept. In the absence of concept labels, automated concept discovery approaches can further streamline this process (Ghorbani et al., 2019; Zhang et al., 2021). Various methods leverage CAVs as latent concept representation. For instance, TCAV measures a

model's sensitivity towards specific concepts. ClArC aims to unlearn model shortcuts, *i.e.*, prediction strategies based on unintended correlations between target labels and data artifacts, represented by CAVs. Post-hoc concept bottleneck models project latent representations into a space spanned by CAVs to obtain an interpretable latent representation. Beyond these applications, CAVs have been employed to understand the strategies learned by AlphaZero in playing chess (McGrath et al., 2022) and to identify meaningful directions for manipulation (*e.g. no-smile → smile*) in diffusion autoencoders (Preechakul et al., 2022). Related works aim to enhance CAV robustness by alleviating the linear separability assumption (Chen et al., 2020; Pfau et al., 2021), for example by representing concepts as regions (Crabbé & van der Schaar, 2022). In contrast, our approach adheres to the linear separability assumption but improves the precision of the modeled direction.

## 3 ESTIMATING SIGNAL OF CONCEPT DIRECTION

We view a DNN as a function $f : \mathcal{X} \to \mathcal{Y}$, mapping input samples $\mathbf{x} \in \mathcal{X}$ to target labels $y \in \mathcal{Y}$. Without loss of generality, we assume that at any layer $l$ with $m$ neurons, $f$ can be split into a feature extractor $\mathbf{a} : \mathcal{X} \to \mathbb{R}^m$, computing latent activations at layer $l$, and a model head $\tilde{f} : \mathbb{R}^m \to \mathcal{Y}$, mapping latent activations to target labels. We further assume binary concept labels $t \in \{+1, -1\}$. CAVs are intended to point from latent activations of samples without concept $\mathcal{A}^- = \{\mathbf{a}(\mathbf{x}_i) \in \mathbb{R}^m \mid t_i = -1\}$ to activations of samples with concept $\mathcal{A}^+ = \{\mathbf{a}(\mathbf{x}_i) \in \mathbb{R}^m \mid t_i = +1\}$. The optimal choice of layer $l$ depends on the type of concept, as simple concepts (*e.g.*, color and edges) are learned on earlier layers, while more abstract concepts (*e.g.*, band-aid) are learned closer to the model output (Olah et al., 2017; Radford et al., 2017; Bau et al., 2020).

### 3.1 FILTER-BASED CAV COMPUTATION

Traditionally, a CAV $\mathbf{h}$ is identified as the weight vector $\mathbf{w} \in \mathbb{R}^m$ from a linear classifier, describing a hyperplane separating latent activations of samples with the concept $\mathcal{A}^+$ from activations of samples without the concept $\mathcal{A}^-$. Commonly (Kim et al., 2018; Yuksekgonul et al., 2023), linear SVMs are used, minimizing the hinge loss with L2 regularization (Cortes & Vapnik, 1995). Other options include Lasso (Tibshirani, 1996), Logistic, or Ridge (Hoerl & Kennard, 1970) regression.

Concretely, the classification task is usually described as a linear regression problem. With concept labels $t$ as dependent variable and latent activations $\mathbf{a}(\mathbf{x}) \in \mathbb{R}^m$ as regressors, we assume a linear model $f_{\text{linear}}(\mathbf{x}) = \mathbf{a}(\mathbf{x})^\top \mathbf{h} + b$ with weight vector (or *filter*) $\mathbf{h} \in \mathbb{R}^m$ and bias $b \in \mathbb{R}$. Using ridge regression as an example, the optimization task to find a filter-CAV $\mathbf{h}^{\text{filt}}$ is then given by

$$\mathbf{h}^{\text{filt}} : \min_{\mathbf{h}, b} \|\mathbf{t} - A\mathbf{h} - \mathbf{1}b\|_2 + \lambda\|\mathbf{h}\|_2, \tag{1}$$

where $A \in \mathbb{R}^{n \times m}$ is summarizing latent activations for all $n$ samples in $\mathcal{X}$ in matrix form, $\mathbf{t} \in \mathbb{R}^n$ is the vector with concept label $t_i$ as its $i^{\text{th}}$ element and $\mathbf{1} \in \mathbb{R}^n$ is a vector of 1s. The optimization objectives differ by the type of linear model (see Appendix B.1).

However, research from the neuroimaging realm suggests that filters from linear classifiers not only model the signal separating the two classes but also capture a distractor component (Haufe et al., 2014). This component can arise from noise, but also from unrelated features in the data, which are not directly related to the signal. In the context of CAVs, *any* information unrelated to the concept is considered a distractor. The filters are optimized to weigh all features to achieve optimal separability w.r.t. $t$. However, this optimization does not disentangle concept signals from distractor signals. As a result, distractor pattern present in the training data influence the direction of filter-CAVs.

### 3.2 PATTERN-BASED CAV

We introduce a pattern-based CAV, which is based on the assumption that we can model latent activations given the concept label $t$ via the linear function $\widetilde{f}_{\text{linear}}(t) = t\mathbf{h} + \mathbf{b}$ for a vector $\mathbf{h} \in \mathbb{R}^m$ and a bias vector $\mathbf{b} \in \mathbb{R}^m$. The difference in activations with and without the concept, $\mathbf{h}$, can be obtained by optimizing the following objective (Haufe et al., 2014):

$$\mathbf{h}^{\text{pat}} : \min_{\mathbf{h},\mathbf{b}} \|A - \mathbf{t}\mathbf{h}^\top - \mathbf{1}\mathbf{b}^\top\|_F, \tag{2}$$

where $\|\cdot\|_F$ denotes the Frobenius norm. Contrary to Eq. (1), which finds an $\mathbf{h}$ maximizing the class-separability, Eq. (2) finds a pattern best explaining $\mathcal{A}$ w.r.t. concept label $t$. This is solved as linear regression task for each feature dimension, leading to

$$\mathbf{h}^{\text{pat}} = \frac{1}{\sigma_t^2 |\mathcal{X}|} \sum_{\mathbf{x},t \in \mathcal{X}} (\mathbf{a}(\mathbf{x}) - \bar{\mathcal{A}})(t - \bar{t}) \tag{3}$$

with mean latent activation $\bar{\mathcal{A}}$, mean concept label $\bar{t}$ and sample concept label variance $\sigma_t^2$, which is equal to the sample covariance between the latent activations $\mathbf{a}(\mathbf{x})$ and the concept labels $t$ divided by the sample concept label variance. In contrast to filter-CAVs, the resulting pattern-CAV is invariant under feature scaling and more robust to noise, as further outlined in Appendix B.2. Given binary concept labels, Eq. (3) simplifies to the difference of cluster means, as shown in Appendix B.3. Note, that the computation of pattern in regression manner as described in Eq. (3) allows to further incorporate prior knowledge, *e.g.*, sparseness constraints (Haufe et al., 2014).

### 3.3 2D TOY EXPERIMENTS

We demonstrate the difference between filter- and pattern-CAVs in a toy experiment inspired by Kindermans et al. (2018). We simulate $n$ activations $\mathbf{A}_i \in \mathbb{R}^2$ split equally between the concept labels $t_i \in \{+1, -1\}$ in the following manner: Each activation $\mathbf{A}_i = \mathbf{s}_i + \mathbf{D}_i$ is decomposed into a deterministic signal part $\mathbf{s}_i$ and a random (non-signal-, noise-) distractor part $\mathbf{D}_i$. The signal part $\mathbf{s}_i = \mathbb{1}(t_i = +1)(1 \ 0)^\top$ is aligned with the *x-axis*, the distractor part $\mathbf{D}_i$ is modeled by identically distributed independent two-dimensional Gaussians of mean 0, variance $\sigma^2$ in each dimension and no correlation between dimensions. The distractor contains true noise and signal related to other concepts. Both are "noise" for the concept signal estimation. We experiment with two distractor patterns in Figure 1 (*right*):

**Scaling**: We multiply values on the x-axis with scaling factor $\lambda = 10^3$, such that the signal $\mathbf{s}_i$ is scaled proportionally and therefore signal features are on a larger scale than distractor features. The filter-CAV diverges from the true concept direction $(1 \ 0)^\top$, as the entry of the weight vector in direction of the signal scales anti-proportionally to the scaling factor in logistic regression (see Appendix B.6 for the derivation). Feature normalization is commonly disregarded in CAV training.

**Noise Rotation**: We add another distractor term $\mathbf{D}_i^{rot} = \mathbf{r}_\tau \varepsilon_i$ with $\varepsilon_i \overset{i.i.d.}{\sim} \mathcal{N}(0, 1)$, which is oriented parallel to the vector $\mathbf{r}_\tau = (\sin\tau \ \cos\tau)^\top$. This rotates the distractor direction based on $\tau$. Only the pattern-CAV $\mathbf{h}^{\text{pat}}$ obtained via Eq. (3) precisely identifies the concept direction, while the filter-CAV prefers diverging directions which increase the angle to $\mathbf{r}_\tau$, thus minimizing the variance of the datapoints in direction of the weight vector (see Appendix B.7 for the mathematical derivation).

Moreover, filter-based CAVs face further challenges, including sensitivity to regularization strength and random seeds, particularly in low data scenarios, as demonstrated in Appendix B.4.

## 4 EXPERIMENTS

After describing our experimental setup (Section 4.1), we measure how precise CAVs represent true concept directions (Section 4.2), as well as the impact of CAVs on applications, including concept sensitivity testing with TCAV (Section 4.3.1) and CAV-based model correction (Section 4.3.2).

### 4.1 EXPERIMENT DETAILS

We conduct experiments with three controlled and one real-word datasets. For the former, we insert artificial concepts into ISIC2019 (Codella et al., 2018; Tschandl et al., 2018; Combalia et al., 2019), a dermatologic dataset for skin cancer detection with images of benign and malignant lesions, and a Pediatric Bone Age dataset (Halabi et al., 2019), with the task to predict bone age based on hand radiographs. Specifically, we insert timestamps as a text layover into 1% of samples of class

Figure 2: Example for timestamp artifact inserted into ISIC2019 samples (*left*) and RelMax visualization for neurons (*right*) corresponding to the largest absolute values in filter- and pattern-CAVs, along with the Conv filter ID and the fraction of all (absolute) CAV values. While the filter-CAV picks up noisy neurons, the pattern-CAV uses neurons related to the relevant concept.

"Melanoma" of ISIC2019, encouraging the model to learn the timestamps as a shortcut. For the Bone Age dataset, we insert an unlocalizable concept by increasing the brightness (*i.e.*, increase pixel values) of 20% of samples of only one class. We implemented bone age prediction as a classification task, with target ages binned into five equal-sized groups. Lastly, we use FunnyBirds (Hesse et al., 2023), a synthetic dataset with part-level annotations, to synthesize a dataset with 10 classes of birds, where each category is defined by exactly one part (*e.g.*, wings, beak). Other parts are chosen randomly per sample, forcing the model to use the class-defining part (*i.e.*, concept) as the only valid feature. Detailed class definitions and examples for synthesized images are provided in Appendix C.1. Further, we consider real data artifacts present in ISIC2019, including "band-aid", "ruler", and "skin marker". We finetune VGG16 (Simonyan & Zisserman, 2015), ResNet18/50 (He et al., 2016; Wightman et al., 2021), ResNeXt50 (Xie et al., 2017), ReXNet100 (Han et al., 2021), EfficientNet-B0 (Tan & Le, 2019), EfficientNetV2-(Tan & Le, 2021), and Vision Transformer (Dosovitskiy et al., 2020) models pre-trained on ImageNet (Deng et al., 2009; Ridnik et al., 2021) for all datasets with training details given in Appendix C.2.

## 4.2 PRECISENESS OF CONCEPT REPRESENTATION

The primary goal of Pattern-CAVs is the optimization of the precision of concept representations. Therefore, to assess how precisely CAVs represent true concept directions both qualitatively and quantitatively, we (1) visualize the key neurons associated with the CAV, and (2) quantify the alignment between CAVs and the ground truth direction.

**How clean are CAVs qualitatively?** We investigate the focus of CAV $\mathbf{h}$ fitted on layer $l$ by employing feature visualization to neurons corresponding to the largest absolute, hence most impactful, values in $\mathbf{h}$. Specifically, we use RelMax (Achtibat et al., 2023) to retrieve input samples maximizing the relevance, computed by feature attribution methods, for the neurons with the largest absolute values in $\mathbf{h}$. We further use receptive field information to zoom into the most relevant region and mask out irrelevant information. Results for filter- and pattern-based CAVs for the timestamp artifact in ISIC2019 are shown in Fig. 2. Whereas the pattern-CAV leverages neurons focusing on the desired concept, *i.e.*, the timestamp, the filter-CAV is distracted by other features. In addition, we show the percentage for the value associated with the neuron of the entire CAV and higher values, as observed for Pattern-CAV, indicate a less uniform distribution over neurons and a larger focus on the corresponding top neurons. Additional neuron visualizations for different filter- and pattern-CAVs are shown in Appendix D.2.

**CAV Alignment with True Concept Direction** Using our controlled datasets, we generate pairs of samples with and without the concept ($\mathbf{x}_i^+$ and $\mathbf{x}_i^-$) and compute the sample-wise true latent concept direction $\mathbf{h}_i^{\text{gt}} = \mathbf{a}(\mathbf{x}_i^+) - \mathbf{a}(\mathbf{x}_i^-)$.[2] This definition aligns with TCAV's intuition, *i.e.*, adding activations along the concept direction corresponds to adding the concept in input space. To quantify the alignment between CAV $\mathbf{h}$ and $\mathbf{h}_i^{\text{gt}}$ per sample, we use cosine similarity as the similarity function

---

[2]In FunnyBirds, we remove concepts by randomizing the class-defining part, while keeping others identical.

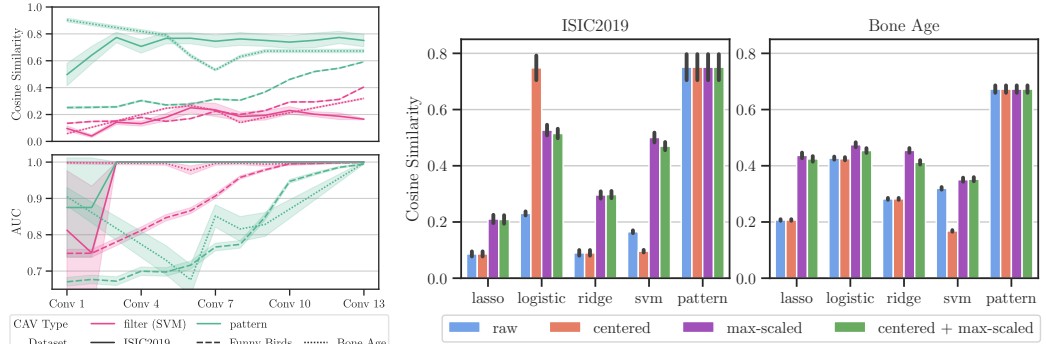

Figure 3: *Left:* Comparison of cosine similarity between CAVs and true concept direction (*top*) and concept separability (*bottom*), using filter- (SVM) and pattern-CAV for all Conv layers of VGG16 trained on ISIC2019, Bone Age, and FunnyBirds. While expectedly filter-CAVs have superior class-separability, pattern-CAVs have a better alignment with the true concept direction. *Right*: Cosine similarity between true concept direction $\mathbf{h}_i^{\text{gt}}$ and CAVs with different feature pre-processing methods fitted on the last Conv layer of VGG16 trained on ISIC2019 and Bone Age. Compared to filter-CAVs, pattern-CAV has a higher alignment with $\mathbf{h}_i^{\text{gt}}$ and is invariant to feature pre-processing.

$\text{sim}(\mathbf{h}, \mathbf{h}_i^{\text{gt}})$. We calculate the overall alignment $\bar{a}$ by averaging the alignment scores for all samples:

$$\bar{a} = 1/|\mathcal{X}| \sum_i \text{sim}(\mathbf{h}, \mathbf{h}_i^{\text{gt}}) \, . \tag{4}$$

Moreover, we measure the separability of samples w.r.t. concept label $t$ by computing the AUC of $\mathbf{h}^\top \mathbf{a}(\mathbf{x})$. Fig. 3 (*left*) presents the results, including standard errors, for both CAV alignment (*top*) and separability (*bottom*) across all 13 convolutional (Conv) layers in the VGG16 models for all three controlled datasets. We estimate standard errors of AUC scores using the Wilcoxon-Mann-Whitney statistic as an equivalence (Cortes & Mohri, 2004). The alignment with $\mathbf{h}_i^{\text{gt}}$ is significantly higher for pattern-based CAVs across all layers for all datasets, confirming a more precise estimation of the true concept direction. As expected, filter-based CAVs exhibit higher concept separability. Additional experiments in Appendix D.3 demonstrate the superior robustness of pattern-CAVs towards the reduction of concept set sizes and concept labeling errors. Moreover, the experiments show that pattern-CAVs outperform concept directions found in unsupervised manner in terms of precision. We further investigate the relation between the distribution of noise in the activations and the divergence of estimated concept directions from the true concept direction in Appendix D.6.

Moreover, we study the sensitivity of CAVs to different pre-processing methods for latent activations, specifically centering, max-scaling, and their combination. Results for CAVs fitted on the last Conv layer of VGG16 for ISIC2019 and Bone Age are shown in Fig. 3 (*right*). While filter-CAVs have better alignment with true concept directions when features are re-scaled, which is often overlooked in practice, pattern-CAVs consistently outperform filter-CAVs regardless of activation pre-processing. This can be attributed to the fact that the covariance (see Eq. (3)) is translation invariant, while scale invariance is proven in Appendix B.2. Another disadvantage of filter-CAVs is their dependence on hyperparameters, *e.g.*, regularization strength. In contrast, pattern-CAVs do not require parameter tuning and are therefore more computationally efficient.

### 4.3 IMPACT OF DIRECTIONAL SHIFTS ON CAV APPLICATIONS

We measure the impact of different CAVs on applications requiring precise concept directions, namely concept sensitivity testing with TCAV and concept-based model correction with ClArC.

#### 4.3.1 TESTING WITH CAV

TCAV (Kim et al., 2018) is a technique to assess the sensitivity of a DNNs' prediction w.r.t. a given concept represented by CAV $\mathbf{h}$. Specifically, given the directional derivative $\nabla_{\mathbf{a}} \tilde{f}(\mathbf{a}(\mathbf{x}))$, we measure

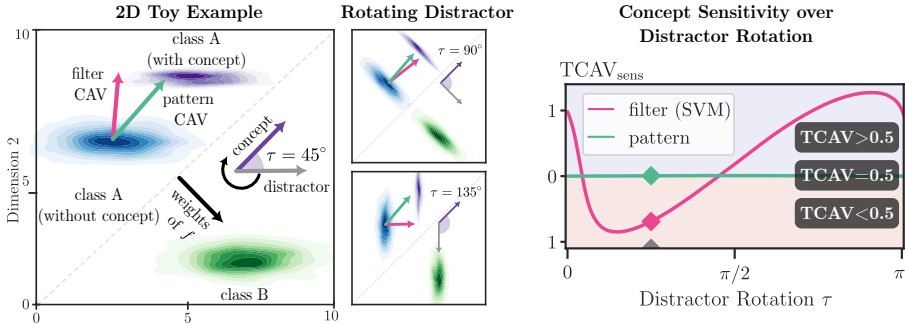

Figure 4: *Left:* 2D TCAV experiment with distractor rotated by $\tau = 45°$ with samples from class A (purple with concept, blue without concept) and class B (green). The model $f$ classifies between classes A and B. CAVs are fitted on samples with and without concept from class A. The pattern-CAV aligns with the concept direction, while the filter-CAV diverges to optimize class-separability. *Right:* $\text{TCAV}_{\text{sens}}$ for model $f$ plotted over distractor rotation $\tau$. Positive and negative values indicate a positive and negative influence of the concept direction and 0 indicates insensitivity (TCAV = 0.5).

the model's sensitivity towards the concept for a sample $\mathbf{x}$ as

$$\text{TCAV}_{\text{sens}}(\mathbf{x}) = \boldsymbol{\nabla}_{\mathbf{a}} \tilde{f}(\mathbf{a}(\mathbf{x})) \cdot \mathbf{h} \, . \tag{5}$$

The TCAV score measures the fraction of the sample subset containing the concept $\mathcal{X}^+ = \{\mathbf{x}_i \in \mathcal{X} \mid t_i = +1\}$ where the model shows positive sensitivity towards changes along the estimated concept direction $\mathbf{h}$:

$$\text{TCAV} = \frac{|\{\mathbf{x} \in \mathcal{X}^+ \mid \text{TCAV}_{\text{sens}}(\mathbf{x}) > 0\}|}{|\mathcal{X}^+|} \, . \tag{6}$$

Hence, to truthfully measure the model's sensitivity, a precise estimated concept direction $\mathbf{h}$ is required. A TCAV score $\approx 0.5$ indicates minimal influence of the concept on the model's decisions, while scores above and below $0.5$ indicate positive and negative impacts. We show the effects of directional divergence by conducting experiments in 2D and with our controlled FunnyBirds dataset.

**TCAV in 2D Toy Experiment**    Consider samples $\mathbf{x} \in \mathbb{R}^2$ with class labels $y \in \{+1, -1\}$, referred to as class A and B, perfectly separable by a linear model $f$ with weights $\mathbf{w} = (1 \;\; -1)^\top$ and bias $b = 0$. We introduce a data artifact in class A where some samples contain concept $c$ with concept direction $\mathbf{c} = (1 \;\; 1)^\top$ perpendicular to $\mathbf{w}$. As $f$ is insensitive to concept $c$, we expect a TCAV score of 0.5. Using the notation from Section 3.3, we rotate the distractor $\mathbf{r}_\tau$ with $\tau \in [0, \pi]$ relative to $\mathbf{c}$. CAVs are fitted to separate samples with and without $c$ from class A, using concept labels $t$ instead of class labels $y$. Results are shown in Fig. 4. For $\tau = \pi/4 = 45°$(*left*), $\mathbf{h}^{\text{pat}}$ aligns with the concept direction $\mathbf{c}$, whereas $\mathbf{h}^{\text{filt}}$ diverges significantly from the true concept direction. Plotting the models sensitivity towards $c$, here measured as $\text{TCAV}_{\text{sens}} = \mathbf{w}^\top \mathbf{h}$, with $\mathbf{w}$ as the gradient of $f$ w.r.t. $\mathbf{x}$, over $\tau$ (*right*), we observe that $\mathbf{h}^{\text{pat}}$ consistently achieves $\text{TCAV}_{\text{sens}} = 0$ (corresponding to the expected TCAV score 0.5), while for $\mathbf{h}^{\text{filt}}$, the sensitivity towards $c$ incorrectly depends on $\tau$. These results demonstrate that relying on the widely used SVM-CAVs (i.e., filter-CAV) may produce arbitrary TCAV scores, making the concept sensitivity testing procedure highly unreliable. In contrast, our proposed pattern-CAV is invariant to the distractors and leads to consistent TCAV scores.

**Controlled Experiment with FunnyBirds**    The comparison of TCAV scores computed with different CAV methods requires ground truth information on the true concept sensitivity, which is commonly unavailable for DNNs. To address this, we use our FunnyBirds dataset designed to *enforce* certain concepts, as for each class all concepts but one are randomized per sample. For each class $k$, we define a subset $\mathcal{X}_k = \{\mathbf{x}_i \in \mathcal{X} \mid y_i = k\}$ and compute a TCAV score w.r.t. to the class-defining concept (see Appendix C.1). These TCAV scores are expected to be $\neq 0.5$, as the concepts are the only valid features. Fig. 5 (*left*) presents the results averaged across all 10 classes with pattern-CAVs

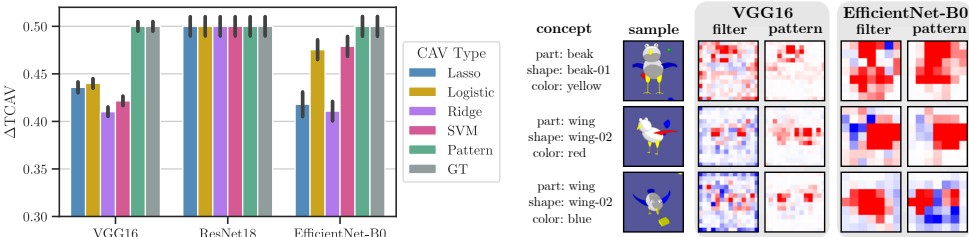

Figure 5: *Left:* $\Delta$TCAV (averaged over class-defining concepts) for different CAVs fitted on last Conv layers of VGG16, ResNet18, and EfficientNet-B0 trained on FunnyBirds. As models *must* use these concepts by experimental design, high scores are better. In contrast to filter-CAVs, pattern-CAVs achieve best scores for all models. *Right:* Concept-sensitivity maps, measured as element-wise product $\nabla_{\mathbf{a}}\tilde{f}(\mathbf{a}(\mathbf{x})) \odot \mathbf{h}$ using filter- and pattern-CAVs for three concepts with VGG16 and EfficientNet-B0. Results are shown for the last Conv layer, upsampled to input space dimensions. While pattern-CAVs precisely localize the concepts, filter-CAVs lead to noisy sensitivity maps.

and different filter-CAVs computed for the last Conv layers of VGG16, ResNet18, and EfficientNet-B0. Additionally, we report the TCAV scores using the ground truth concept direction $\mathbf{h}_i^{\mathrm{gt}}$ (*GT*). The TCAV score is reported as $\Delta$TCAV $= |$TCAV $- 0.5|$, *i.e.* the delta from the score representing no sensitivity. Higher values reflect a stronger impact on the model's decision and are expected in this experiment. To evaluate statistical significance, we run a two-sided t-test and found that all $\Delta$TCAV scores are significantly different from the random baseline score of 0. We report the corresponding p-values, accuracies for filter-CAVs on the test set, and results for additional model architectures in Appendix D.5. While for VGG16 and EfficientNet-B0, pattern-based CAVs achieve a perfect score of 0.5, the TCAV score for filter-based CAVs does not fully indicate the model's dependence on the concept. Interestingly, all CAV variants achieve a perfect score for ResNet18, which can be explained by not well localized concepts, as further qualitative investigations in Appendix D.4 indicate.

The above observations are supported by qualitative results in Fig. 5 (*right*), where pattern-CAVs precisely localize concepts and measure positive concept sensitivity (*red*) correctly. In contrast, filter-CAVs produce noisy concept-sensitivity maps, negatively impacting the TCAV score. This is because TCAV$_{\mathrm{sens}}(\mathbf{x})$ for sample $\mathbf{x}$ is computed over all elements of the concept-sensitivity map $\nabla_{\mathbf{a}}\tilde{f}(\mathbf{a}(\mathbf{x})) \odot \mathbf{h}$. For instance, for the "wing"-concept samples ($2^{\mathrm{nd}}$ and $3^{\mathrm{rd}}$ row), the dominance of negative sensitivity (*blue*) caused by noise over positive sensitivity (*red*) in VGG16's filter-CAVs leads to an incorrect negative overall concept sensitivity.

### 4.3.2 CAV-based Model Correction (ClArC)

The ClArC framework (Anders et al., 2022) uses CAVs to model data artifacts in latent space to unlearn shortcuts, *i.e.*, prediction strategies based on artifacts present in the training data with unintended relation to the task. Specifically, Right Reason ClArC (RR-ClArC) (Dreyer et al., 2024) is a recent approach that finetunes the model with an additional loss term $L_{\mathrm{RR}}(\mathbf{x}) = \left(\nabla_{\mathbf{a}}\tilde{f}(\mathbf{a}(\mathbf{x})) \cdot \mathbf{h}\right)^2$. This loss term penalizes the use of latent features, measured via the gradient, pointing into the direction of CAV $\mathbf{h}$, representing the data artifact. An accurate estimated concept direction is crucial to ensure that the intended direction is penalized. Hence, we intentionally poison models by encouraging them to use our controllable concepts (timestamp and brightness) as shortcuts, followed by the application of ClArC to unlearn these concepts. We further correct models trained on ISIC2019 w.r.t. the known artifacts "band-aid", "ruler", and "skin marker" with artifact-specific CAVs. Training details are given in Appendix D.7.

**Quantitative Evaluation** We evaluate the effectiveness of model correction with different CAVs by studying the impact of data poisoning on the model's accuracy and its sensitivity to data artifacts. For the former, we measure the accuracy on a clean (artifact-free) and a biased test set, with the artifact inserted into *all* samples. For the real artifacts in ISIC2019, we automatically compute input localization masks (Pahde et al., 2023) to cut (localizable) artifacts from known artifact samples and paste them onto clean test samples. To probe the model's sensitivity to the artifact, we measure the

Table 1: Model correction results with RR-ClArC for VGG16, ResNet50, EfficientNet-B0, and ViT trained on Bone Age | ISIC2019 (controlled) | ISIC2019 (real). We report accuracy on clean and biased test set, the fraction of relevance on the region of localizable artifacts, and the TCAV score (as $\Delta\text{TCAV}^{\text{gt}}$) with the sample-wise ground-truth concept direction $\mathbf{h}_i^{\text{gt}}$, measuring the models' sensitivity towards the artifacts *after* model correction. Stars indicate statistical significance according to z-tests with significance level 0.05, and arrows whether low (↓) or high (↑) are better.

| model | CAV | Accuracy (clean) ↑ | | | Accuracy (biased) ↑ | | | Artifact relevance ↓ | | | $\Delta\text{TCAV}^{\text{gt}}$ ↓ | | |
|---|---|---|---|---|---|---|---|---|---|---|---|---|---|
| VGG-16 | *Vanilla* | 0.78 | 0.82 | 0.83 | 0.50 | 0.28 | 0.75 | - | 0.62 | 0.51 | 0.29 | 0.14 | 0.10 |
| | lasso | 0.77 | 0.82 | 0.82 | 0.55 | 0.30 | 0.76 | - | 0.60 | 0.49 | 0.25 | 0.13 | 0.12 |
| | logistic | 0.72 | 0.82 | 0.82 | 0.63 | 0.37 | 0.78 | - | 0.54 | 0.43 | 0.25 | **0.07***| 0.09 |
| | ridge | 0.71 | 0.82 | 0.82 | 0.61 | 0.31 | 0.76 | - | 0.59 | 0.50 | 0.24 | 0.13 | 0.12 |
| | SVM | 0.69 | 0.81 | 0.82 | 0.70 | 0.36 | 0.78 | - | 0.55 | 0.46 | 0.24 | 0.10 | 0.11 |
| | Pattern (ours) | 0.78 | 0.80 | 0.82 | **0.75***| **0.69***| **0.79** | - | **0.26***| **0.31***| **0.14***| 0.10 | **0.03*** |
| ResNet50 | *Vanilla* | 0.77 | 0.85 | 0.87 | 0.48 | 0.51 | 0.82 | - | 0.46 | 0.34 | 0.14 | 0.04 | 0.27 |
| | lasso | 0.77 | 0.84 | 0.87 | 0.53 | 0.58 | 0.82 | - | 0.44 | 0.30 | 0.03 | **0.02** | 0.05 |
| | logistic | 0.77 | 0.85 | 0.87 | 0.55 | 0.69 | **0.83** | - | 0.39 | 0.24 | 0.04 | 0.05 | 0.05 |
| | ridge | 0.77 | 0.84 | 0.87 | 0.52 | 0.58 | 0.82 | - | 0.45 | 0.30 | 0.13 | 0.03 | 0.05 |
| | SVM | 0.77 | 0.85 | 0.87 | 0.55 | 0.68 | **0.83** | - | 0.40 | 0.26 | 0.04 | 0.04 | 0.05 |
| | Pattern (ours) | 0.78 | 0.84 | 0.87 | **0.59***| **0.71** | 0.83 | - | **0.37***| **0.22***| **0.01** | 0.03 | **0.04** |
| EfficientNet-B0 | *Vanilla* | 0.79 | 0.87 | 0.88 | 0.46 | 0.55 | 0.83 | - | 0.55 | 0.22 | 0.46 | 0.39 | 0.12 |
| | lasso | 0.79 | 0.86 | 0.88 | 0.70 | 0.64 | **0.83** | - | 0.52 | **0.22** | 0.01 | 0.11 | 0.11 |
| | logistic | 0.77 | 0.85 | 0.88 | **0.75** | 0.67 | **0.83** | - | 0.51 | **0.22** | **0.00** | **0.02** | 0.12 |
| | ridge | 0.78 | 0.82 | 0.88 | 0.74 | 0.67 | **0.83** | - | 0.52 | **0.22** | 0.21 | 0.12 | 0.12 |
| | SVM | 0.77 | 0.85 | 0.88 | **0.75** | 0.65 | **0.83** | - | 0.52 | **0.22** | **0.00** | 0.03 | 0.11 |
| | Pattern (ours) | 0.77 | 0.85 | 0.88 | **0.75** | **0.72***| 0.83 | - | **0.48***| 0.22 | **0.00** | 0.05 | **0.03*** |
| ViT | *Vanilla* | 0.73 | 0.88 | 0.89 | 0.38 | 0.67 | 0.82 | - | 0.15 | 0.10 | 0.47 | 0.25 | 0.05 |
| | lasso | 0.74 | 0.88 | 0.89 | 0.39 | 0.67 | **0.82** | - | **0.16** | **0.10** | 0.42 | **0.25** | 0.06 |
| | logistic | 0.73 | 0.88 | 0.89 | **0.62** | 0.72 | **0.82** | - | **0.16** | **0.10** | **0.02***| **0.25** | **0.04** |
| | ridge | 0.74 | 0.88 | 0.89 | 0.48 | 0.66 | **0.82** | - | **0.16** | **0.10** | 0.12 | **0.25** | 0.06 |
| | SVM | 0.73 | 0.87 | 0.89 | 0.48 | 0.61 | **0.82** | - | 0.22 | **0.10** | 0.46 | 0.50 | **0.04** |
| | Pattern | 0.74 | 0.88 | 0.89 | 0.61 | **0.73** | **0.82** | - | **0.16** | **0.10** | 0.06 | **0.25** | 0.11 |

fraction of relevance, computed with Layer-wise Relevance Propagation (LRP) (Bach et al., 2015) for convolutional architectures and SHapley Additive exPlanations (SHAP) (Lundberg & Lee, 2017) for transformer-based models, on the artifact region using our localization masks. No artifact relevance is reported for the brightness artifact, as it is considered unlocalizable. Moreover, we compute the TCAV score *after* model correction using the ground truth concept direction $\mathbf{h}_i^{\text{gt}}$. For real artifacts, the ground truth direction is computed for "attacked" samples $\mathbf{x}^{\text{att}}$ with artificially inserted artifacts, as $\mathbf{h}_i^{\text{gt}} = \mathbf{a}(\mathbf{x}_i^{\text{att}}) - \mathbf{a}(\mathbf{x}_i)$. The model correction results for VGG16, ResNet50, EfficientNet-B0, and Vision Transformer (ViT) for ISIC2019 (timestamp artifact, *controlled*), Pediatric Bone Age (brightness, *controlled*), and ISIC2019 ("band-aid", *real*) are shown in Table 1. We perform model correction on one of the last three Conv layers for the former three architectures, and on the last fully-connected linear layer for ViT. We use filter-based (lasso, logistic, ridge regression, and SVM) and pattern-based CAVs as introduced in Eq. (3) to represent the direction to be unlearned. The models are finetuned and compared to a Vanilla model, which is trained without added loss term. Further training details are provided in Appendix C.2. For VGG16, the accuracy on clean test sets remains largely unaffected, while pattern-CAVs outperform other methods in terms of accuracy on the biased test set. Moreover, pattern-CAVs yield best results for reduced artifact sensitivity, measured through artifact relevance and $\Delta\text{TCAV}^{\text{gt}}$. Similar artifact sensitivity results can be observed for the other architectures. Furthermore, pattern-CAVs achieve the highest accuracies on biased test sets in the controlled settings. For the "band-aid" artifact, all CAVs yield similar accuracy scores on both clean and biased test sets. This can be attributed to the minimal impact of the artifact on EfficientNet-B0 and ResNet50, as indicated by the small accuracy difference between the two test sets. Detailed results with standard errors for additional model architectures, *e.g.*, ResNeXt50, ReXNet100, and EfficientNetV2, are shown in Appendix D.7.

**Qualitative Evaluation** We compare attribution heatmaps for the Vanilla model with heatmaps for models corrected with RR-ClArC using filter- (SVM) and pattern-based CAVs w.r.t. the band-

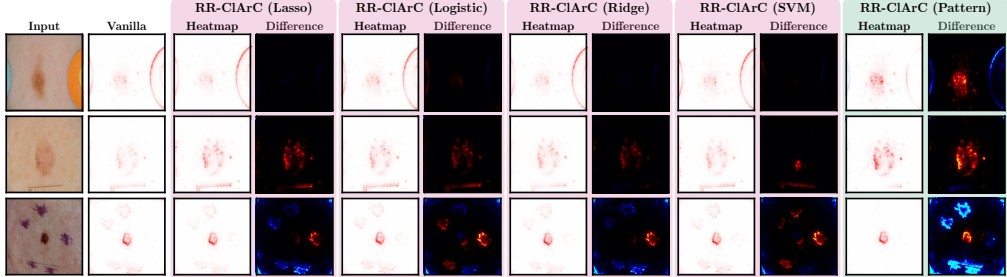

Figure 6: Qualitative results for model correction w.r.t. real artifacts band-aid (*top*), ruler (*middle*), and skin marker (*bottom*) in ISIC2019 using VGG16. In addition to attribution heatmaps for models corrected with filter- and pattern-CAVs, we show heatmaps highlighting the differences compared to the Vanilla model attribution heatmap, with red and blue indicating higher and lower relevance after correction, respectively. Whereas filter-CAVs have limited impact, pattern-CAVs successfully increases the relevance on the mole and decreases the relevance on data artifacts.

aid, ruler, and skin marker artifacts using the VGG16 model trained on ISIC2019 in Fig. 6. In addition to the attribution heatmap computed with LRP using the $\varepsilon z^+\flat$-composite (Kohlbrenner et al., 2020) in `zennit` (Anders et al., 2021), we show another heatmap highlighting the difference between the normalized relevance heatmaps of the corrected and the Vanilla model, with blue and red showing areas with lower and higher relevance after correction. Pattern-CAVs reduce the relevance of data artifacts after model correction significantly, while traditional SVM-CAVs have little impact. Additional examples are shown in Appendix D.7.

## 5    CONCLUSION

While filters from linear classifiers can accurately predict the presence of concepts, they fall short in precisely modeling the direction of the concept signal. As many applications of CAVs, including TCAV and ClArC, heavily rely on accurate concept directions, we address this drawback by introducing pattern-based CAVs, which disregard distractor signals and focus solely on the concept signal. We provide both theoretical and empirical evidence to support the improved estimation of the true concept direction compared to widely used filter-based CAVs. Furthermore, we demonstrate the positive impact on applications leveraging CAVs, such as estimating the model's sensitivity towards concepts and correcting model shortcut behavior caused by data artifacts. Future research might explore the optimization of concept directions beyond binary labels, the incorporation of prior knowledge, semi-supervised concept discovery, and the disentanglement of correlated concept directions.

**Limitations**    Our results confirm that pattern-CAVs exhibit superior alignment with ground truth concept directions compared to filter-CAVs. This has a positive impact on CAV applications heavily relying on precise concept directions, such as concept sensitivity testing (TCAV) and model correction with ClArC. However, for CAV applications in which class-separability is more important, *i.e.*, determining whether a concept is present in a given sample, filter-based CAVs might be a better choice. For instance, post-hoc concept bottleneck models (Yuksekgonul et al., 2023) project latent embeddings into an interpretable concept space spanned by CAVs and fit a linear classifier in the resulting concept space. The linear classifier can handle directional divergence in CAVs and requires a precise decision hyperplane, making filter-based CAVs superior in such scenarios. Thus, the choice of CAV computation methods should be carefully considered based on the specific task at hand.

## ACKNOWLEDGEMENTS

This work was supported by the Federal Ministry of Education and Research (BMBF) as grant BIFOLD (01IS18025A, 01IS180371I); the German Research Foundation (DFG) as research unit DeSBi [KI-FOR 5363] (459422098); the European Union's Horizon Europe research and innovation programme (EU Horizon Europe) as grant TEMA (101093003); and the European Union's Horizon 2020 research and innovation programme (EU Horizon 2020) as grant iToBoS (965221).

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

# APPENDIX

## A  BROADER IMPACT

This work presents drawbacks of widely used filter-based concept activation vectors (CAV), specifically their tendency to deviate from the true concept direction. To address this, our paper introduces robust pattern-based CAVs, providing more accurate concept directions. This advancement directly impacts safety-critical CAV applications like concept sensitivity testing and model debugging, thereby promoting the transparency, accountability, and understandability of deep neural networks. Ultimately, this work contributes to increasing the trustworthiness of AI and advancing the development of reliable and explainable AI systems, extending its impact to societal dimensions.

## B  METHODS

In the following, we provide additional details and proofs related to our methods. Specifically, we provide details for linear models considered for filter-based CAVs in Section B.1, prove the robustness to noise and scaling of pattern-CAVs in Section B.2, and further prove that in case of binary target labels the pattern is equivalent to the difference of cluster means in Section B.3. Moreover, we present additional 2D toy experiments in Section B.4, scale the toy experiment to high dimension in Section B.5, and provide proofs of divergence for filter-based approaches in Section B.6 and Section B.7 for feature scaling and noise rotation, respectively.

### B.1  DETAILS FOR FILTER-BASED CAV APPROACHES

We briefly summarize the optimization objectives for linear models for filter-based CAV approaches, including lasso, logistic, and ridge regression, as well as SVMs. All methods aim to find a hyperplane that separates a dataset $\mathcal{X} \subset \mathbb{R}^m$ of size $n$ into two sets, defined by their concept label $t \in \{+1, -1\}$. This is achieved by fitting a weight vector $\mathbf{w} \in \mathbb{R}^m$ and a bias $b \in \mathbb{R}$ such that the hyperplane consists of all $\mathbf{x}$ which satisfy $\mathbf{w}^\top \mathbf{x} + b = 0$.

Lasso regression (Tibshirani, 1996) aims to minimize residuals $r_i = \left(t_i - (\mathbf{w}^\top \mathbf{x}_i + b)\right)$ with $L_1$-norm regularization, thereby encouraging sparse coefficients. The optimization problem is given by

$$\min_{\mathbf{w},b} \left\{ \frac{1}{n} \sum_{i \in [n]} r_i^2 + \lambda \sum_{j \in [m]} |w_j| \right\}. \tag{7}$$

Similarly, ridge regression (Hoerl & Kennard, 1970) fits a linear model which minimizes the residuals $r_i$ with $L_2$-norm regularization by solving

$$\min_{\mathbf{w},b} \left\{ \frac{1}{n} \sum_{i \in [n]} r_i^2 + \lambda \sqrt{\sum_{j \in [m]} w_j^2} \right\}. \tag{8}$$

The logistic regression model estimates probabilities via

$$\widehat{\mathbb{P}}_{\mathbf{w},b}(t = +1 \mid \mathbf{x}) = \frac{e^{\mathbf{w}^\top \mathbf{x} + b}}{1 + e^{\mathbf{w}^\top \mathbf{x} + b}} = \sigma(\mathbf{w}^\top \mathbf{x} + b),$$

where $\sigma$ denotes the sigmoid function $\sigma(z) = \frac{e^z}{1+e^z}$. The linear model is now fitted by maximizing the log likelihood of the observed data:

$$
\begin{aligned}
\max_{\mathbf{w},b} L(\mathbf{w}, b; \mathcal{X}) = \max_{\mathbf{w},b} \sum_{i \in [n]} & \mathbb{1}(t_i = +1) \log \widehat{\mathbb{P}}_{\mathbf{w},b}(t = +1 \mid \mathbf{x}_i) \\
& + \mathbb{1}(t_i = -1) \log \left(1 - \widehat{\mathbb{P}}_{\mathbf{w},b}(t = +1 \mid \mathbf{x}_i)\right).
\end{aligned}
\tag{9}
$$

Lastly, and most commonly used for CAVs, SVMs (Cortes & Vapnik, 1995) fit a linear model by finding a hyperplane that maximizes the margin between two classes using the hinge loss, defined

as $l_i = \max\left(0, 1 - t_i\left(\mathbf{w}^\top \mathbf{x}_i + b\right)\right)$ and $L_2$-norm regularization with the following optimization objective:

$$\min_{\mathbf{w},b} \left\{ \frac{1}{n} \sum_{i \in [n]} l_i + \lambda \sqrt{\sum_{j \in [m]} w_j^2} \right\}. \tag{10}$$

## B.2 Feature Scaling and Noise on Pattern-CAV

In this section, we investigate the effect of feature scaling and additive noise on the resulting pattern-CAV. We start from the known solution for a simple linear regression task provided in Equation 3, resulting in a pattern-CAV of

$$\mathbf{h}^{\text{pat}} = \frac{1}{\sigma_t^2 |\mathcal{X}|} \sum_{\mathbf{x},t \in \mathcal{X}} (\mathbf{a}(\mathbf{x}) - \bar{A})(t - \bar{t}). \tag{11}$$

**Feature Scaling** We start with the effect of feature scaling on $\mathbf{h}^{\text{pat}}$. Specifically, we investigate the effect on the CAV, when we scale a specific dimension $k$ of features $\mathbf{a} \in \mathbb{R}^m$ (of dimension $m$) with a factor $\gamma$, *i.e.*,

$$a_i^\gamma = \begin{cases} a_i & \text{if } i \neq k \\ \gamma a_i & \text{if } i = k. \end{cases} \tag{12}$$

Then, with Eq. (11), we get for the corresponding pattern-CAV

$$\begin{aligned} (h^\gamma)_i^{\text{pat}} &= \frac{1}{\sigma_t^2 |\mathcal{X}|} \sum_{\mathbf{x},t \in \mathcal{X}_\mathbf{h}} (a_i^\gamma(\mathbf{x}) - \bar{a}_i^\gamma)(t - \bar{t}) \\ &= \begin{cases} h_i^{\text{pat}} & \text{if } i \neq k \\ \gamma h_i^{\text{pat}} & \text{if } i = k. \end{cases} \end{aligned} \tag{13}$$

meaning that the CAV $\mathbf{h}^{\text{pat}}$ scales *with* the features (contrary to many classification-based CAVs).

**Additive Noise** We add random noise $\epsilon$ with zero mean $\mathbb{E}[\epsilon] = 0$, that is independent to the concept labels $t$ to a feature dimension $k$. Then in expectation

$$\begin{aligned} \mathbb{E}[h'^{\text{pat}}_i] &= \frac{1}{\sigma_t^2} \mathbb{E}[(a_i - \bar{a}_i + \delta_{ik}\epsilon)(t - \bar{t})] \\ &= \frac{1}{\sigma_t^2} \left[ \mathbb{E}[(a_i - \bar{a}_i)(t - \bar{t})] + \mathbb{E}[\epsilon_k(t - \bar{t})] \right] \\ &= \frac{1}{\sigma_t^2} \mathbb{E}[(a_i - \bar{a}_i)(t - \bar{t})] = h_i^{\text{pat}}, \end{aligned} \tag{14}$$

where we used the independence of $\epsilon$ and $t$, *i.e.*, $\mathbb{E}[\epsilon t] = \mathbb{E}[\epsilon]t = 0$.

## B.3 Pattern-CAV Reducing to the Difference of Means

Assuming for easier notation we have binary concept labels $t \in \{0, 1\}$. We start from Eq. (3) for the pattern-CAV, given by the known solution for a simple linear regression task given as

$$\mathbf{h}^{\text{pat}} = \frac{1}{\sigma_t^2 |\mathcal{X}|} \sum_{\mathbf{x},t \in \mathcal{X}} (\mathbf{a}(\mathbf{x}) - \bar{A})(t - \bar{t}). \tag{15}$$

For the sample covariance term, we get

$$\frac{1}{|\mathcal{X}|} \sum_{\mathbf{x},t \in \mathcal{X}} (\mathbf{a}(\mathbf{x}) - \bar{A})(t - \bar{t}) = \frac{1}{|\mathcal{X}|} \left[ \sum_{\mathbf{x},t \in \mathcal{X}^+} (\mathbf{a}(\mathbf{x}) - \bar{A})(1 - \bar{t}) - \sum_{\mathbf{x},t \in \mathcal{X}^-} (\mathbf{a}(\mathbf{x}) - \bar{A})\bar{t} \right]. \tag{16}$$

We have $|\mathcal{X}^+|$ positive (with concept) sample activations $\mathcal{A}^+$ and $|\mathcal{X}^-|$ negative (without concept) sample activations $\mathcal{A}^-$. We further introduce $\alpha^{\pm} = \frac{|\mathcal{X}^{\pm}|}{|\mathcal{X}|}$, therefore, $\bar{t} = \alpha^+$ and $1 - \bar{t} = \alpha^-$ using $t \in \{0, 1\}$. Thus, we can write

$$
\begin{aligned}
\frac{1}{|\mathcal{X}|} \sum_{\mathbf{x},t \in \mathcal{X}} (\mathbf{a}(\mathbf{x}) - \bar{\mathcal{A}})(t - \bar{t}) &= \frac{1}{|\mathcal{X}|} \left[ \alpha^- \sum_{\mathbf{x},t \in \mathcal{X}^+} (\mathbf{a}(\mathbf{x}) - \bar{\mathcal{A}}) - \alpha^+ \sum_{\mathbf{x},t \in \mathcal{X}^-} (\mathbf{a}(\mathbf{x}) - \bar{\mathcal{A}}) \right] \\
&= \frac{1}{|\mathcal{X}|} \left[ \alpha^- |\mathcal{X}^+|(\bar{\mathcal{A}}^+ - \bar{\mathcal{A}}) - \alpha^+ |\mathcal{X}^-|(\bar{\mathcal{A}}^- - \bar{\mathcal{A}}) \right],
\end{aligned}
\tag{17}
$$

where we used for the last step that

$$
\bar{\mathcal{A}}^{\pm} = \frac{1}{|\mathcal{X}^{\pm}|} \sum_{\mathbf{x},t \in \mathcal{X}^{\pm}} \mathbf{a}(\mathbf{x}).
\tag{18}
$$

Finally, we receive

$$
\frac{1}{|\mathcal{X}|} \sum_{\mathbf{x},t \in \mathcal{X}} (\mathbf{a}(\mathbf{x}) - \bar{\mathcal{A}})(t - \bar{t}) = \frac{|\mathcal{X}^+||\mathcal{X}^-|}{|\mathcal{X}|^2} (\bar{\mathcal{A}}^+ - \bar{\mathcal{A}}^-) = \sigma_t^2 (\bar{\mathcal{A}}^+ - \bar{\mathcal{A}}^-)
\tag{19}
$$

and

$$
\mathbf{h}^{\text{pat}} = \bar{\mathcal{A}}^+ - \bar{\mathcal{A}}^-.
\tag{20}
$$

### B.4 2D TOY EXAMPLES

In addition to the 2D experiments conducted in Section 3.3 in the main paper, we investigate two more scenarios in which we (1) increase the standard deviation and (2) vary the random seed. For the former, we randomly sample data points for class A from $\mathcal{N}((0\ 1)^\top, \Sigma)$ and for class B from $\mathcal{N}((5\ 1)^\top, \Sigma)$ with $\Sigma = \begin{pmatrix} \sigma^2 & 0 \\ 0 & \sigma^2 \end{pmatrix}$ and incrementally increase $\sigma$. For the latter, we sample from the same distributions with fixed $\sigma = 1$, but use different random seeds for each run. We fit both pattern- and filter-CAVs for both experiments. As filter, we use a hard-margin SVMs. Fig. 7 presents results for additional runs for the settings discussed in the main paper, namely noise rotation (*1st row*) and feature scaling (*2nd row*), as well as the new 2D settings, including increased standard deviation (*3rd row*) and different random seeds (*4th row*). In addition to the observations discussed in the main paper, we can see that filter-CAVs from hard-margin SVMs diverge for increased values for $\sigma$, as samples are not perfectly separable anymore. Moreover, the filter-CAVs is sensitive to random seeds. In contrast, pattern-CAVs constantly point into the correct direction for all settings. Animated visualizations for all challenges discussed can be found here: https://github.com/frederikpahde/pattern-cav/tree/main/animations.

### B.5 HIGH-DIMENSIONAL TOY EXPERIMENT

We extended our 2D toy experiment to 1024-dimensional data with the concept signal in one dimension, noise rotation, and further 100 distractor signals in randomly selected dimensions. We measure the cosine similarity with the ground truth concept direction and report results in Fig. 8. The quality of pattern-CAV remains high, while SVM-CAVs are distracted by the rotated noise.

### B.6 PROOF OF DIVERGENCE: SCALING

We consider the general case of logistic regression on a set of activations $\mathcal{A} \subset \mathbb{R}^m$. For a weight vector $\mathbf{w} \in \mathbb{R}^m$ and bias term $b \in \mathbb{R}$ logistic regression models the probability of an activation $\mathbf{a}$ corresponding to concept label $t = +1$ as

$$
\widehat{\mathbb{P}}_{\mathbf{w},b}(t = +1 \mid \mathbf{a}) = \frac{e^{\mathbf{w}^\top \mathbf{a}+b}}{1 + e^{\mathbf{w}^\top \mathbf{a}+b}} = \sigma(\mathbf{w}^\top \mathbf{a} + b),
\tag{21}
$$

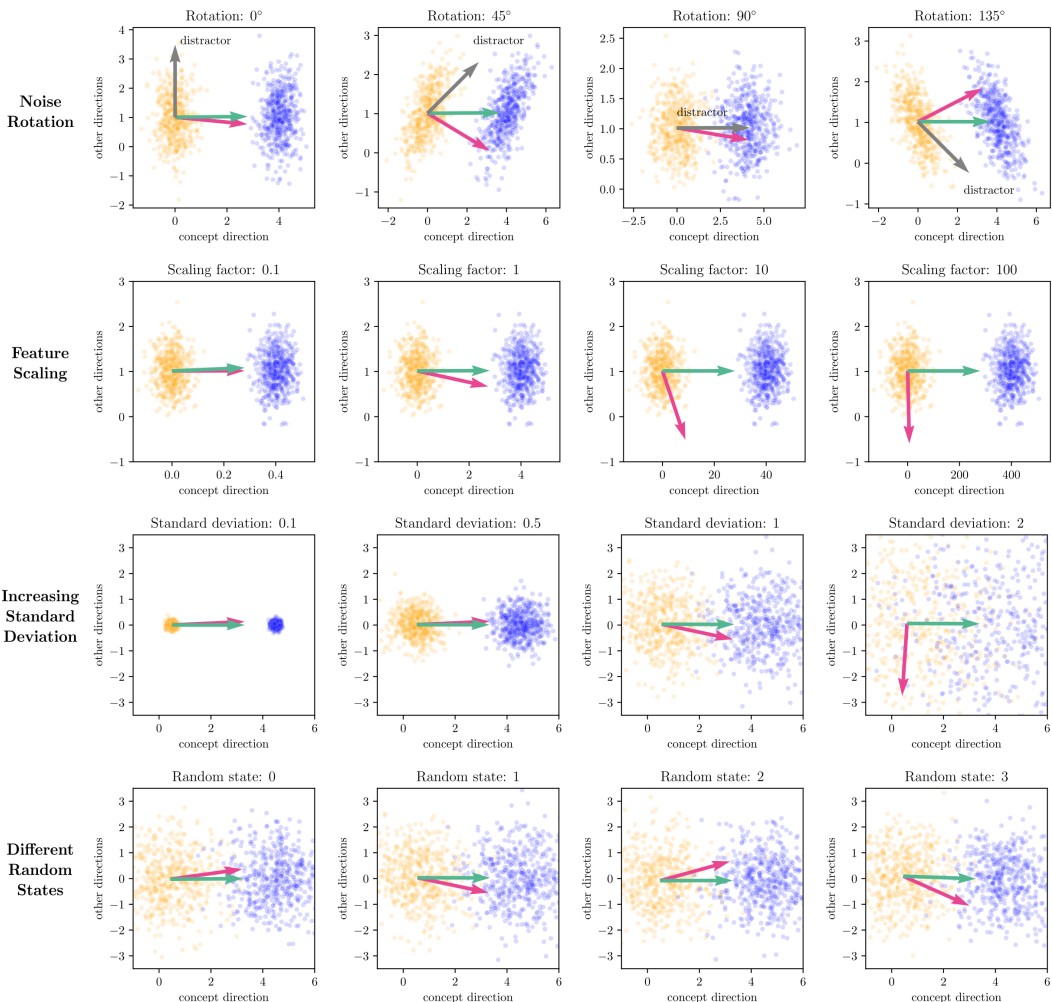

Figure 7: Multiple runs for 2D toy experiments. For noise rotation (*1ˢᵗ row*), the filter-CAV (*magenta*) diverges depending on the distractor direction, while pattern-CAV (*green*) stays constant. When increasing the scale of the x-axis (*2ⁿᵈ row*), the filter CAV scales antiproportional. For increased standard deviation (*3ʳᵈ row*), filter-CAVs (here: hard-margin SVM) diverge when the clusters are not perfectly separable anymore. Lastly, different random seeds for not perfectly separable clusters lead to varying directions for filter-based CAVs (*4ᵗʰ row*).

where $\sigma$ denotes the sigmoid function $\sigma(z) = \frac{e^z}{1+e^z}$. We predict $t = +1$ for an activation $\mathbf{a}$ if $\widehat{\mathbb{P}}_{\mathbf{w},b}(t = +1 \mid \mathbf{a}) > 0.5$ and $t = -1$ otherwise. To train an unpenalized logistic regression classifier, we seek to maximize the log likelihood of our observed activations $\mathcal{A}$

$$L(\mathbf{w}, b; \mathcal{A}) = \sum_{i \in [n]} \mathbb{1}(t_i = +1) \log \widehat{\mathbb{P}}_{\mathbf{w},b}(t = +1 \mid \mathbf{a}_i) + \mathbb{1}(t_i = -1) \log \left(1 - \widehat{\mathbb{P}}_{\mathbf{w},b}(t = +1 \mid \mathbf{a}_i)\right).$$
(22)

Assume for our unscaled set $\mathcal{A}$, we have found an optimal choice

$$\widehat{\mathbf{w}}, \hat{b} \in \operatorname*{argmax}_{\mathbf{w} \in \mathbb{R}^m, b \in \mathbb{R}} L(\mathbf{w}, b; \mathcal{A}).$$
(23)

To introduce the scaling along an axis, for a given vector $\mathbf{a} \in \mathbb{R}^n$ and a dimension $k \in [n]$ we denote by $\mathbf{a}^\gamma$ the vector which has the same entries as $\mathbf{a}$ except for the $k$-th entry, which has been replaced by $\gamma a_k$. Further, let $\mathcal{A}^\gamma = \{\mathbf{a}^\gamma \mid \mathbf{a} \in \mathcal{A}\}$ denote the set of scaled activations. Finally, for the weight

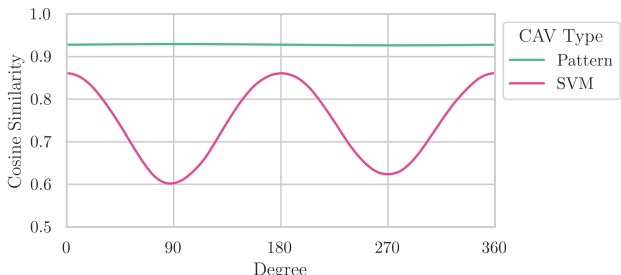

Figure 8: Cosine similarity with ground truth direction over degree of noise rotation in experiment with 1024-dimensional generated toy data with true concept direction oriented along the first dimension, noise rotation, and 100 additive distractor signals in randomly sampled directions. Pattern-CAV consistently points into the correct direction, while SVM-CAV is distracted by the rotated noise.

vector $\mathbf{w}$ we introduce the equivalent notation $\mathbf{w}^{1/\gamma}$ for the vector in which only the $k$-th entry of $\mathbf{w}$ has been changed to $\frac{1}{\gamma} w_k$. Then we derive the following equality

$$\left(\mathbf{w}^{1/\gamma}\right)^{\top} \mathbf{a}^{\gamma} + b = w_1 a_1 + \ldots + \left(\frac{1}{\gamma} w_k\right)(\gamma a_k) + \ldots + w_m a_m + b = \mathbf{w}^{\top}\mathbf{a} + b, \quad (24)$$

which implies the equalities of the predicted probabilities

$$\widehat{\mathbb{P}}_{\mathbf{w}^{1/\gamma}, b}(t = +1 \mid \mathbf{a}^{\gamma}) = \widehat{\mathbb{P}}_{\mathbf{w}, b}(t = +1 \mid \mathbf{a}) \quad (25)$$

and thus of the log likelihoods

$$L(\mathbf{w}^{1/\gamma}, b; \mathcal{A}^{\gamma}) = L(\mathbf{w}, b; \mathcal{A}). \quad (26)$$

Therefore it follows that the optimal solution to logistic regression on the scaled dataset relates to our original solution on the unscaled dataset via

$$\widehat{\mathbf{w}}^{1/\gamma}, \hat{b} = \underset{\mathbf{w} \in \mathbb{R}^m, b \in \mathbb{R}}{\operatorname{argmax}} L(\mathbf{w}, b; \mathcal{A}^{\gamma}). \quad (27)$$

In conclusion, scaling the activations by a factor of $\gamma$ in one dimension leads the signal to also scale by factor of $\gamma$ in this dimension. The filter-based CAV calculated as the weight vector of an unpenalized logistic regression, however, exhibits a scaling in the same dimension which is *antiproportional* to the scaling factor $\gamma$. Such antipropotional scaling will misalign the filter-based CAV unless it is either perfectly aligned or perfectly orthogonal to the direction of scaling. This shows that even if the filter-based CAV theoretically lies aligned or orthogonal to the scaling dimension due to noise and constraints in machine precision, logistic regression may be hugely affected by the lack of feature scaling.

### B.7    PROOF OF DIVERGENCE: NOISE ROTATION

With the additional rotational noise term and assuming the concept label $t_i$ to be fixed, our activations $\mathbf{A}_i$ are distributed according to independent multivariate normal distributions $\mathcal{N}(\mu_i, \Sigma)$ with

$$\begin{aligned} \mu_i &= \begin{pmatrix} 1 \\ 0 \end{pmatrix} \mathbb{1}(t_i = +1), \\ \Sigma &= \begin{pmatrix} \sigma^2 + \sin^2 \tau & \sin \tau \cos \tau \\ \sin \tau \cos \tau & \sigma^2 + \cos^2 \tau \end{pmatrix}. \end{aligned} \quad (28)$$

From this formulation we can see that this adds a noise which is *correlated* in the direction parallel and orthogonal to the CAVs, unless $\tau$ is a multiple of $\frac{\pi}{2}$ in which case we only add noise parallel or orthogonal to the CAVs respectively. It thus follows that the random variable $\mathbf{w}^{\top}\mathbf{A}_i + b$ has the following distribution:

$$\mathbf{w}^{\top}\mathbf{A}_i + b \overset{ind.}{\sim} \mathcal{N}\left(w_1 \mathbb{1}(t_i = +1) + b, \mathbb{V}(\mathbf{w}^{\top}\mathbf{A}_i)\right). \quad (29)$$

Note that the choice of $w_2$ does not affect the expected value of $\mathbf{w}^\top \mathbf{A}_i + b$ but may change its variance. To study this effect on the variance, for a given $\lambda \neq 0$ define the family of weight vectors

$$\mathcal{W}_\lambda = \left\{ \lambda \begin{pmatrix} 1 \\ w_2 \end{pmatrix} \,\middle|\, w_2 \in \mathbb{R} \right\}. \tag{30}$$

**Theorem B.1.** *Define the vector* $\widetilde{\mathbf{w}}_\lambda = \lambda \begin{pmatrix} 1 \\ \widetilde{w}_2 \end{pmatrix}$ *where* $\widetilde{w}_2 = -\frac{\sin \tau \cos \tau}{\sigma^2 + \cos^2 \tau}$. *Then* $\widetilde{\mathbf{w}}_\lambda$ *is the unique minimizer*

$$\widetilde{\mathbf{w}}_\lambda = \underset{\mathbf{w} \in \mathcal{W}_\lambda}{\operatorname{argmin}} \, \mathbb{V}(\mathbf{w}^\top \mathbf{A}_i). \tag{31}$$

*Proof.* The variance for $\mathbf{w} \in \mathcal{W}_\lambda$ is given by

$$\begin{aligned}
\mathbb{V}(\mathbf{w}^\top \mathbf{A}_i) &= \lambda^2 \begin{pmatrix} 1 & w_2 \end{pmatrix} \begin{pmatrix} \sigma^2 + \sin^2 \tau & \sin \tau \cos \tau \\ \sin \tau \cos \tau & \sigma^2 + \cos^2 \tau \end{pmatrix} \begin{pmatrix} 1 \\ w_2 \end{pmatrix} \\
&= \lambda^2 \left\{ \sigma^2 + \sin^2 \tau + 2 w_2 \sin \tau \cos \tau + w_2^2 \sigma^2 + w_2^2 \cos^2 \tau \right\} \\
&= \lambda^2 \left\{ \sigma^2 + w_2^2 \sigma^2 + (\sin \tau + w_2 \cos \tau)^2 \right\}.
\end{aligned} \tag{32}$$

Differentiating with respect to $w_2$ gives the following expression

$$\frac{\partial}{\partial w_2} \mathbb{V}(\mathbf{w}^\top \mathbf{A}_i) = \lambda^2 \left\{ 2 w_2 \sigma^2 + 2 \cos \tau (\sin \tau + w_2 \cos \tau) \right\} = 2\lambda^2 \left\{ (\sigma^2 + \cos^2 \tau) w_2 + \sin \tau \cos \tau \right\} \tag{33}$$

This is set to zero if and only if $\widetilde{w}_2 = -\frac{\sin \tau \cos \tau}{\sigma^2 + \cos^2 \tau}$. Furthermore, the second derivative is

$$\frac{\partial^2}{\partial w_2^2} \mathbb{V}(\mathbf{w}^\top \mathbf{A}_i) = 2\lambda^2 \left( \sigma^2 + \cos^2 \tau \right) > 0, \tag{34}$$

so $\widetilde{w}_2$ indeed minimizes the variance. $\qquad\square$

The proofs of divergence for both logistic regression and SVMs are now analogous: Assuming there are two vectors $\mathbf{w}, \widetilde{\mathbf{w}}$ for which $\mathbf{w}^\top \mathbf{A}_i + b$ has the same expected value but $\widetilde{\mathbf{w}}$ yields a smaller variance, then the expected value of the objective function of the optimization problem of the model (the log likelihood for logistic regression or the size of the margin for SVMs respectively) will be larger for $\widetilde{\mathbf{w}}$. Together with Theorem B.1, this proves that a vector of the form $\widetilde{\mathbf{w}}_\lambda$ maximizes the expected value of the objective function and is thus preferred as the weight vector over the true CAV $\begin{pmatrix} 1 & 0 \end{pmatrix}^\top$ with non-zero probability.

### B.7.1 LOGISTIC REGRESSION

For logistic regression, we intend to maximize the log likelihood of our observed data. We may express our log likelihood in terms of the random variables $\mathbf{w}^\top \mathbf{A}_i + b$ by the formula

$$\begin{aligned}
L(\mathbf{w}, b; \mathcal{A}) &= \sum_{i \in [N]} \mathbb{1}(t_i = +1) \log(\sigma(\mathbf{w}^\top \mathbf{A}_i + b)) + \mathbb{1}(t_i = -1) \log(1 - \sigma(\mathbf{w}^\top \mathbf{A}_i + b)) \\
&= \sum_{i \in [N]} \mathbb{1}(t_i = +1) \log(\sigma(\mathbf{w}^\top \mathbf{A}_i + b)) + \mathbb{1}(t_i = -1) \log(\sigma(-\mathbf{w}^\top \mathbf{A}_i - b)),
\end{aligned} \tag{35}$$

where $\sigma$ denotes the sigmoid function.

**Theorem B.2.** *Let* $\mathbf{w}, \widetilde{\mathbf{w}}$ *be two weight vectors with* $\mathbb{E}\left[\mathbf{w}^\top \mathbf{A}_i + b\right] = \mathbb{E}\left[\widetilde{\mathbf{w}}^\top \mathbf{A}_i + b\right]$ *for all* $i \in [n]$ *and* $\mathbb{V}(\mathbf{w}^\top \mathbf{A}_i + b) > \mathbb{V}(\widetilde{\mathbf{w}}^\top \mathbf{A}_i + b)$. *Then*

$$\mathbb{E}\left[L(\widetilde{\mathbf{w}}, b; \mathcal{A})\right] > \mathbb{E}\left[L(\mathbf{w}, b; \mathcal{A})\right]. \tag{36}$$

*Proof.* To focus on the effect of the variance on the log likelihood, we define independently distributed random variables $Y_i \sim \mathcal{N}(\mu_i, \varsigma^2)$ with means $\mu_i$ and shared variance $\varsigma^2 > 0$. We allow the $Y_i$ to have different means as the mean of the random variables $\mathbf{w}^\top \mathbf{A}_i + b$ also differ depending on the concept label $t_i$. We may now define the functions $f$ and $g_i$ which both depend on $\varsigma^2$ via

$$g_i(Y_i) = \mathbb{1}(t_i = +1) \log(\sigma(Y_i)) + \mathbb{1}(t_i = -1) \log(\sigma(-Y_i)) \tag{37}$$

and

$$f(\varsigma^2) = \mathbb{E}\left[\sum_{i \in [n]} g_i(Y_i)\right] = \sum_{i \in [n]} \mathbb{E}\left[g_i(Y_i)\right], \tag{38}$$

where we used the linearity of the expected value in the last step. After proving that the function $f$ is strictly decreasing the claim follows from inserting $\mathbf{w}^\top \mathbf{A}_i + b$ for $Y_i$. We prove first that $\log(\sigma(z))$ is a strictly concave function by calculating the second derivative:

$$(\log(\sigma(z)))'' = \left(\frac{1}{\sigma(z)}\sigma(z)(1-\sigma(z))\right)' = (1-\sigma(z))' = -\sigma(z)(1-\sigma(z)) < 0 \text{ for } z \in \mathbb{R}, \tag{39}$$

which holds as the image of the sigmoid function is the open interval $(0, 1)$. Because the logarithm of the sigmoid is a strictly concave function, so is $\log(\sigma(-z))$, hence each summand

$$g_i(Y_i) = \mathbb{1}(t_i = +1)\log(\sigma(Y_i)) + \mathbb{1}(t_i = -1)\log(\sigma(-Y_i)) \tag{40}$$

is strictly concave in $Y_i$.

Now consider two variances $\varsigma_1^2 < \varsigma_2^2$ and define independent random variables $Y_i^1 \sim \mathcal{N}(\mu_i, \varsigma_1^2)$ and $Y_i' \sim \mathcal{N}(0, \varsigma_2^2 - \varsigma_1^2)$ such that their sum are independently distributed random variables $Y_i^2 := Y_i^1 + Y_i' \sim \mathcal{N}(\mu_i, \varsigma_2^2)$. Using the conditional version of Jensen's inequality on the strictly concave functions $g_i$, we derive

$$\begin{aligned}
\mathbb{E}\left[g_i(Y_i^2)\right] &= \mathbb{E}\left[g_i(Y_i^1 + Y_i')\right] = \mathbb{E}\left[\mathbb{E}\left[g_i(Y_i^1 + Y_i') \mid Y_i^1\right]\right] \\
&< \mathbb{E}\left[g_i\left(\mathbb{E}\left[Y_i^1 + Y_i' \mid Y_i^1\right]\right)\right] = \mathbb{E}\left[g_i\left(Y_i^1 + \mathbb{E}\left[Y_i'\right]\right)\right] = \mathbb{E}\left[g_i\left(Y_i^1\right)\right],
\end{aligned} \tag{41}$$

where the second-to-last step follows from the properties of conditional expectation for completely dependent and independent random variables. Summing over all $i$ finally proves the desired inequality

$$f(\varsigma_1^2) = \sum_{i \in [n]} \mathbb{E}\left[g_i\left(Y_i^1\right)\right] > \sum_{i \in [n]} \mathbb{E}\left[g_i\left(Y_i^2\right)\right] = f(\varsigma_2^2). \tag{42}$$

$\square$

### B.7.2 SVMs

We inspect the behavior of a linear hard-margin SVM, assuming that our data can be perfectly separated by a linear hyperplane. Then the optimization problem for this particular SVM is given by

$$\max_{\mathbf{w}, b} \frac{2}{\|\mathbf{w}\|_2} \tag{43}$$
$$\text{subject to } t_i(\mathbf{w}^\top \mathbf{A}_i + b) \geq 1 \text{ for all } i \in [n].$$

This states that we aim to maximize the margin which has length $2/\|\mathbf{w}\|_2$ subject to every datapoint lying on the correct side of the margin. A fitted SVM will have at least one vector of each class, the so-called support vectors, on its margin, which can be equally formulated as $\min_{t_i=+1}\left(\mathbf{w}^\top \mathbf{A}_i + b\right) = 1$ and $\max_{t_i=-1}\left(\mathbf{w}^\top \mathbf{A}_i + b\right) = -1$. We may use these quantities to reformulate the length of the margin as

$$\frac{1}{\|\mathbf{w}\|_2}\left\{\min_{t_i=+1}\left(\mathbf{w}^\top \mathbf{A}_i + b\right) - \max_{t_i=-1}\left(\mathbf{w}^\top \mathbf{A}_i + b\right)\right\}, \tag{44}$$

which is what we are trying to maximize in order to find the direction of our weight vector $\mathbf{w}$.

**Theorem B.3.** *Let $\mathbf{w}, \widetilde{\mathbf{w}}$ be two weight vectors with $\mathbb{E}\left[\mathbf{w}^\top \mathbf{A}_i + b\right] = \mathbb{E}\left[\widetilde{\mathbf{w}}^\top \mathbf{A}_i + b\right]$ for all $i \in [n]$, $\mathbb{V}(\mathbf{w}^\top \mathbf{A}_i + b) > \mathbb{V}(\widetilde{\mathbf{w}}^\top \mathbf{A}_i + b)$ and $\|\mathbf{w}\|_2 \leq \|\widetilde{\mathbf{w}}\|_2$. Then for sufficiently large sample size $n$ the expected margin size for the SVM with normal vector in direction of $\widetilde{\mathbf{w}}$ is bigger than for the SVM with normal vector in direction of $\mathbf{w}$.*

*Proof.* We denote $\mu_+ = \mathbb{E}\left[\mathbf{w}^\top \mathbf{A}_i + b \mid t_i = +1\right]$, $\mu_- = \mathbb{E}\left[\mathbf{w}^\top \mathbf{A}_i + b \mid t_i = -1\right]$ and $\varsigma^2 = \mathbb{V}\left(\mathbf{w}^\top \mathbf{A}_i + b\right)$, and further define random variables $N_i, N_i'$ as independent standard normal distributions. We can now write the expected size of the margin as

$$\mathbb{E}\left[\frac{1}{\|\mathbf{w}\|_2}\left\{\min_{t_i=+1}\left(\mathbf{w}^\top \mathbf{A}_i + b\right) - \max_{t_i=-1}\left(\mathbf{w}^\top \mathbf{A}_i + b\right)\right\}\right]$$

$$= \frac{1}{\|\mathbf{w}\|_2}\left\{\mu_+ - \mu_- + \varsigma\mathbb{E}\left[\min_{i\in[n/2]} N_i - \max_{i\in[n/2]} N_i'\right]\right\}$$

$$= \frac{1}{\|\mathbf{w}\|_2}\left\{\mu_+ - \mu_- - \varsigma\mathbb{E}\left[\max_{i\in[n/2]} N_i + \max_{i\in[n/2]} N_i'\right]\right\} \qquad (45)$$

$$= \frac{1}{\|\mathbf{w}\|_2}\left\{\mu_+ - \mu_- - 2\varsigma\mathbb{E}\left[\max_{i\in[n/2]} N_i\right]\right\}$$

$$= \frac{1}{\|\mathbf{w}\|_2}\left\{\mu_+ - \mu_- - 2\varsigma m(n)\right\},$$

where we define $m(n) := \mathbb{E}\left[\max_{i\in[n/2]} N_i\right]$ and the step from the second to third line follows by the symmetry of the standard normal distribution and the fact that $\min(S) = -\max(S)$ for symmetric sets $S$. Firstly, we show that the quantity $m(n)$ grows unbounded. Let $M > \mathbb{E}\left[\max\left(N_1, 0\right)\right] = \frac{1}{\sqrt{2\pi}}$. Then

$$m(n) = \mathbb{E}\left[\max\left(\max_{i\in[n/2]} N_i, 0\right)\right] + \mathbb{E}\left[\min\left(\max_{i\in[n/2]} N_i, 0\right)\right]$$

$$\geq 4M \cdot \mathbb{P}(m(N) \geq 4M) - \mathbb{E}\left[\max\left(N_1, 0\right)\right]$$

$$\geq 4M \cdot \left\{1 - \mathbb{P}(N_i < 4M \text{ for all } i)\right\} - M \qquad (46)$$

$$= 4M \cdot \left\{1 - \mathbb{P}(N_1 < 4M)^{[n/2]}\right\} - M$$

$$\geq 4M \cdot \left\{1 - \frac{1}{2}\right\} - M = 2M - M = M,$$

where the last inequality holds for sufficiently large $n$. As $M$ may be chosen arbitrarily large, this proves that $m(n)$ is unbounded.

Now for two weight vectors $\mathbf{w}, \widetilde{\mathbf{w}}$ with the same associated expected values $\mu_+, \mu_-$, variances $\varsigma^2 = \mathbb{V}\left(\mathbf{w}^\top \mathbf{A}_i + b\right) > \mathbb{V}\left(\widetilde{\mathbf{w}}^\top \mathbf{A}_i + b\right) = \widetilde{\varsigma}^2$ and $\|\mathbf{w}\|_2 \leq \|\widetilde{\mathbf{w}}\|_2$ it follows by simple arithmetic that the inequality

$$\frac{1}{\|\widetilde{\mathbf{w}}\|_2}\left\{\mu_+ - \mu_- - 2\widetilde{\varsigma}m(n)\right\} > \frac{1}{\|\mathbf{w}\|_2}\left\{\mu_+ - \mu_- - 2\varsigma m(n)\right\} \qquad (47)$$

is equivalent to

$$2\left(\frac{\varsigma}{\|\mathbf{w}\|_2} - \frac{\widetilde{\varsigma}}{\|\widetilde{\mathbf{w}}\|_2}\right)m(n) > \left(\frac{1}{\|\mathbf{w}\|_2} - \frac{1}{\|\widetilde{\mathbf{w}}\|_2}\right)\left\{\mu_+ - \mu_-\right\}. \qquad (48)$$

Note, that since $\varsigma > \widetilde{\varsigma}$ and $\|\mathbf{w}\|_2 < \|\widetilde{\mathbf{w}}\|_2$, it follows that

$$\frac{\varsigma}{\|\mathbf{w}\|_2} - \frac{\widetilde{\varsigma}}{\|\widetilde{\mathbf{w}}\|_2} > 0. \qquad (49)$$

So the left side of the inequality grows unbounded with $n$ while the right side remains constant. Hence, for $n$ sufficiently large, the inequality is fulfilled and the expected margin of the SVM associated with $\widetilde{\mathbf{w}}$ is greater than the expected margin for the vector $\mathbf{w}$. $\qquad\square$

## C EXPERIMENT DETAILS

We provide dataset details in Section C.1 and training details in Section C.2. The former includes details for controlled "Clever Hans" datasets (Section C.1.1) and the synthetic FunnyBirds dataset (Section C.1.2).

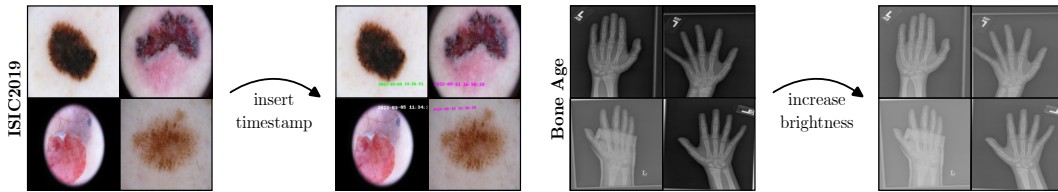

Figure 9: Examples from controlled datasets with clean and attacked samples for ISIC2019 (*left*) and Bone Age (*right*), with timestamp and brightness artifacts, respectively.

## C.1    DATASETS

### C.1.1    CONTROLLED "CLEVER HANS" DATASETS

Details for our controlled datasets with artificial "Clever Hans" artifacts, *i.e.*, shortcut features, are provided in Tab. 2. Examples are shown in Fig. 9.

Table 2: Details for our controlled "Clever Hans" datasets, including artifact type, number of samples, class names, the biased class, percentage of samples with artifact in the biased class ($p$-bias), and train/val/test split.

| dataset | artifact | number samples | classes | biased class | $p$-bias | train / val / test split |
|---------|----------|----------------|---------|--------------|----------|--------------------------|
| Bone Age | brightness | 12,611 | 0-46, 47-91, 92-137, 138-182, 183-228 (months) | 92-137 | 20% | 80%/10%/10% |
| ISIC2019 | timestamp | 25,331 | MEL, NV, BCC, AK, BKL, DF, VASC, SCC | MEL | 1% | 80%/10%/10% |

### C.1.2    FUNNYBIRDS DATASET

FunnyBirds (Hesse et al., 2023) provides a framework to synthesize images of different classes of birds. Specifically, a bird is defined using 5 parts, for which the authors manually designed different types (4 beaks, 3 eyes, 4 feet, 9 tails, 6 wings). Further varying color, this leads to 2592 possible combinations, *i.e.*, classes. We define a concept as a combination of part, type and color. For example, the concept "beak::beak-01::yellow" entails the beak shape `beak-01` in color `yellow`. As outlined in Section 4.1 in the main paper, we construct a new version of FunnyBirds with 10 classes, with *exactly* one valid feature, *i.e.* concept, per class. While the class-defining concept is identical for all samples per class, all other concepts are chosen randomly per sample. The class-defining concepts are listed in Tab. 3. When training models on this dataset, the class-defining property *must* be used by the model. We synthesize 500 training samples and 100 test samples per class, totaling to 5000 training and 1000 test samples. The training set is further split into training/validation splits (90%/10%). In order to remove concepts, *e.g.*, for the computation of sample-wise ground truth concept directions, we replace the class-defining property with another randomly chosen concept (*e.g.*, "beak::beak-01::yellow" → "beak::beak-03::yellow"), while keeping other parts unchanged. Examples for original and manipulated samples are shown in Fig. 10.

## C.2    TRAINING DETAILS

Tab. 4 provides training details for all models and datasets, including the source of the pre-trained model checkpoint, optimizer, learning rate (LR), number of epochs, and milestones, after which we divide the LR by 10. All models are pre-trained on ImageNet (Deng et al., 2009; Ridnik et al., 2021) with weights provided from `timm` (Wightman, 2019) or `torchvision` (maintainers & contributors, 2016).

Table 3: Class-defining concepts (part/shape/color) for all 10 classes in our synthetic FunnyBirds dataset.

|       | class-defining concept | | |
|-------|------|---------|--------|
| class | part | shape | color |
| 1 | beak | beak-01 | yellow |
| 2 | beak | beak-02 | yellow |
| 3 | beak | beak-03 | yellow |
| 4 | beak | beak-04 | yellow |
| 5 | wing | wing-01 | red |
| 6 | wing | wing-02 | red |
| 7 | wing | wing-01 | green |
| 8 | wing | wing-02 | green |
| 9 | wing | wing-01 | blue |
| 10 | wing | wing-02 | blue |

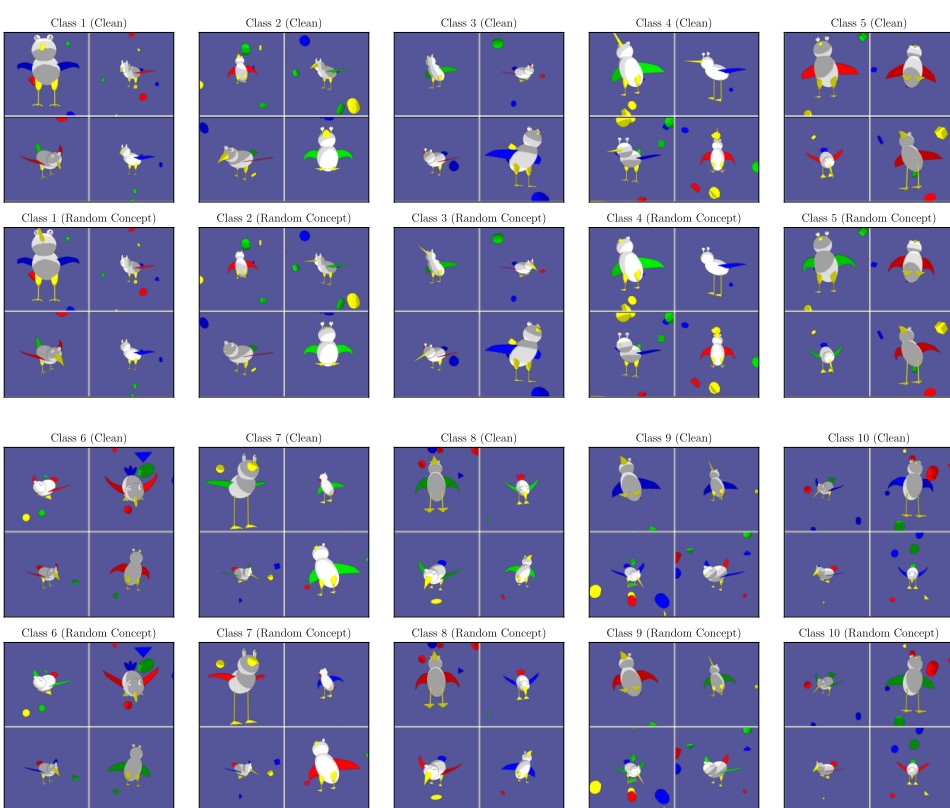

Figure 10: Examples for samples from all 10 classes from our synthetic FunnyBirds dataset, including clean samples (*top*) and identical samples with class-defining concept randomized (*bottom*).

## C.3 COMPUTATIONAL RESOURCES

We ran all model training and correction jobs on GPUs of type NVIDIA Ampere A100 with 40 GB RAM. Depending on the architecture and correction layer, a model correction job including evaluation took between 20 minutes and 2 hours. Depending on the architecture, model training took 6-12h for ISIC2019, 1-4h for Bone Age, and 8-30mins for FunnyBirds.

## C.4 LICENSES FOR EXISTING ASSETS

Existing assets used in this paper have the following licenses and terms of uses:

Table 4: Model training details including the pre-trained checkpoint, optimizer, learning Rate (LR), number of epochs, and milestones, after which the learning rate is divided by 10.

| dataset | model | pre-trained checkpoint | optimizer | LR | epochs (milestones) |
|---|---|---|---|---|---|
| Bone Age | VGG16 | `torchvision/IMAGENET1K_V1` | SGD | 0.005 | 100 (50,80) |
| | ResNet18 | `timm/resnet18.a1_in1k` | Adam | 0.005 | 100 (50,80) |
| | ResNet50 | `timm/resnet50.a1_in1k` | Adam | 0.005 | 100 (50,80) |
| | ResNeXt50 | `timm/resnext50_32x4d.a1h_in1k` | Adam | 0.001 | 100 (50,80) |
| | ReXNet100 | `timm/rexnet_100.nav_in1k` | Adam | 0.005 | 100 (50,80) |
| | EfficientNet-B0 | `torchvision/IMAGENET1K_V1` | Adam | 0.001 | 100 (50,80) |
| | EfficientNet-V2-s | `torchvision/IMAGENET1K_V1` | Adam | 0.001 | 100 (50,80) |
| | Vision Transformer | `timm/vit_base_patch16_224.augreg_in21k` | SGD | 0.0005 | 100 (50,80) |
| | Swin Transformer | `timm/swin_base_patch4_window7_224.ms_in22k` | SGD | 0.0005 | 100 (50,80) |
| ISIC2019 (controlled) | VGG16 | `torchvision/IMAGENET1K_V1` | SGD | 0.005 | 300 (150,250) |
| | ResNet18 | `timm/resnet18.a1_in1k` | Adam | 0.0005 | 300 (150,250) |
| | ResNet50 | `timm/resnet50.a1_in1k` | Adam | 0.0005 | 300 (150,250) |
| | ResNeXt50 | `timm/resnext50_32x4d.a1h_in1k` | Adam | 0.0005 | 300 (150,250) |
| | ReXNet100 | `timm/rexnet_100.nav_in1k` | Adam | 0.0005 | 300 (150,250) |
| | EfficientNet-B0 | `torchvision/IMAGENET1K_V1` | Adam | 0.0005 | 300 (150,250) |
| | EfficientNet-V2-s | `torchvision/IMAGENET1K_V1` | Adam | 0.0005 | 300 (150,250) |
| | Vision Transformer | `google/vit_base_patch16_224` | SGD | 0.001 | 300 (150,250) |
| | Swin Transformer | `timm/swin_base_patch4_window7_224.ms_in22k` | SGD | 0.001 | 300 (150,250) |
| ISIC2019 (real) | VGG16 | `torchvision/IMAGENET1K_V1` | SGD | 0.005 | 150 (80,120) |
| | ResNet18 | `timm/resnet18.a1_in1k` | Adam | 0.0005 | 300 (150,250) |
| | ResNet50 | `timm/resnet50.a1_in1k` | Adam | 0.0005 | 300 (150,250) |
| | ResNeXt50 | `timm/resnext50_32x4d.a1h_in1k` | Adam | 0.0005 | 300 (150,250) |
| | ReXNet100 | `timm/rexnet_100.nav_in1k` | Adam | 0.0005 | 300 (150,250) |
| | EfficientNet-B0 | `torchvision/IMAGENET1K_V1` | Adam | 0.0005 | 300 (150,250) |
| | EfficientNet-V2-s | `torchvision/IMAGENET1K_V1` | SGD | 0.001 | 300 (150,250) |
| | Vision Transformer | `google/vit_base_patch16_224` | SGD | 0.0005 | 300 (150,250) |
| Funny Birds | VGG16 | `torchvision/IMAGENET1K_V1` | SGD | 0.005 | 50 (30) |
| | ResNet18 | `timm/resnet18.a1_in1k` | Adam | 0.005 | 50 (30) |
| | ResNeXt50 | `timm/resnext50_32x4d.a1h_in1k` | Adam | 0.001 | 50 (30) |
| | ReXNet100 | `timm/rexnet_100.nav_in1k` | Adam | 0.005 | 50 (30) |
| | EfficientNet-B0 | `torchvision/IMAGENET1K_V1` | Adam | 0.001 | 50 (30) |
| | EfficientNet-V2-s | `torchvision/IMAGENET1K_V1` | Adam | 0.001 | 50 (30) |
| | Vision Transformer | `google/vit_base_patch16_224` | SGD | 0.005 | 50 (30) |

- **ISIC2019:** CC-BY-NC
- **Pediatric Bone Age:** The terms of use are described here: *https://www.rsna.org/-/media/Files/RSNA/Education/AI-resources-and-training/AI-image-challenge/RSNA-2017-AI-Challenge-Terms-of-Use-and-Attribution_Final.ashx?la=en&hash=F28B401E267D05658C85F5D207EC4F9AE9AE6FA9*
- **FunnyBirds:** Apache License 2.0
- **timm model checkpoints:** Apache License
- **torchvision checkpoints:** BSD 3-Clause License

# D  ADDITIONAL EXPERIMENTAL RESULTS

## D.1  DETAILED CAV ALIGNMENT RESULTS

Additional CAV alignment results, including filter-(Lasso, Logistic, Ridge, and SVM) and pattern-CAVs are shown in Figs 11, 12, 13, and 14 for ISIC2019, Figs 15, 16, 17, and 18 for Pediatric Bone Age, and Figs 19, 20, and 21 for FunnyBirds, for VGG16, ResNet18, ResNet50, ResNeXt50, ReXNet100, EfficientNet-B0, and EfficientNetV2 models. The results confirm the trends described in the main paper in Section 4.2, *i.e.*, a higher alignment with the ground truth concept direction for pattern-CAVs and a better concept separability for filter-CAVs.

Moreover, we report the cosine similarities between CAVs obtained with different feature preprocessing methods (centering, max-scaling, and their combination) and the ground truth concept direction for ISIC2019 and Bone Age datasets on the last Conv layers of ResNet18, ResNet50, ResNeXt50, ReXNet100, EfficientNet-B0, EfficientNetV2, Vision Transformer and Swin Transformer in Figs. 22, 23, 24, and 25.

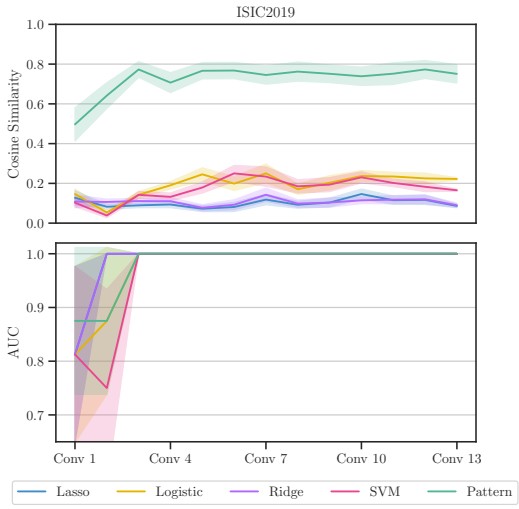

Figure 11: Comparison of cosine similarity between CAVs and true concept direction (*top*) and concept separability as AUC (*bottom*), using filter- (lasso, logistic, ridge, and SVM) and pattern-CAV, and for all Conv layers of VGG16 trained on ISIC2019.

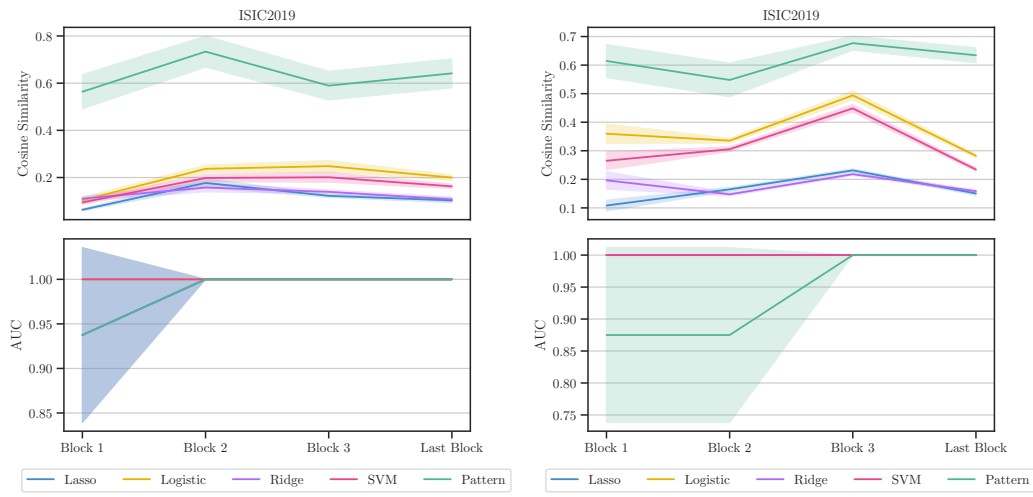

Figure 12: Comparison of cosine similarity between CAVs and true concept direction (*top*) and concept separability as AUC (*bottom*), using filter- (lasso, logistic, ridge, and SVM) and pattern-CAV, and after each block of ResNet18 (*left*) and ResNet50 (*left*) trained on ISIC2019.

## D.2 QUALITATIVE CAV RESULTS

Following-up on the qualitative approach on Section 4.2, we present further RelMax visualizations for the most important neurons for different CAVs in Fig 27. In contrast to Fig. 2 in the main paper, we include all our CAV approaches, namely 4 filter-based (lasso, logistic, ridge, and SVM) and the pattern-based CAV. Again, all filter-CAVs include unrelated neurons, whereas the pattern-CAV mainly includes neurons focusing on the concept of interest.

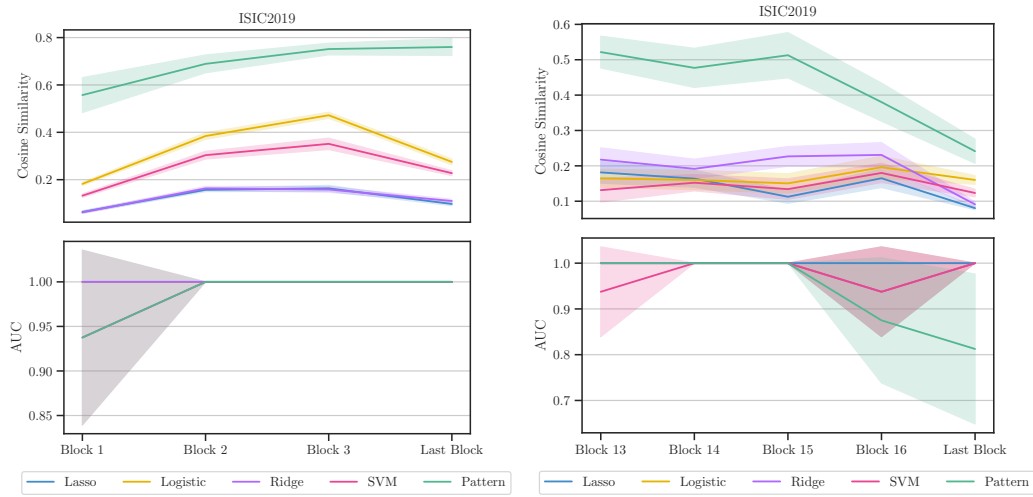

Figure 13: Comparison of cosine similarity between CAVs and true concept direction (*top*) and concept separability as AUC (*bottom*), using filter- (lasso, logistic, ridge, and SVM) and pattern-CAV, and after each block of ResNeXt50 (*left*) and ReXNet100 (*left*) trained on ISIC2019.

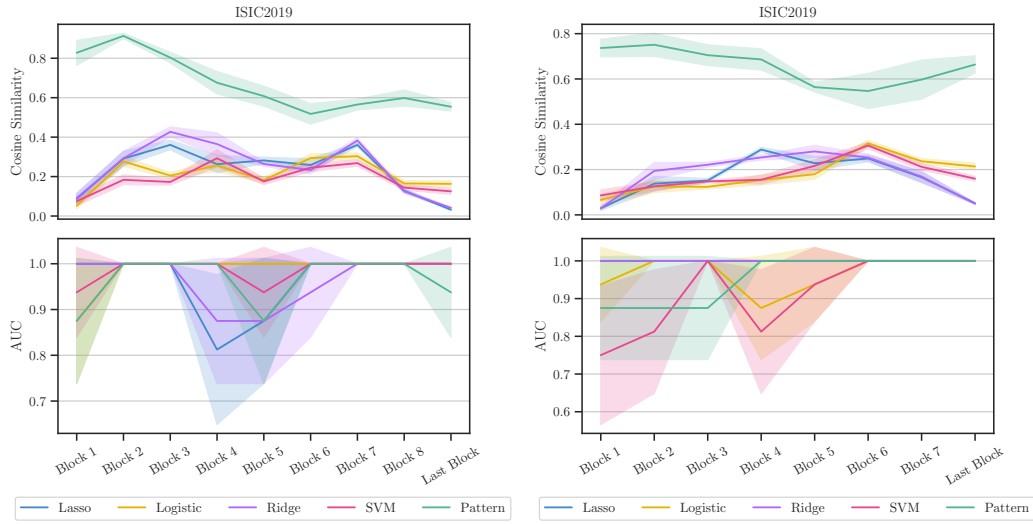

Figure 14: Comparison of cosine similarity between CAVs and true concept direction (*top*) and concept separability as AUC (*bottom*), using filter- (lasso, logistic, ridge, and SVM) and pattern-CAV, and after each block of EfficientNet-B0 (*left*) and EfficientNetV2 (*left*) trained on ISIC2019.

## D.3 REDUCTION OF SUPERVISION

In a further set of experiments, we want to analyze the possibility to reduce the manual labeling efforts by (1) the unsupervised discovery of concept directions in Sec. D.3.1 and (2) the robustness of (supervised) CAV directions towards labeling errors and reduction of data size in Sec. D.3.2.

### D.3.1 ALIGNMENT OF UNSUPERVISED CAV DIRECTIONS

We do an unsupervised concept discovery in the penultimate layer of VGG16 trained on ISIC2019 (timestamp artifact) with CRAFT (Fel et al., 2023) (via Non-negative Matrix Factorization) and compute the cosine similarity of each found concept direction with the ground truth direction and plot a histogram of similarity scores in Fig. 28. It can be seen that the best CRAFT direction outperforms

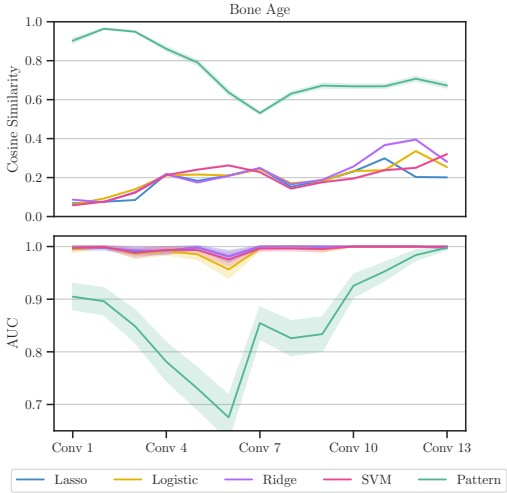

Figure 15: Comparison of cosine similarity between CAVs and true concept direction (*top*) and concept separability as AUC (*bottom*), using filter- (lasso, logistic, ridge, and SVM) and pattern-CAV, and after each Conv layer of VGG16 trained on the Pediatric Bone Age dataset.

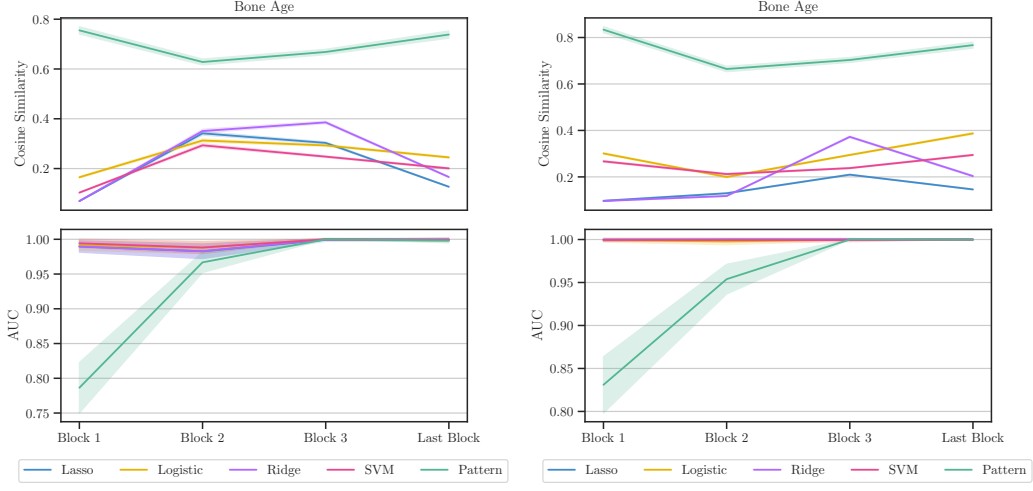

Figure 16: Comparison of cosine similarity between CAVs and true concept direction (*top*) and concept separability as AUC (*bottom*), using filter- (lasso, logistic, ridge, and SVM) and pattern-CAV, and after each block of ResNet18 (*left*) and ResNet50 (*right*) trained on the Pediatric Bone Age dataset.

the SVM CAV, however, it is worse than pattern-CAV. Moreover, it is to note that unsupervised concept discovery comes with two drawbacks in practice: (1) It requires manual inspection of found concepts to decide which direction(s) represent the desired concept. (2) Matrix factorization will find statistical groupings without guidance, hence there is no guarantee that one direction will represent the desired concept.

### D.3.2 ROBUSTNESS TOWARDS MISSING DATA AND LABELING ERRORS

As pattern-CAVs are more robust against noise in activations, they are more stable for low-data or mislabeled samples compared to filter-based CAVs. We verified this in additional experiments was activations from the penultimate layer of VGG16 trained on ISIC2019 (timestamp artifact) with results shown in Fig 29: (1) We gradually decreased the number of known artifact samples before CAV computation and found that pattern-CAV remains more precise than filter-based CAVs with

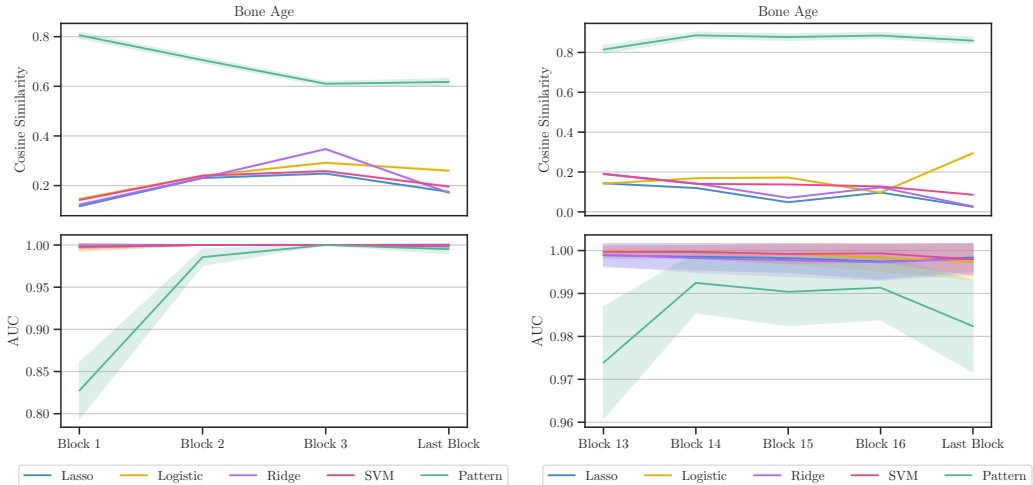

Figure 17: Comparison of cosine similarity between CAVs and true concept direction (*top*) and concept separability as AUC (*bottom*), using filter- (lasso, logistic, ridge, and SVM) and pattern-CAV, and after each block of ResNeXt50 (*left*) and ReXNet100 (*right*) trained on the Pediatric Bone Age dataset.

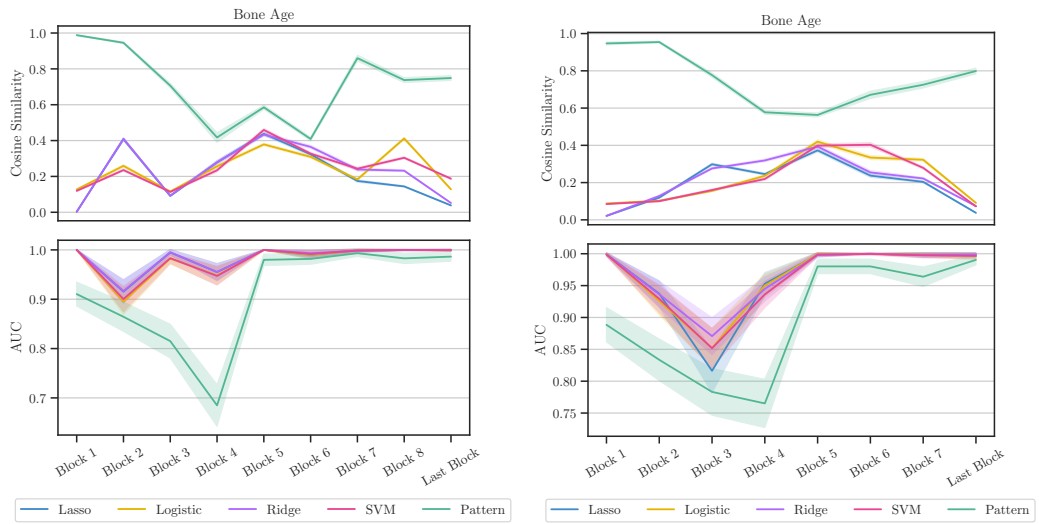

Figure 18: Comparison of cosine similarity between CAVs and true concept direction (*top*) and concept separability as AUC (*bottom*), using filter- (lasso, logistic, ridge, and SVM) and pattern-CAV, and after each block of EfficentNet-B0 (*left*) and EfficientNetV2 (*right*) trained on the Pediatric Bone Age dataset.

reduced data (*left*). (2) We gradually increased artifact mislabeling rate $p$ (false positive rate) and found that the quality of filter-CAVs decrease rapidly, while the quality of pattern-CAVs consistently remains high (*right*).

## D.4    QUALITATIVE TCAV RESULTS

Extending on our qualitative TCAV results from Section 4.3.1, we show further sensitivity heatmaps for all considered CAV types, including four filter- (lasso, logistic, ridge, SVM) and our pattern-CAV in Fig. 31. We observe similar trends as in the main paper. Specifically, filter-CAVs lead to noisy

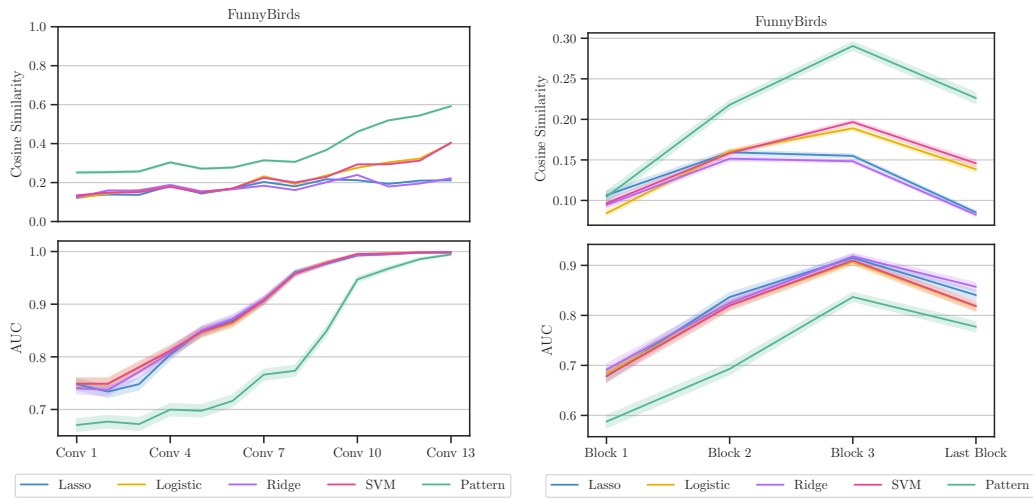

Figure 19: Comparison of cosine similarity between CAVs and true concept direction (*top*) and concept separability as AUC (*bottom*), using filter- (lasso, logistic, ridge, and SVM) and pattern-CAV, and after each Conv layer of VGG16 (*left*) and ResNet18 (*right*) trained on FunnyBirds.

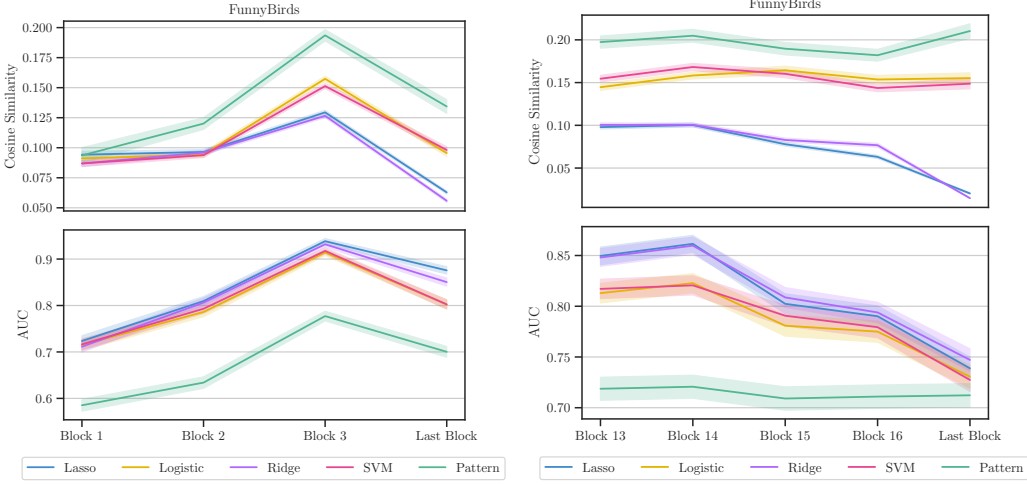

Figure 20: Comparison of cosine similarity between CAVs and true concept direction (*top*) and concept separability as AUC (*bottom*), using filter- (lasso, logistic, ridge, and SVM) and pattern-CAV, and after each block of ResNeXt50 (*left*) and ReXNet100 (*right*) trained on FunnyBirds.

sensitivity heatmaps, negatively impacting the TCAV score, while pattern-CAV precisely localizes the concept with positive sensitivity.

Note, that for ResNet18, instead of precisely localizing concepts, the sensitivity in the last Conv layer spreads over the entire sample ($7 \times 7$ pixels), as shown in Fig. 26. Therefore, TCAV scores for ResNet18 are less impacted by noisy concept sensitivity maps in irrelevant regions. Similar trends have been observed for ResNeXt50 and ReXNet100 models.

## D.5 QUANTITATIVE TCAV RESULTS

In addition to the results for the controlled TCAV experiments with FunnyBirds shown in Fig. 5 in Sec. 4.3.1, we present results for additional model architectures in Fig. 31, including ResNeXt50, ReXNet100, EfficientNetV2, and Vision Transformer. Interestingly, ResNet18, ResNeXt50 and ReXNet100 all share similar behavior, which is further discussed in the main paper. Note that due to

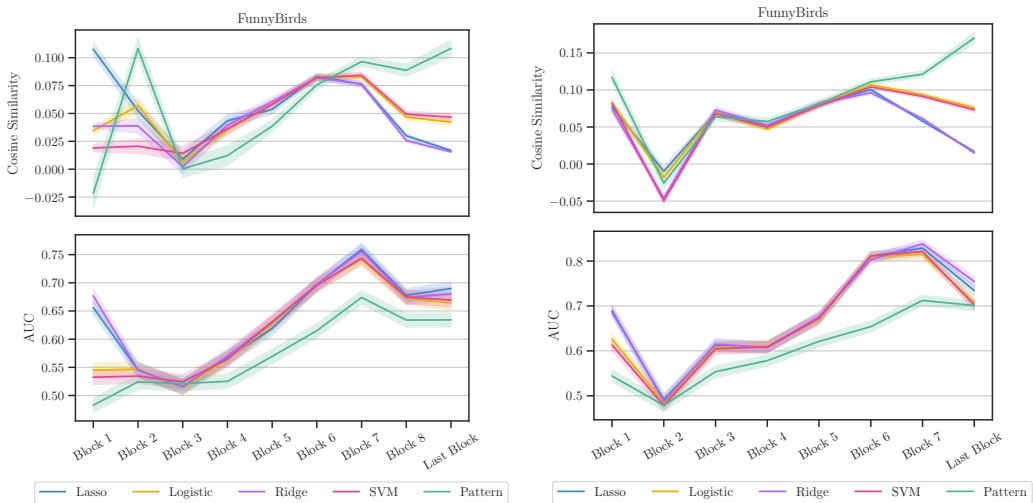

Figure 21: Comparison of cosine similarity between CAVs and true concept direction (*top*) and concept separability as AUC (*bottom*), using filter- (lasso, logistic, ridge, and SVM) and pattern-CAV, and after each block of EfficientNet-B0 (*left*) and EfficientNetV2 (*right*) trained on FunnyBirds.

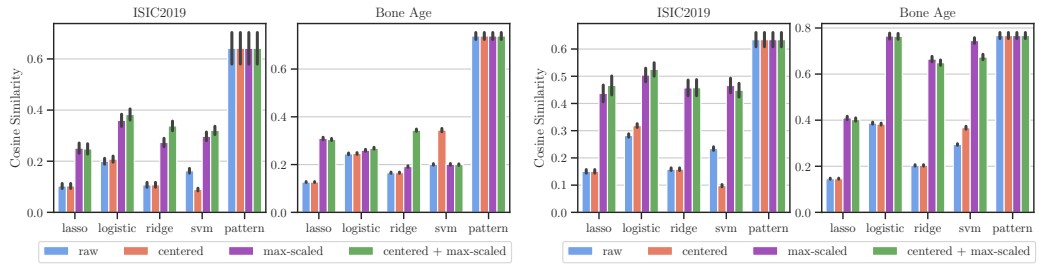

Figure 22: Cosine similarity between *true* concept direction $\mathbf{h}^{\text{gt}}$ and CAVs with different pre-processing methods fitted on the last Conv layer of ResNet18 (*left*) and ResNet50 (*right*) trained on ISIC2019 and Bone Age. Compared to filter-CAVs, pattern-CAV has a higher alignment with $\mathbf{h}^{\text{gt}}$ and is invariant to feature pre-processing.

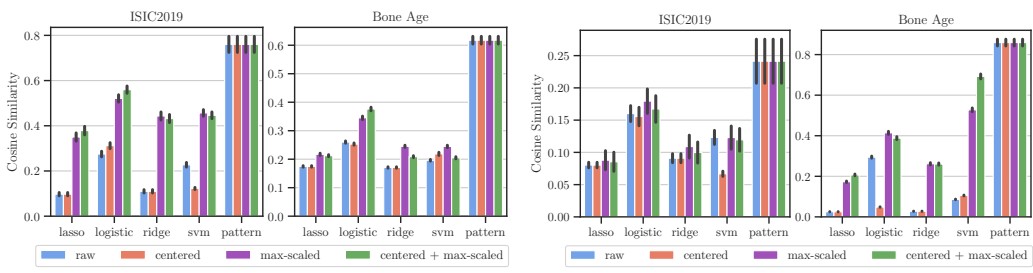

Figure 23: Cosine similarity between *true* concept direction $\mathbf{h}^{\text{gt}}$ and CAVs with different pre-processing methods fitted on the last Conv layer of ResNeXt50 (*left*) and ReXNet100 (*right*) trained on ISIC2019 and Bone Age. Compared to filter-CAVs, pattern-CAV has a higher alignment with $\mathbf{h}^{\text{gt}}$ and is invariant to feature pre-processing.

the fact that the analyzed layer in Vision Transformers is a fully-connected linear layer instead of a convolutional layer, $\text{TCAV}_{\text{sens}}(\mathbf{x})$ in Eq. 5 leads to a scalar per sample instead of per latent pixel. To test for statistical significance, following the original TCAV method, we ran a two-sided t-test for our controlled TCAV experiment with FunnyBirds conducted in Sec. 4.3.1. Specifically, we computed

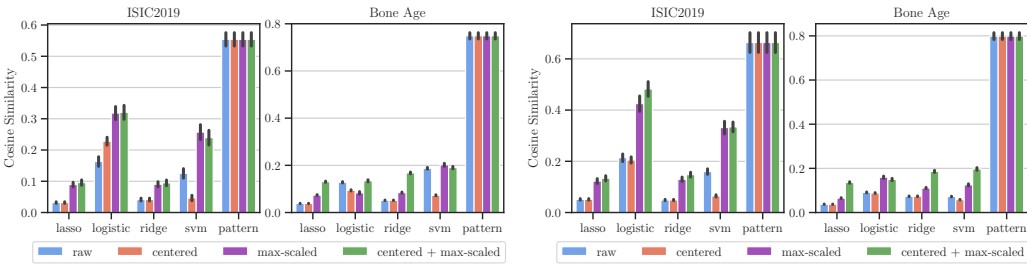

Figure 24: Cosine similarity between *true* concept direction $\mathbf{h}^{gt}$ and CAVs with different pre-processing methods fitted on the last Conv layer of EfficientNet-B0 (*left*) and EfficientNetV2 (*right*) trained on ISIC2019 and Bone Age. Compared to filter-CAVs, pattern-CAV has a higher alignment with $\mathbf{h}^{gt}$ and is invariant to feature pre-processing.

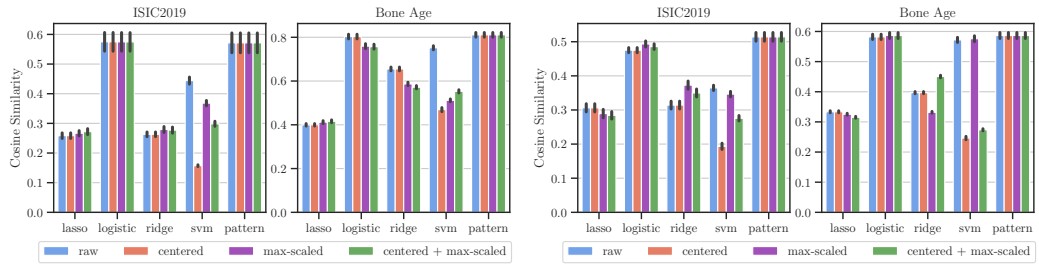

Figure 25: Cosine similarity between *true* concept direction $\mathbf{h}^{gt}$ and CAVs with different pre-processing methods fitted on the last Conv layer of Vision Transformer (*left*) and Swin Transformer (*right*) trained on ISIC2019 and Bone Age. Compared to filter-CAVs, pattern-CAV has a higher alignment with $\mathbf{h}^{gt}$ and is invariant to feature pre-processing.

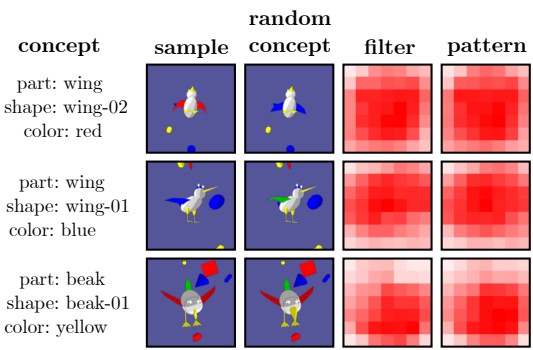

Figure 26: Visualization of concept sensitivity maps, measured as element-wise product $\boldsymbol{\nabla}_{\mathbf{a}}\tilde{f}(\mathbf{a}(\mathbf{a}))\odot \mathbf{h}$ using *filter-* (SVM) and *pattern*-CAVs for three concepts on the last Conv layer of ResNet18. All CAV variants detect a positive concept sensitivity (*red*) across all spatial locations ($7 \times 7$ pixels). This makes ResNet18 less susceptible to noise. Similar trends have been observed for ResNeXt50 and ReXNet100 models.

each CAV 500 times with different, randomly drawn subsets. Using a significance level of $5\%$ and applying a Bonferroni correction, all TCAV scores (for all CAV types, all 10 relevant concepts) are significantly different from the random baseline score of 0.5 (corresponding to $\Delta\mathrm{TCAV} = 0$), except for a few exceptions. Moreover, we collected accuracies for filter-based CAVs on an unseen test set and found that most CAVs achieve scores of above 0.9. This confirms that most filter-CAVs do not fail in fitting a generalizable decision boundary. Note, that hyperparameters for filter-CAVs have been tuned using an validation set. All TCAV scores, p-values, and accuracies for filter-CAVs on an unseen test set are shown in Tab. 5.

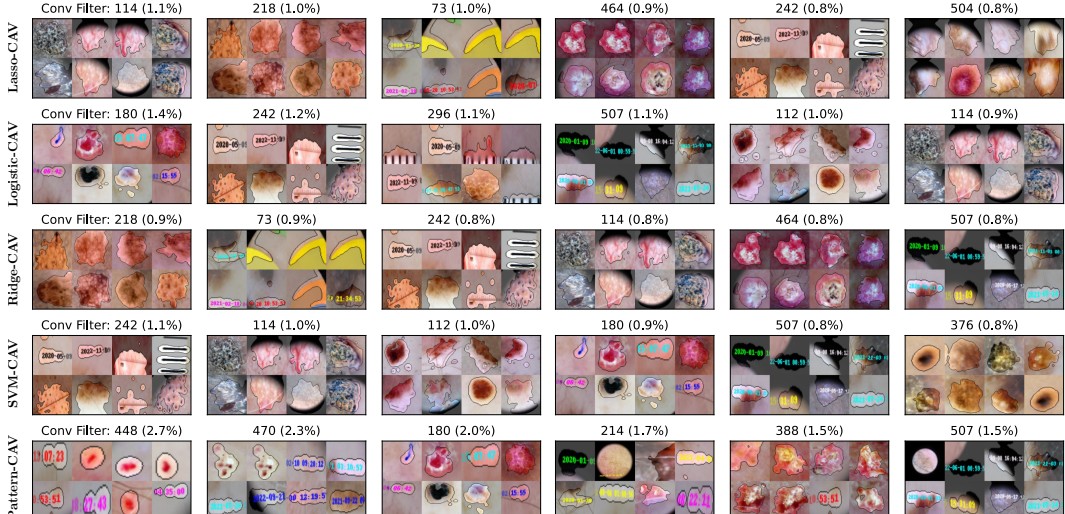

Figure 27: RelMax visualization for neurons corresponding to the largest absolute values in different CAVs, including 4 *filter-* (lasso, logistic, ridge, and SVM) and the *pattern*-CAVs, along with the Conv filter ID and the fraction of all (absolute) CAV values. While the filter-CAV picks up noisy neurons, the pattern-CAV uses neurons related to the relevant concept.

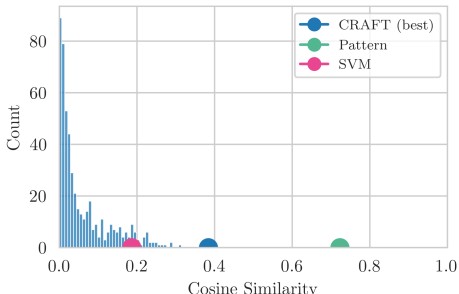

Figure 28: Histogram over CAV quality scores (cosine similarity with ground truth) for all CRAFT directions computed via non-negative matrix factorization on the last Conv layer of a VGG16 for ISIC2019 with timestamp artifact with markers for the best CRAFT direction, SVM-CAV, and pattern-CAV. The best CRAFT direction has a higher quality than SVM-CAV, but lower than pattern-CAV.

## D.6 ANALYSIS OF NOISE DISTRIBUTION

We further investigate the relation between the discrepancies in CAV quality found in our controlled in experiments in Section 4.2 and the issues described in Sec. 3.3, namely feature scaling and rotated noise.

**Feature scaling:** We plot the absolute difference between CAVs and ground truth concept direction (how much does CAV diverge?) over the variance per dimension (how varying are feature scales?). The results for VGG16, ResNet50, and EfficientNet-B0 with our controlled artifacts ISIC2019 and Pediatric Bone Age are shown in Figs. 32, 33, and 34. As expected, in all experiments higher variance leads to higher divergence for filter-CAVs but not for pattern-CAVs.

**Noise rotation:** To analyze the impact of distractor directions, we run a Principal Component Analysis (PCA) to find the direction with the highest within-cluster variance for latent activations of negative samples (without concept). We then computed the cosine similarity between that direction and the CAV. Results for VGG16, ResNet50, and EfficientNet-B0 models trained on ISIC2019 and Bone Age datasets are shown in Tab. 6. For the filter-CAV, we get a cosine similarity close to

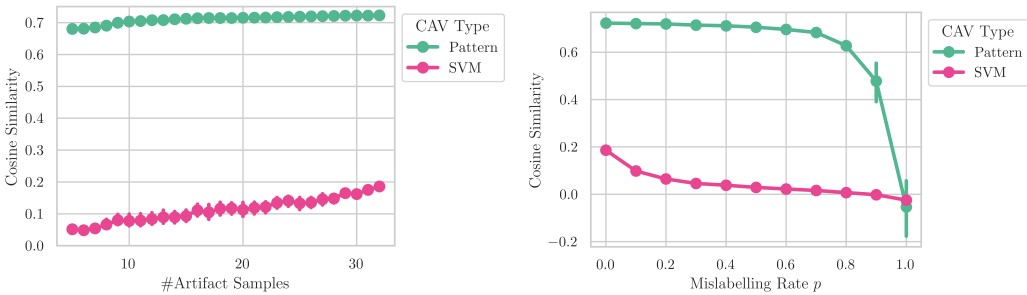

Figure 29: CAV quality (cosine similarity with ground truth) averaged over 10 random seeds plotted over number of known artifact samples (*left*) and mislabeling rate $p$ (*right*) for SVM- and pattern-CAVs trained on the penultimate layer of a VGG16 for ISIC2019 with artificial timestamp artifact. In addition to the fact that pattern-CAVs represent the concept direction more precisely, the cosine similarity stays consistently high even with less known artifact samples and high mislabeling rate.

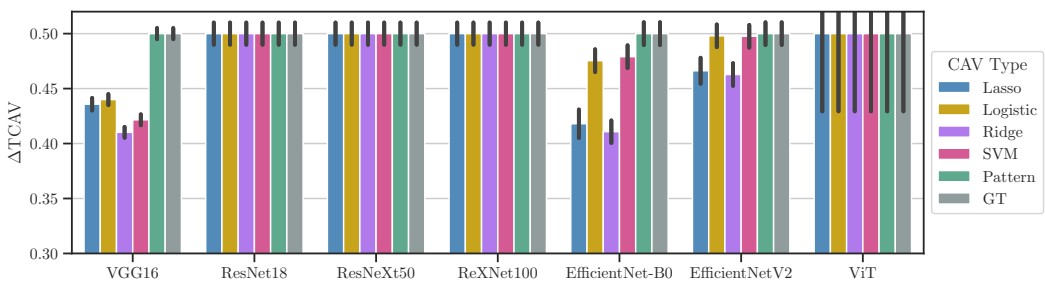

Figure 30: $\Delta$TCAV (averaged over class-defining concepts) for different CAVs fitted on last Conv layers of VGG16, ResNet18, ResNeXt50, ReXNet100, EfficientNet-B0, EfficientNetV2, and the last linear layer of a Vision Transformer trained on FunnyBirds. As models *must* use these concepts by experimental design, high scores are better. In contrast to filter-CAVs, pattern-CAVs achieve best scores for all models.

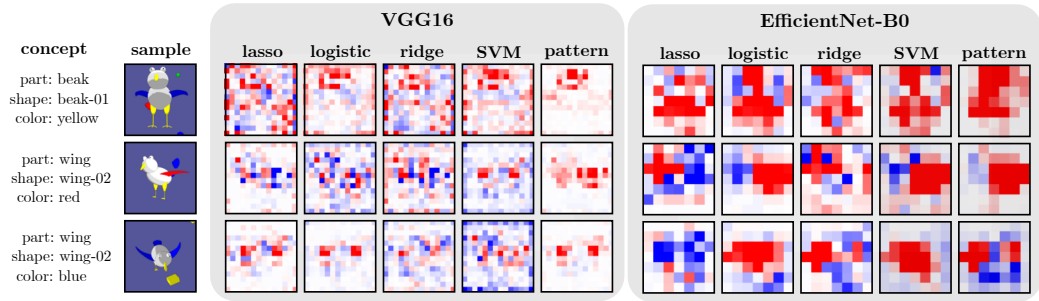

Figure 31: Visualization of sensitivity maps, measured as element-wise product $\boldsymbol{\nabla}_{\mathbf{a}}\tilde{f}(\mathbf{a}(\mathbf{a}))\odot\mathbf{h}$ using different *filter-* (lasso, logistic, ridge, and SVM) and *pattern*-CAVs for three concepts with VGG16 (*middle*) and EfficientNet-B0 (*right*). Results are shown for the respective last Conv layer, upsampled to input space dimensions. While pattern-CAVs precisely localize the concepts, filter-CAVs lead to noisy sensitivity maps.

0, meaning that it orients itself orthogonal to the (non-informative) distractor direction, while the pattern-CAV does not show this behavior. Similar trends can be seen in Fig. 7 in Appendix B.4, where filter-CAVs tend to orient themselves orthogonal to the distractor pattern.

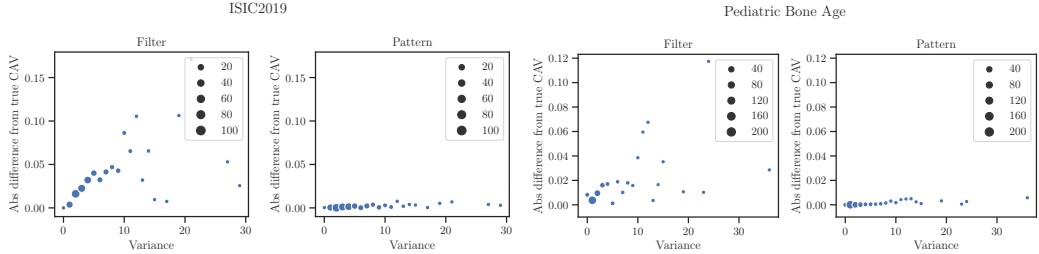

Figure 32: The absolute difference between CAVs and ground truth concept direction $\mathbf{h}^{\text{gt}}$ on the last convolutional layer of VGG16 models trained on ISIC2019 with timestamp artifact (*left*) and Pediatric Bone Age with brightness artifact (*right*) plotted over the variance per feature dimension. Higher variance leads to a larger difference from $\mathbf{h}^{\text{gt}}$ for filter-CAVs but not for pattern-CAVs.

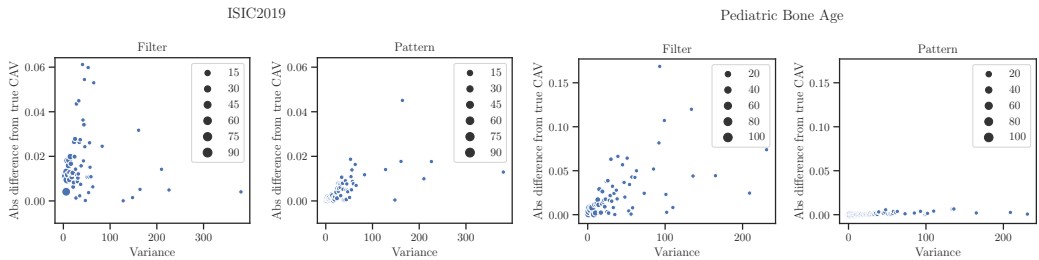

Figure 33: The absolute difference between CAVs and ground truth concept direction $\mathbf{h}^{\text{gt}}$ on the last convolutional layer of ResNet50 models trained on ISIC2019 with timestamp artifact (*left*) and Pediatric Bone Age with brightness artifact (*right*) plotted over the variance per feature dimension. Higher variance leads to a larger difference from $\mathbf{h}^{\text{gt}}$ for filter-CAVs but not for pattern-CAVs.

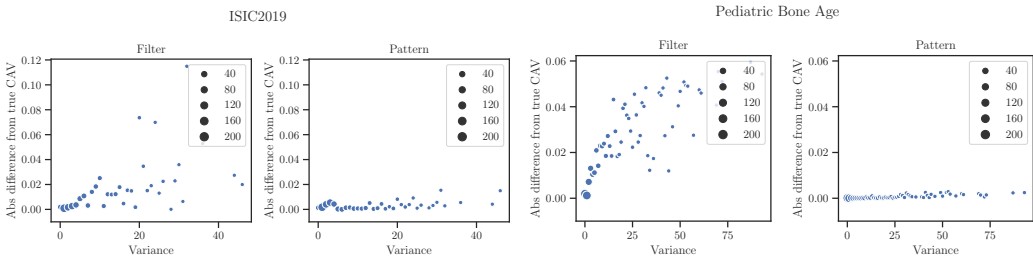

Figure 34: The absolute difference between CAVs and ground truth concept direction $\mathbf{h}^{\text{gt}}$ on the last convolutional layer of EfficientNet-B0 models trained on ISIC2019 with timestamp artifact (*left*) and Pediatric Bone Age with brightness artifact (*right*) plotted over the variance per feature dimension. Higher variance leads to a larger difference from $\mathbf{h}^{\text{gt}}$ for filter-CAVs but not for pattern-CAVs.

## D.7 MODEL CORRECTION WITH RR-CLARC

Model correction is performed with RR-ClArC for 10 epochs with the initial training learning rate (see Table 4) divided by 10. To balance between classification loss and the added loss term $L_{\text{RR}}$, we weigh the latter term with $\lambda \in \{10^5, 10^6, ..., 10^{10}\}$. The parameter is picked on the validation set and selected $\lambda$ values for all model correction experiments are shown in Tab. 7.

The results for our controlled datasets (Bone Age and ISIC2019) including standard errors are shown in Table 8. Moreover, Tab. 9 presents the model correction results for *all* artifacts ("band-aid", "ruler", and "skin marker"). Pattern-CAVs consistently yield better scores for artifact sensitivity, *i.e.*, low artifact relevance and $\Delta\text{TCAV}^{\text{gt}}$ after model correction.

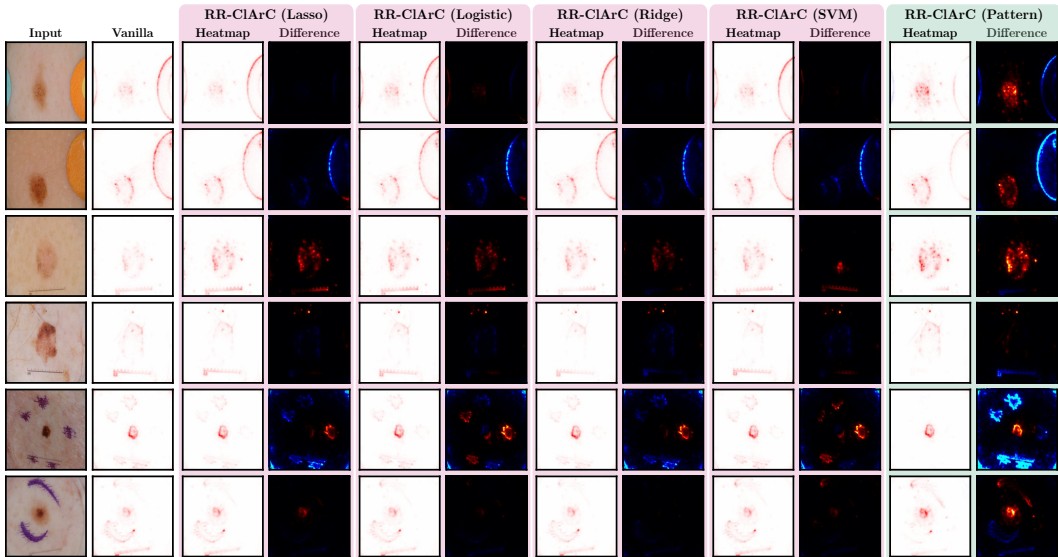

Figure 35: Additional qualitative results for model correction w.r.t. real artifacts band-aid (*top two*), ruler (*middle two*), and skin marker (*bottom two*) in ISIC2019 using VGG16. In addition to attribution heatmaps for models corrected with filter- (lasso, logistic, ridge, and SVM) and pattern-CAVs, we show heatmaps highlighting the differences compared to the Vanilla model attribution heatmap, with red and blue indicating higher and lower relevance after correction, respectively. Whereas filte-CAVs have limited impact, pattern-CAVs successfully increases the relevance on the mole and decreased the relevance on data artifacts.

Fig. 35 presents additional relevance heatmaps after model correction w.r.t. the real ISIC2019 artifacts for all CAV variants and their difference heatmap compared with the Vanilla model.

Table 5: TCAV scores, p-values and accuracies for all CAV types on VGG16, ResNet18, ResNeXt50, ReXNet100, EfficientNet-B0, EfficientNetV2 and Vision Transformer (ViT) models using the controlled FunnyBirds dataset. Bold p-values indicate that $\Delta$TCAV scores are not significantly different from 0 using a significance level of $5\%$ and applying a Bonferroni correction.

| Model | Concept | GT TCAV | GT p-val | Lasso TCAV | Lasso p-val | Lasso acc | Logistic TCAV | Logistic p-val | Logistic acc | Ridge TCAV | Ridge p-val | Ridge acc | SVM TCAV | SVM p-val | SVM acc | Signal TCAV | Signal p-val |
|---|---|---|---|---|---|---|---|---|---|---|---|---|---|---|---|---|---|
| VGG16 | beak01::yellow | 0.5 | 0.0 | 0.25 | **0.066** | 0.96 | 0.50 | 0.0 | 0.96 | 0.48 | 0.0 | 0.96 | 0.50 | 0.0 | 0.96 | 0.5 | 0.0 |
| | beak02::yellow | 0.5 | 0.0 | 0.35 | 0.001 | 0.98 | 0.50 | 0.0 | 0.97 | 0.25 | **0.192** | 0.98 | 0.49 | 0.0 | 0.98 | 0.5 | 0.0 |
| | beak03::yellow | 0.5 | 0.0 | 0.33 | **0.040** | 0.98 | 0.50 | 0.0 | 0.98 | 0.02 | **0.942** | 0.96 | 0.50 | 0.0 | 0.98 | 0.5 | 0.0 |
| | beak04::yellow | 0.5 | 0.0 | 0.45 | 0.0 | 0.99 | 0.08 | **0.457** | 0.98 | 0.38 | 0.0 | 0.98 | 0.16 | **0.015** | 0.99 | 0.5 | 0.0 |
| | wing01::blue | 0.5 | 0.0 | 0.50 | 0.0 | 0.94 | 0.44 | 0.0 | 0.94 | 0.50 | 0.0 | 0.94 | 0.36 | 0.0 | 0.99 | 0.5 | 0.0 |
| | wing01::green | 0.5 | 0.0 | 0.49 | 0.0 | 0.97 | 0.50 | 0.0 | 0.97 | 0.50 | 0.0 | 0.98 | 0.49 | 0.0 | 0.98 | 0.5 | 0.0 |
| | wing01::red | 0.5 | 0.0 | 0.49 | 0.0 | 0.95 | 0.40 | 0.005 | 0.91 | 0.49 | 0.0 | 0.96 | 0.27 | 0.0 | 1.00 | 0.5 | 0.0 |
| | wing02::blue | 0.5 | 0.0 | 0.50 | 0.0 | 0.97 | 0.48 | 0.0 | 0.99 | 0.50 | 0.0 | 0.98 | 0.50 | 0.0 | 0.99 | 0.5 | 0.0 |
| | wing02::green | 0.5 | 0.0 | 0.50 | 0.0 | 0.96 | 0.50 | 0.0 | 0.98 | 0.50 | 0.0 | 0.98 | 0.50 | 0.0 | 0.99 | 0.5 | 0.0 |
| | wing02::red | 0.5 | 0.0 | 0.50 | 0.0 | 0.98 | 0.50 | 0.0 | 0.98 | 0.49 | 0.0 | 0.98 | 0.45 | 0.0 | 0.99 | 0.5 | 0.0 |
| ResNet18 | beak01::yellow | 0.5 | 0.0 | 0.5 | 0.0 | 0.48 | 0.5 | 0.0 | 0.90 | 0.5 | 0.0 | 0.78 | 0.5 | 0.0 | 0.97 | 0.5 | 0.0 |
| | beak02::yellow | 0.5 | 0.0 | 0.5 | 0.0 | 0.98 | 0.5 | 0.0 | 0.93 | 0.5 | 0.0 | 0.98 | 0.5 | 0.0 | 0.98 | 0.5 | 0.0 |
| | beak03::yellow | 0.5 | 0.0 | 0.5 | 0.0 | 0.98 | 0.5 | 0.0 | 0.98 | 0.5 | 0.0 | 0.98 | 0.5 | 0.0 | 0.99 | 0.5 | 0.0 |
| | beak04::yellow | 0.5 | 0.0 | 0.5 | 0.0 | 0.95 | 0.5 | 0.0 | 0.98 | 0.5 | 0.0 | 0.95 | 0.5 | 0.0 | 0.98 | 0.5 | 0.0 |
| | wing01::blue | 0.5 | 0.0 | 0.5 | 0.0 | 0.80 | 0.5 | 0.0 | 0.95 | 0.5 | 0.0 | 0.87 | 0.5 | 0.0 | 0.99 | 0.5 | 0.0 |
| | wing01::green | 0.5 | 0.0 | 0.5 | 0.0 | 0.98 | 0.5 | 0.0 | 0.97 | 0.5 | 0.0 | 0.95 | 0.5 | 0.0 | 1.00 | 0.5 | 0.0 |
| | wing01::red | 0.5 | 0.0 | 0.5 | 0.0 | 0.98 | 0.5 | 0.0 | 0.94 | 0.5 | 0.0 | 0.98 | 0.5 | 0.0 | 0.99 | 0.5 | 0.0 |
| | wing02::blue | 0.5 | 0.0 | 0.5 | 0.0 | 0.94 | 0.5 | 0.0 | 0.93 | 0.5 | 0.0 | 0.95 | 0.5 | 0.0 | 1.00 | 0.5 | 0.0 |
| | wing02::green | 0.5 | 0.0 | 0.5 | 0.0 | 1.00 | 0.5 | 0.0 | 0.98 | 0.5 | 0.0 | 1.00 | 0.5 | 0.0 | 1.00 | 0.5 | 0.0 |
| | wing02::red | 0.5 | 0.0 | 0.5 | 0.0 | 0.98 | 0.5 | 0.0 | 0.97 | 0.5 | 0.0 | 0.97 | 0.5 | 0.0 | 0.99 | 0.5 | 0.0 |
| ResNeXt50 | beak01::yellow | 0.5 | 0.0 | 0.5 | 0.0 | 0.96 | 0.5 | 0.0 | 0.99 | 0.5 | 0.0 | 0.98 | 0.5 | 0.0 | 0.99 | 0.5 | 0.0 |
| | beak02::yellow | 0.5 | 0.0 | 0.5 | 0.0 | 0.97 | 0.5 | 0.0 | 1.00 | 0.5 | 0.0 | 0.97 | 0.5 | 0.0 | 1.00 | 0.5 | 0.0 |
| | beak03::yellow | 0.5 | 0.0 | 0.5 | 0.0 | 0.98 | 0.5 | 0.0 | 0.99 | 0.5 | 0.0 | 0.99 | 0.5 | 0.0 | 0.99 | 0.5 | 0.0 |
| | beak04::yellow | 0.5 | 0.0 | 0.5 | 0.0 | 0.93 | 0.5 | 0.0 | 0.96 | 0.5 | 0.0 | 0.92 | 0.5 | 0.0 | 0.99 | 0.5 | 0.0 |
| | wing01::blue | 0.5 | 0.0 | 0.5 | 0.0 | 0.96 | 0.5 | 0.0 | 0.99 | 0.5 | 0.0 | 0.99 | 0.5 | 0.0 | 0.99 | 0.5 | 0.0 |
| | wing01::green | 0.5 | 0.0 | 0.5 | 0.0 | 0.70 | 0.5 | 0.0 | 0.90 | 0.5 | 0.0 | 0.98 | 0.5 | 0.0 | 0.99 | 0.5 | 0.0 |
| | wing01::red | 0.5 | 0.0 | 0.5 | 0.0 | 0.93 | 0.5 | 0.0 | 0.92 | 0.5 | 0.0 | 0.94 | 0.5 | 0.0 | 0.99 | 0.5 | 0.0 |
| | wing02::blue | 0.5 | 0.0 | 0.5 | 0.0 | 0.98 | 0.5 | 0.0 | 1.00 | 0.5 | 0.0 | 0.98 | 0.5 | 0.0 | 1.00 | 0.5 | 0.0 |
| | wing02::green | 0.5 | 0.0 | 0.5 | 0.0 | 0.95 | 0.5 | 0.0 | 0.98 | 0.5 | 0.0 | 0.96 | 0.5 | 0.0 | 0.99 | 0.5 | 0.0 |
| | wing02::red | 0.5 | 0.0 | 0.5 | 0.0 | 0.90 | 0.5 | 0.0 | 0.99 | 0.5 | 0.0 | 0.98 | 0.5 | 0.0 | 1.00 | 0.5 | 0.0 |
| ReXNet100 | beak01::yellow | 0.5 | 0.0 | 0.5 | 0.0 | 0.98 | 0.5 | 0.0 | 0.90 | 0.5 | 0.0 | 0.98 | 0.5 | 0.0 | 0.99 | 0.5 | 0.0 |
| | beak02::yellow | 0.5 | 0.0 | 0.5 | 0.0 | 0.99 | 0.5 | 0.0 | 0.99 | 0.5 | 0.0 | 0.98 | 0.5 | 0.0 | 0.99 | 0.5 | 0.0 |
| | beak03::yellow | 0.5 | 0.0 | 0.5 | 0.0 | 0.99 | 0.5 | 0.0 | 0.99 | 0.5 | 0.0 | 0.99 | 0.5 | 0.0 | 0.99 | 0.5 | 0.0 |
| | beak04::yellow | 0.5 | 0.0 | 0.5 | 0.0 | 0.97 | 0.5 | 0.0 | 0.98 | 0.5 | 0.0 | 0.97 | 0.5 | 0.0 | 0.98 | 0.5 | 0.0 |
| | wing01::blue | 0.5 | 0.0 | 0.5 | 0.0 | 0.99 | 0.5 | 0.0 | 0.95 | 0.5 | 0.0 | 0.99 | 0.5 | 0.0 | 1.00 | 0.5 | 0.0 |
| | wing01::green | 0.5 | 0.0 | 0.5 | 0.0 | 1.00 | 0.5 | 0.0 | 0.99 | 0.5 | 0.0 | 1.00 | 0.5 | 0.0 | 1.00 | 0.5 | 0.0 |
| | wing01::red | 0.5 | 0.0 | 0.5 | 0.0 | 0.99 | 0.5 | 0.0 | 0.99 | 0.5 | 0.0 | 1.00 | 0.5 | 0.0 | 0.99 | 0.5 | 0.0 |
| | wing02::blue | 0.5 | 0.0 | 0.5 | 0.0 | 0.99 | 0.5 | 0.0 | 0.99 | 0.5 | 0.0 | 0.99 | 0.5 | 0.0 | 0.99 | 0.5 | 0.0 |
| | wing02::green | 0.5 | 0.0 | 0.5 | 0.0 | 0.99 | 0.5 | 0.0 | 0.99 | 0.5 | 0.0 | 0.99 | 0.5 | 0.0 | 1.00 | 0.5 | 0.0 |
| | wing02::red | 0.5 | 0.0 | 0.5 | 0.0 | 0.99 | 0.5 | 0.0 | 0.96 | 0.5 | 0.0 | 0.99 | 0.5 | 0.0 | 1.00 | 0.5 | 0.0 |
| EfficientNet-B0 | beak01::yellow | 0.5 | 0.0 | 0.43 | 0.0 | 0.92 | 0.44 | 0.0 | 0.86 | 0.31 | 0.0 | 0.93 | 0.44 | 0.0 | 0.97 | 0.5 | 0.0 |
| | beak02::yellow | 0.5 | 0.0 | 0.18 | **0.124** | 0.98 | 0.33 | 0.0 | 0.99 | 0.22 | **0.038** | 0.99 | 0.36 | 0.001 | 0.99 | 0.5 | 0.0 |
| | beak03::yellow | 0.5 | 0.0 | 0.32 | 0.0 | 0.99 | 0.50 | 0.0 | 0.98 | 0.46 | 0.0 | 0.98 | 0.50 | 0.0 | 0.98 | 0.5 | 0.0 |
| | beak04::yellow | 0.5 | 0.0 | 0.48 | 0.0 | 0.98 | 0.49 | 0.0 | 0.99 | 0.49 | 0.0 | 0.97 | 0.49 | 0.0 | 0.99 | 0.5 | 0.0 |
| | wing01::blue | 0.5 | 0.0 | 0.43 | 0.0 | 0.98 | 0.50 | 0.0 | 0.99 | 0.46 | 0.0 | 0.99 | 0.50 | 0.0 | 0.99 | 0.5 | 0.0 |
| | wing01::green | 0.5 | 0.0 | 0.50 | 0.0 | 0.50 | 0.50 | 0.0 | 0.99 | 0.48 | 0.0 | 0.99 | 0.50 | 0.0 | 0.99 | 0.5 | 0.0 |
| | wing01::red | 0.5 | 0.0 | 0.47 | 0.0 | 0.99 | 0.50 | 0.0 | 0.99 | 0.48 | 0.0 | 0.99 | 0.50 | 0.0 | 0.99 | 0.5 | 0.0 |
| | wing02::blue | 0.5 | 0.0 | 0.39 | **0.007** | 0.79 | 0.50 | 0.0 | 0.98 | 0.34 | 0.0 | 0.99 | 0.50 | 0.0 | 1.00 | 0.5 | 0.0 |
| | wing02::green | 0.5 | 0.0 | 0.49 | 0.0 | 0.71 | 0.50 | 0.0 | 0.99 | 0.50 | 0.0 | 0.96 | 0.50 | 0.0 | 0.99 | 0.5 | 0.0 |
| | wing02::red | 0.5 | 0.0 | 0.50 | 0.0 | 0.49 | 0.50 | 0.0 | 0.98 | 0.38 | 0.0 | 0.97 | 0.50 | 0.0 | 1.00 | 0.5 | 0.0 |
| EfficientNetV2 | beak01::yellow | 0.5 | 0.0 | 0.50 | 0.0 | 0.80 | 0.50 | 0.0 | 0.70 | 0.50 | 0.0 | 0.87 | 0.50 | 0.0 | 0.98 | 0.5 | 0.0 |
| | beak02::yellow | 0.5 | 0.0 | 0.30 | **0.012** | 0.93 | 0.50 | 0.0 | 0.97 | 0.37 | 0.001 | 0.93 | 0.50 | 0.0 | 0.98 | 0.5 | 0.0 |
| | beak03::yellow | 0.5 | 0.0 | 0.46 | 0.0 | 0.98 | 0.49 | 0.0 | 0.99 | 0.46 | 0.0 | 0.98 | 0.49 | 0.0 | 0.99 | 0.5 | 0.0 |
| | beak04::yellow | 0.5 | 0.0 | 0.49 | 0.0 | 0.97 | 0.49 | 0.0 | 0.96 | 0.44 | 0.0 | 0.97 | 0.49 | 0.0 | 0.99 | 0.5 | 0.0 |
| | wing01::blue | 0.5 | 0.0 | 0.50 | 0.0 | 0.98 | 0.50 | 0.0 | 0.96 | 0.50 | 0.0 | 0.99 | 0.50 | 0.0 | 1.00 | 0.5 | 0.0 |
| | wing01::green | 0.5 | 0.0 | 0.45 | 0.0 | 1.00 | 0.50 | 0.0 | 1.00 | 0.48 | 0.0 | 1.00 | 0.50 | 0.0 | 1.00 | 0.5 | 0.0 |
| | wing01::red | 0.5 | 0.0 | 0.50 | 0.0 | 0.99 | 0.50 | 0.0 | 0.99 | 0.50 | 0.0 | 0.99 | 0.50 | 0.0 | 0.99 | 0.5 | 0.0 |
| | wing02::blue | 0.5 | 0.0 | 0.48 | 0.0 | 0.93 | 0.50 | 0.0 | 0.94 | 0.50 | 0.0 | 0.92 | 0.50 | 0.0 | 1.00 | 0.5 | 0.0 |
| | wing02::green | 0.5 | 0.0 | 0.50 | 0.0 | 0.51 | 0.50 | 0.0 | 0.99 | 0.47 | 0.0 | 0.99 | 0.50 | 0.0 | 0.99 | 0.5 | 0.0 |
| | wing02::red | 0.5 | 0.0 | 0.48 | 0.0 | 0.99 | 0.50 | 0.0 | 1.00 | 0.41 | 0.0 | 0.98 | 0.50 | 0.0 | 1.00 | 0.5 | 0.0 |
| ViT | beak01::yellow | 0.5 | 0.0 | 0.5 | 0.0 | 0.80 | 0.5 | 0.0 | 0.92 | 0.5 | 0.0 | 0.81 | 0.5 | 0.0 | 0.95 | 0.5 | 0.0 |
| | beak02::yellow | 0.5 | 0.0 | 0.5 | 0.0 | 0.94 | 0.5 | 0.0 | 0.97 | 0.5 | 0.0 | 0.95 | 0.5 | 0.0 | 0.98 | 0.5 | 0.0 |
| | beak03::yellow | 0.5 | 0.0 | 0.5 | 0.0 | 0.95 | 0.5 | 0.0 | 0.97 | 0.5 | 0.0 | 0.95 | 0.5 | 0.0 | 0.98 | 0.5 | 0.0 |
| | beak04::yellow | 0.5 | 0.0 | 0.5 | 0.0 | 0.97 | 0.5 | 0.0 | 0.96 | 0.5 | 0.0 | 0.98 | 0.5 | 0.0 | 0.99 | 0.5 | 0.0 |
| | wing01::blue | 0.5 | 0.0 | 0.5 | 0.0 | 0.94 | 0.5 | 0.0 | 0.99 | 0.5 | 0.0 | 0.96 | 0.5 | 0.0 | 0.99 | 0.5 | 0.0 |
| | wing01::green | 0.5 | 0.0 | 0.5 | 0.0 | 0.96 | 0.5 | 0.0 | 0.99 | 0.5 | 0.0 | 0.96 | 0.5 | 0.0 | 0.99 | 0.5 | 0.0 |
| | wing01::red | 0.5 | 0.0 | 0.5 | 0.0 | 0.94 | 0.5 | 0.0 | 0.98 | 0.5 | 0.0 | 0.97 | 0.5 | 0.0 | 0.99 | 0.5 | 0.0 |
| | wing02::blue | 0.5 | 0.0 | 0.5 | 0.0 | 0.96 | 0.5 | 0.0 | 0.99 | 0.5 | 0.0 | 0.97 | 0.5 | 0.0 | 0.99 | 0.5 | 0.0 |
| | wing02::green | 0.5 | 0.0 | 0.5 | 0.0 | 0.95 | 0.5 | 0.0 | 0.95 | 0.5 | 0.0 | 0.96 | 0.5 | 0.0 | 0.99 | 0.5 | 0.0 |
| | wing02::red | 0.5 | 0.0 | 0.5 | 0.0 | 0.97 | 0.5 | 0.0 | 1.00 | 0.5 | 0.0 | 0.98 | 0.5 | 0.0 | 1.00 | 0.5 | 0.0 |

Table 6: Cosine similarity between CAVs and (non-informative) distractor direction computed as direction with highest within-cluster variance for activations of negative samples (without concept) using PCA. Filter-CAVs orient themselves orthogonal to the distractor direction (*i.e.*, cosine similarity close to 0), while pattern-CAVs do not show this behavior.

| Dataset | Model | Cosine Similarity | |
| --- | --- | --- | --- |
| | | Filter-CAV | Pattern-CAV |
| ISIC (timestamp) | VGG16 | -0.036 | 0.536 |
| | ResNet50 | 0.048 | 0.155 |
| | EfficientNet-B0 | -0.008 | 0.041 |
| Bone (brightness) | VGG16 | -0.003 | 0.169 |
| | ResNet50 | 0.005 | 0.076 |
| | EfficientNet-B0 | 0.005 | -0.057 |

Table 7: Selected $\lambda$ values for model correction with RR-ClArC as weight for the added loss term for experiments with VGG16, ResNet18, ResNet50, ResNeXt50, ReXNet100, EfficientNet-B0, EfficientNetV2, and ViT with Bone Age (controlled), ISIC2019 (controlled), and ISIC2019 (real artifacts). For the latter, we run separate corrections w.r.t. the "band-aid" (BA), "ruler" (R), and "skin marker" (SM) artifacts.

| model | CAV | Bone Age (controlled) | ISIC2019 (controlled) | ISIC2019 (BA\|R\|SM) |
|---|---|---|---|---|
| VGG16 | Lasso | $10^7$ | $10^7$ | $10^7\|10^8\|10^7$ |
| | Logistic | $10^7$ | $10^7$ | $10^7\|10^8\|10^7$ |
| | Ridge | $10^7$ | $10^7$ | $10^7\|10^8\|10^7$ |
| | SVM | $10^7$ | $10^7$ | $10^7\|10^7\|10^7$ |
| | Pattern | $10^7$ | $10^7$ | $10^6\|10^9\|10^7$ |
| ResNet18 | Lasso | $10^9$ | $10^7$ | $10^5\|10^7\|10^6$ |
| | Logistic | $10^{10}$ | $10^6$ | $10^5\|10^5\|10^5$ |
| | Ridge | $10^9$ | $10^5$ | $10^5\|10^5\|10^5$ |
| | SVM | $10^{10}$ | $10^7$ | $10^5\|10^5\|10^5$ |
| | Pattern | $10^4$ | $10^6$ | $10^5\|10^5\|10^5$ |
| ResNet50 | Lasso | $10^4$ | $10^{10}$ | $10^5\|10^5\|10^5$ |
| | Logistic | $10^4$ | $10^6$ | $10^5\|10^5\|10^9$ |
| | Ridge | $10^4$ | $10^5$ | $10^5\|10^5\|10^5$ |
| | SVM | $10^5$ | $10^5$ | $10^5\|10^5\|10^7$ |
| | Pattern | $10^4$ | $10^6$ | $10^5\|10^5\|10^5$ |
| ResNeXt50 | Lasso | $10^5$ | $10^6$ | $10^6\|10^7\|10^6$ |
| | Logistic | $10^8$ | $10^6$ | $10^5\|10^8\|10^6$ |
| | Ridge | $10^9$ | $10^6$ | $10^6\|10^8\|10^6$ |
| | SVM | $10^{10}$ | $10^6$ | $10^6\|10^8\|10^5$ |
| | Pattern | $10^4$ | $10^5$ | $10^5\|10^7\|10^5$ |
| ReXNet100 | Lasso | $10^6$ | $10^9$ | $10^6\|10^5\|10^9$ |
| | Logistic | $10^{10}$ | $10^7$ | $10^5\|10^5\|10^5$ |
| | Ridge | $10^{10}$ | $10^8$ | $10^5\|10^5\|10^5$ |
| | SVM | $10^{10}$ | $10^8$ | $10^6\|10^5\|10^5$ |
| | Pattern | $10^4$ | $10^7$ | $10^5\|10^5\|10^6$ |
| Efficient Net-B0 | Lasso | $10^9$ | $10^9$ | $10^6\|10^8\|10^8$ |
| | Logistic | $10^{10}$ | $10^{10}$ | $10^5\|10^6\|10^5$ |
| | Ridge | $10^9$ | $10^9$ | $10^5\|10^5\|10^5$ |
| | SVM | $10^{10}$ | $10^9$ | $10^5\|10^5\|10^5$ |
| | Pattern | $10^{10}$ | $10^9$ | $10^8\|10^5\|10^6$ |
| EfficientNetV2 | Lasso | $10^7$ | $10^5$ | $10^5\|10^9\|10^8$ |
| | Logistic | $10^6$ | $10^7$ | $10^5\|10^5\|10^5$ |
| | Ridge | $10^6$ | $10^5$ | $10^5\|10^9\|10^7$ |
| | SVM | $10^6$ | $10^6$ | $10^5\|10^8\|10^5$ |
| | Pattern | $10^7$ | $10^6$ | $10^5\|10^5\|10^5$ |
| ViT | Lasso | $10^6$ | $10^5$ | $10^6\|10^5\|10^5$ |
| | Logistic | $10^4$ | $10^5$ | $10^5\|10^6\|10^9$ |
| | Ridge | $10^6$ | $10^7$ | $10^6\|10^5\|10^8$ |
| | SVM | $10^6$ | $10^6$ | $10^6\|10^5\|10^5$ |
| | Pattern | $10^4$ | $10^5$ | $10^5\|10^5\|10^6$ |

Table 8: Results after model correction with RR-ClArC for VGG16, ResNet18/50, ResNeXt50, ReXNet100, EfficientNet-B0, EfficientNetV2, and ViT trained on Bone Age | ISIC2019 (controlled) including standard errors. We report accuracy on clean and biased test set, the fraction of relevance put onto the data artifact region for localizable artifacts, and the TCAV score (reported as $\Delta\mathrm{TCAV}^{\mathrm{gt}}$) using the sample-wise ground-truth concept direction $\mathbf{h}^{\mathrm{gt}}$, measuring the models' sensitivity towards the artifacts *after* model correction. Stars indicate statistical significance according to z-tests with a significance level of 0.05, and arrows whether low ($\downarrow$) or high ($\uparrow$) are better.

| model | CAV | Accuracy (clean) $\uparrow$ | | Accuracy (biased) $\uparrow$ | | Artifact relevance $\downarrow$ | | $\Delta\mathrm{TCAV}^{\mathrm{gt}} \downarrow$ | |
|---|---|---|---|---|---|---|---|---|---|
| VGG-16 | *Vanilla* | $0.78 \pm 0.01$ | $0.82 \pm 0.01$ | $0.50 \pm 0.01$ | $0.28 \pm 0.01$ | - | $0.62 \pm 0.01$ | $0.29 \pm 0.00$ | $0.14 \pm 0.02$ |
| | lasso | $0.77 \pm 0.01$ | $0.82 \pm 0.01$ | $0.55 \pm 0.01$ | $0.30 \pm 0.01$ | - | $0.60 \pm 0.01$ | $0.25 \pm 0.00$ | $0.13 \pm 0.02$ |
| | logistic | $0.72 \pm 0.01$ | $0.82 \pm 0.01$ | $0.63 \pm 0.01$ | $0.37 \pm 0.01$ | - | $0.54 \pm 0.01$ | $0.25 \pm 0.00$ | $\mathbf{0.07 \pm 0.02}^*$ |
| | ridge | $0.71 \pm 0.01$ | $0.82 \pm 0.01$ | $0.61 \pm 0.01$ | $0.31 \pm 0.01$ | - | $0.59 \pm 0.01$ | $0.24 \pm 0.00$ | $0.13 \pm 0.02$ |
| | SVM | $0.69 \pm 0.01$ | $0.81 \pm 0.01$ | $0.70 \pm 0.01$ | $0.36 \pm 0.01$ | - | $0.55 \pm 0.01$ | $0.24 \pm 0.00$ | $0.10 \pm 0.02$ |
| | Pattern | $0.78 \pm 0.01$ | $0.80 \pm 0.01$ | $\mathbf{0.75 \pm 0.01}^*$ | $\mathbf{0.69 \pm 0.01}^*$ | - | $\mathbf{0.26 \pm 0.01}^*$ | $\mathbf{0.14 \pm 0.00}^*$ | $0.10 \pm 0.02$ |
| ResNet-18 | *Vanilla* | $0.75 \pm 0.01$ | $0.84 \pm 0.01$ | $0.46 \pm 0.01$ | $0.43 \pm 0.01$ | - | $0.31 \pm 0.01$ | $0.50 \pm 0.00$ | $0.50 \pm 0.00$ |
| | lasso | $0.76 \pm 0.01$ | $0.83 \pm 0.01$ | $0.55 \pm 0.01$ | $0.51 \pm 0.01$ | - | $0.27 \pm 0.01$ | $0.08 \pm 0.00$ | $0.01 \pm 0.02$ |
| | logistic | $0.76 \pm 0.01$ | $0.83 \pm 0.01$ | $0.55 \pm 0.01$ | $0.59 \pm 0.01$ | - | $0.25 \pm 0.01$ | $0.07 \pm 0.00$ | $0.01 \pm 0.02$ |
| | ridge | $0.77 \pm 0.01$ | $0.83 \pm 0.01$ | $0.57 \pm 0.01$ | $0.51 \pm 0.01$ | - | $0.27 \pm 0.01$ | $\mathbf{0.00 \pm 0.00}^*$ | $\mathbf{0.00 \pm 0.02}$ |
| | SVM | $0.76 \pm 0.01$ | $0.83 \pm 0.01$ | $0.54 \pm 0.01$ | $0.57 \pm 0.01$ | - | $0.26 \pm 0.01$ | $0.07 \pm 0.00$ | $0.01 \pm 0.02$ |
| | Pattern | $0.75 \pm 0.01$ | $0.83 \pm 0.01$ | $\mathbf{0.59 \pm 0.01}$ | $\mathbf{0.66 \pm 0.01}^*$ | - | $\mathbf{0.22 \pm 0.01}^*$ | $0.22 \pm 0.01$ | $0.05 \pm 0.02$ |
| ResNet50 | *Vanilla* | $0.77 \pm 0.01$ | $0.85 \pm 0.01$ | $0.48 \pm 0.01$ | $0.51 \pm 0.01$ | - | $0.46 \pm 0.01$ | $0.14 \pm 0.00$ | $0.04 \pm 0.02$ |
| | lasso | $0.77 \pm 0.01$ | $0.84 \pm 0.01$ | $0.53 \pm 0.01$ | $0.58 \pm 0.01$ | - | $0.44 \pm 0.01$ | $0.03 \pm 0.00$ | $\mathbf{0.02 \pm 0.01}$ |
| | logistic | $0.77 \pm 0.01$ | $0.85 \pm 0.01$ | $0.55 \pm 0.01$ | $0.69 \pm 0.01$ | - | $0.39 \pm 0.01$ | $0.04 \pm 0.00$ | $0.05 \pm 0.02$ |
| | ridge | $0.77 \pm 0.01$ | $0.84 \pm 0.01$ | $0.52 \pm 0.01$ | $0.58 \pm 0.01$ | - | $0.45 \pm 0.01$ | $0.13 \pm 0.00$ | $0.03 \pm 0.01$ |
| | SVM | $0.77 \pm 0.01$ | $0.85 \pm 0.01$ | $0.55 \pm 0.01$ | $0.68 \pm 0.01$ | - | $0.40 \pm 0.01$ | $0.04 \pm 0.00$ | $0.04 \pm 0.02$ |
| | Pattern | $0.78 \pm 0.01$ | $0.84 \pm 0.01$ | $\mathbf{0.59 \pm 0.01}^*$ | $\mathbf{0.71 \pm 0.01}$ | - | $\mathbf{0.37 \pm 0.01}^*$ | $\mathbf{0.01 \pm 0.00}$ | $0.03 \pm 0.02$ |
| ResNeXt50 | *Vanilla* | $0.78 \pm 0.01$ | $0.86 \pm 0.01$ | $0.50 \pm 0.01$ | $0.45 \pm 0.01$ | - | $0.60 \pm 0.01$ | $0.04 \pm 0.00$ | $0.12 \pm 0.02$ |
| | lasso | $0.80 \pm 0.01$ | $0.84 \pm 0.01$ | $0.57 \pm 0.01$ | $0.64 \pm 0.01$ | - | $0.56 \pm 0.01$ | $\mathbf{0.01 \pm 0.00}$ | $\mathbf{0.01 \pm 0.01}$ |
| | logistic | $0.80 \pm 0.01$ | $0.85 \pm 0.01$ | $0.56 \pm 0.01$ | $0.73 \pm 0.01$ | - | $0.52 \pm 0.01$ | $0.05 \pm 0.00$ | $0.08 \pm 0.02$ |
| | ridge | $0.80 \pm 0.01$ | $0.84 \pm 0.01$ | $0.57 \pm 0.01$ | $0.63 \pm 0.01$ | - | $0.56 \pm 0.01$ | $0.06 \pm 0.00$ | $\mathbf{0.01 \pm 0.01}$ |
| | SVM | $0.78 \pm 0.01$ | $0.85 \pm 0.01$ | $0.54 \pm 0.01$ | $0.69 \pm 0.01$ | - | $0.54 \pm 0.01$ | $0.06 \pm 0.00$ | $0.09 \pm 0.02$ |
| | Pattern | $0.79 \pm 0.01$ | $0.85 \pm 0.01$ | $\mathbf{0.64 \pm 0.01}^*$ | $\mathbf{0.75 \pm 0.01}$ | - | $\mathbf{0.49 \pm 0.01}^*$ | $0.45 \pm 0.00$ | $0.03 \pm 0.02$ |
| ReXNet-100 | *Vanilla* | $0.76 \pm 0.01$ | $0.88 \pm 0.01$ | $0.47 \pm 0.01$ | $0.71 \pm 0.01$ | - | $0.22 \pm 0.01$ | $0.29 \pm 0.01$ | $0.50 \pm 0.00$ |
| | lasso | $0.77 \pm 0.01$ | $0.88 \pm 0.01$ | $0.46 \pm 0.01$ | $0.73 \pm 0.01$ | - | $0.23 \pm 0.01$ | $0.39 \pm 0.01$ | $0.16 \pm 0.03$ |
| | logistic | $0.77 \pm 0.01$ | $0.88 \pm 0.01$ | $0.47 \pm 0.01$ | $0.74 \pm 0.01$ | - | $0.22 \pm 0.01$ | $0.24 \pm 0.01$ | $0.17 \pm 0.03$ |
| | ridge | $0.77 \pm 0.01$ | $0.88 \pm 0.01$ | $0.46 \pm 0.01$ | $0.74 \pm 0.01$ | - | $0.22 \pm 0.01$ | $0.27 \pm 0.01$ | $0.15 \pm 0.03$ |
| | SVM | $0.77 \pm 0.01$ | $0.88 \pm 0.01$ | $0.47 \pm 0.01$ | $0.74 \pm 0.01$ | - | $0.22 \pm 0.01$ | $0.26 \pm 0.01$ | $0.16 \pm 0.03$ |
| | Pattern | $0.76 \pm 0.01$ | $0.88 \pm 0.01$ | $\mathbf{0.57 \pm 0.01}^*$ | $\mathbf{0.78 \pm 0.01}^*$ | - | $\mathbf{0.20 \pm 0.01}^*$ | $\mathbf{0.22 \pm 0.01}^*$ | $\mathbf{0.05 \pm 0.04}^*$ |
| EfficientNet-B0 | *Vanilla* | $0.79 \pm 0.01$ | $0.87 \pm 0.01$ | $0.46 \pm 0.01$ | $0.55 \pm 0.01$ | - | $0.55 \pm 0.01$ | $0.46 \pm 0.00$ | $0.39 \pm 0.02$ |
| | lasso | $0.79 \pm 0.01$ | $0.86 \pm 0.01$ | $0.70 \pm 0.01$ | $0.64 \pm 0.01$ | - | $0.52 \pm 0.01$ | $0.01 \pm 0.00$ | $0.11 \pm 0.03$ |
| | logistic | $0.77 \pm 0.01$ | $0.85 \pm 0.01$ | $\mathbf{0.75 \pm 0.01}$ | $0.67 \pm 0.01$ | - | $0.51 \pm 0.01$ | $\mathbf{0.00 \pm 0.00}$ | $0.02 \pm 0.04$ |
| | ridge | $0.78 \pm 0.01$ | $0.82 \pm 0.01$ | $0.74 \pm 0.01$ | $0.67 \pm 0.01$ | - | $0.52 \pm 0.01$ | $0.21 \pm 0.01$ | $0.12 \pm 0.02$ |
| | SVM | $0.77 \pm 0.01$ | $0.85 \pm 0.01$ | $\mathbf{0.75 \pm 0.01}$ | $0.65 \pm 0.01$ | - | $0.52 \pm 0.01$ | $\mathbf{0.00 \pm 0.00}$ | $0.03 \pm 0.04$ |
| | Pattern | $0.77 \pm 0.01$ | $0.85 \pm 0.01$ | $\mathbf{0.75 \pm 0.01}$ | $\mathbf{0.72 \pm 0.01}^*$ | - | $\mathbf{0.48 \pm 0.01}^*$ | $\mathbf{0.00 \pm 0.00}$ | $0.05 \pm 0.04$ |
| EfficientNetV2 | *Vanilla* | $0.77 \pm 0.01$ | $0.86 \pm 0.01$ | $0.46 \pm 0.01$ | $0.51 \pm 0.01$ | - | $0.36 \pm 0.01$ | $0.25 \pm 0.00$ | $0.32 \pm 0.03$ |
| | lasso | $0.78 \pm 0.01$ | $0.85 \pm 0.01$ | $\mathbf{0.75 \pm 0.01}$ | $0.65 \pm 0.01$ | - | $0.35 \pm 0.01$ | $\mathbf{0.00 \pm 0.00}$ | $0.05 \pm 0.02$ |
| | logistic | $0.78 \pm 0.01$ | $0.85 \pm 0.01$ | $0.73 \pm 0.01$ | $0.67 \pm 0.01$ | - | $0.34 \pm 0.01$ | $0.03 \pm 0.00$ | $\mathbf{0.04 \pm 0.02}$ |
| | ridge | $0.78 \pm 0.01$ | $0.85 \pm 0.01$ | $\mathbf{0.75 \pm 0.01}$ | $0.65 \pm 0.01$ | - | $0.35 \pm 0.01$ | $\mathbf{0.00 \pm 0.00}$ | $0.05 \pm 0.02$ |
| | SVM | $0.78 \pm 0.01$ | $0.85 \pm 0.01$ | $\mathbf{0.75 \pm 0.01}$ | $0.66 \pm 0.01$ | - | $0.34 \pm 0.01$ | $\mathbf{0.00 \pm 0.00}$ | $\mathbf{0.04 \pm 0.02}$ |
| | Pattern | $0.79 \pm 0.01$ | $0.85 \pm 0.01$ | $0.71 \pm 0.01$ | $\mathbf{0.70 \pm 0.01}^*$ | - | $\mathbf{0.32 \pm 0.01}^*$ | $0.03 \pm 0.00$ | $0.06 \pm 0.02$ |
| ViT | *Vanilla* | $0.73 \pm 0.01$ | $0.88 \pm 0.01$ | $0.38 \pm 0.01$ | $0.67 \pm 0.01$ | - | $0.15 \pm 0.00$ | $0.47 \pm 0.02$ | $0.25 \pm 0.22$ |
| | lasso | $0.74 \pm 0.01$ | $0.88 \pm 0.01$ | $0.39 \pm 0.01$ | $0.67 \pm 0.01$ | - | $\mathbf{0.16 \pm 0.00}$ | $0.42 \pm 0.03$ | $\mathbf{0.25 \pm 0.22}$ |
| | logistic | $0.73 \pm 0.01$ | $0.88 \pm 0.01$ | $\mathbf{0.62 \pm 0.01}$ | $0.72 \pm 0.01$ | - | $\mathbf{0.16 \pm 0.00}$ | $\mathbf{0.02 \pm 0.06}^*$ | $\mathbf{0.25 \pm 0.22}$ |
| | ridge | $0.74 \pm 0.01$ | $0.88 \pm 0.01$ | $0.48 \pm 0.01$ | $0.66 \pm 0.01$ | - | $\mathbf{0.16 \pm 0.00}$ | $0.12 \pm 0.05$ | $\mathbf{0.25 \pm 0.22}$ |
| | SVM | $0.73 \pm 0.01$ | $0.87 \pm 0.01$ | $0.48 \pm 0.01$ | $0.61 \pm 0.01$ | - | $0.22 \pm 0.00$ | $0.46 \pm 0.02$ | $0.50 \pm 0.00$ |
| | Pattern | $0.74 \pm 0.01$ | $0.88 \pm 0.01$ | $0.61 \pm 0.01$ | $\mathbf{0.73 \pm 0.01}$ | - | $\mathbf{0.16 \pm 0.00}$ | $0.06 \pm 0.06$ | $\mathbf{0.25 \pm 0.22}$ |

Table 9: Results for VGG16, ResNet18, ResNet50, ResNeXt50, ReXNet100, EfficientNet-B0, EfficientNetV2, and Vision Transformer trained on ISIC2019 after model correction with RR-ClArC w.r.t. to the real artifacts "band-aid"|"ruler"|"skin marker". We report accuracy on clean and biased test set, the fraction of relevance put onto the data artifact region for localizable artifacts, and the TCAV score (reported as $\Delta\text{TCAV}^{\text{gt}}$) using the sample-wise ground-truth concept direction $\mathbf{h}^{\text{gt}}$, measuring the models' sensitivity towards the artifacts *after* model correction. Note, that we artificially insert artifacts using estimated localization masks to create a biased test set and to compute $\mathbf{h}^{\text{gt}}$. Stars indicate statistical significance according to z-tests with a significance level of 0.05, and arrows whether low (↓) or high (↑) are better.

| model | CAV | Accuracy (clean) ↑ | | | Accuracy (biased) ↑ | | | Artifact relevance ↓ | | | $\Delta\text{TCAV}^{\text{gt}}$ ↓ | | |
|---|---|---|---|---|---|---|---|---|---|---|---|---|---|
| VGG-16 | *Vanilla* | 0.83 | 0.83 | 0.83 | 0.75 | 0.72 | 0.75 | 0.51 | 0.32 | 0.23 | 0.10 | 0.07 | 0.04 |
| | lasso | 0.81 | 0.82 | 0.82 | 0.77 | 0.77 | **0.75** | 0.48 | 0.25 | 0.22 | 0.12 | 0.07 | **0.05** |
| | logistic | 0.82 | 0.81 | 0.81 | 0.78 | 0.75 | **0.75** | 0.46 | 0.28 | 0.22 | 0.09 | 0.07 | **0.05** |
| | ridge | 0.81 | 0.82 | 0.82 | 0.76 | 0.77 | **0.75** | 0.49 | 0.26 | 0.22 | 0.12 | 0.07 | **0.05** |
| | SVM | 0.82 | 0.82 | 0.81 | 0.78 | 0.74 | **0.75** | 0.46 | 0.28 | 0.22 | 0.11 | 0.07 | **0.05** |
| | Pattern | 0.82 | 0.82 | 0.82 | **0.79** | **0.79*** | 0.75 | **0.31*** | **0.18*** | **0.18*** | **0.03*** | 0.06 | 0.05 |
| ResNet18 | *Vanilla* | 0.85 | 0.85 | 0.85 | 0.79 | 0.83 | 0.80 | 0.22 | 0.13 | 0.15 | 0.32 | 0.18 | 0.02 |
| | lasso | 0.85 | 0.85 | 0.85 | 0.80 | **0.83** | **0.79** | 0.18 | 0.13 | 0.15 | 0.06 | 0.18 | 0.03 |
| | logistic | 0.85 | 0.85 | 0.85 | **0.81** | **0.83** | **0.79** | 0.17 | **0.11** | **0.14** | **0.05** | **0.03** | **0.02** |
| | ridge | 0.85 | 0.85 | 0.85 | 0.80 | **0.83** | **0.79** | 0.18 | **0.11** | 0.15 | 0.06 | **0.03** | 0.03 |
| | SVM | 0.85 | 0.85 | 0.85 | **0.81** | **0.83** | **0.79** | 0.18 | 0.12 | 0.15 | **0.05** | **0.03** | **0.02** |
| | Pattern (ours) | 0.85 | 0.84 | 0.84 | **0.81** | **0.83** | **0.79** | **0.16*** | **0.11** | **0.14** | 0.09 | 0.05 | 0.04 |
| ResNet50 | *Vanilla* | 0.87 | 0.87 | 0.87 | 0.82 | 0.84 | 0.81 | 0.34 | 0.14 | 0.19 | 0.27 | 0.03 | 0.07 |
| | lasso | 0.87 | 0.87 | 0.87 | 0.82 | 0.85 | **0.81** | 0.30 | **0.12** | 0.19 | 0.05 | **0.03** | 0.07 |
| | logistic | 0.87 | 0.87 | 0.87 | **0.83** | 0.85 | **0.81** | 0.24 | 0.13 | 0.18 | 0.05 | **0.03** | 0.07 |
| | ridge | 0.87 | 0.87 | 0.87 | 0.82 | 0.85 | **0.81** | 0.30 | **0.12** | 0.18 | 0.05 | **0.03** | 0.02 |
| | SVM | 0.87 | 0.87 | 0.87 | **0.83** | 0.85 | **0.81** | 0.26 | 0.13 | 0.18 | 0.05 | **0.03** | 0.08 |
| | Pattern (ours) | 0.87 | 0.87 | 0.87 | 0.83 | **0.86** | 0.81 | **0.22*** | 0.12 | **0.17*** | **0.04** | 0.03 | **0.02** |
| ResNeXt50 | *Vanilla* | 0.87 | 0.87 | 0.87 | 0.82 | 0.85 | 0.80 | 0.37 | 0.16 | 0.24 | 0.06 | 0.03 | 0.05 |
| | lasso | 0.87 | 0.87 | 0.87 | 0.82 | 0.85 | 0.80 | 0.34 | 0.15 | 0.23 | 0.06 | 0.04 | 0.05 |
| | logistic | 0.87 | 0.86 | 0.87 | **0.83** | 0.85 | 0.80 | 0.30 | 0.15 | **0.22** | 0.06 | 0.04 | 0.05 |
| | ridge | 0.87 | 0.86 | 0.87 | 0.82 | 0.85 | 0.80 | 0.34 | **0.14** | **0.22** | 0.07 | 0.04 | 0.05 |
| | SVM | 0.87 | 0.86 | 0.87 | **0.83** | 0.85 | **0.81** | 0.31 | 0.15 | 0.23 | 0.06 | 0.04 | 0.05 |
| | Pattern (ours) | 0.87 | 0.86 | 0.87 | **0.83** | **0.86** | **0.81** | **0.27*** | **0.14** | 0.23 | **0.04** | **0.03** | **0.04** |
| ReXNet100 | *Vanilla* | 0.88 | 0.88 | 0.88 | 0.82 | 0.86 | 0.83 | 0.21 | 0.10 | 0.14 | 0.16 | 0.11 | 0.11 |
| | lasso | 0.88 | 0.88 | 0.88 | **0.82** | **0.86** | **0.83** | 0.21 | 0.10 | 0.14 | 0.33 | 0.11 | **0.01** |
| | logistic | 0.88 | 0.88 | 0.88 | **0.82** | **0.86** | **0.83** | 0.21 | 0.10 | 0.14 | 0.34 | 0.11 | **0.01** |
| | ridge | 0.88 | 0.88 | 0.88 | **0.82** | **0.86** | **0.83** | 0.21 | 0.10 | 0.14 | 0.33 | 0.11 | **0.01** |
| | SVM | 0.88 | 0.88 | 0.88 | **0.82** | **0.86** | **0.83** | 0.21 | 0.10 | 0.14 | 0.33 | 0.11 | **0.01** |
| | Pattern (ours) | 0.88 | 0.88 | 0.88 | **0.82** | **0.86** | **0.83** | **0.19*** | **0.08*** | **0.13*** | 0.30 | **0.04*** | **0.01** |
| EfficientNet-B0 | *Vanilla* | 0.88 | 0.88 | 0.88 | 0.83 | 0.86 | 0.83 | 0.22 | 0.08 | 0.11 | 0.12 | 0.04 | 0.02 |
| | lasso | 0.88 | 0.88 | 0.88 | **0.83** | **0.86** | **0.83** | **0.22** | **0.08** | **0.11** | 0.11 | 0.03 | **0.02** |
| | logistic | 0.88 | 0.88 | 0.88 | **0.83** | **0.86** | **0.83** | **0.22** | **0.08** | **0.11** | 0.12 | 0.03 | **0.02** |
| | ridge | 0.88 | 0.88 | 0.88 | **0.83** | **0.86** | **0.83** | **0.22** | **0.08** | **0.11** | 0.12 | 0.04 | **0.02** |
| | SVM | 0.88 | 0.88 | 0.88 | **0.83** | **0.86** | **0.83** | **0.22** | **0.08** | **0.11** | 0.11 | 0.04 | **0.02** |
| | Pattern (ours) | 0.88 | 0.88 | 0.88 | **0.83** | **0.86** | 0.82 | **0.22** | **0.08** | **0.11** | **0.03*** | **0.02** | **0.02** |
| EfficientNetV2 | *Vanilla* | 0.89 | 0.89 | 0.89 | 0.85 | 0.86 | 0.83 | 0.22 | 0.08 | 0.12 | 0.12 | 0.16 | 0.10 |
| | lasso | 0.89 | 0.89 | 0.89 | **0.85** | 0.86 | **0.84** | **0.22** | **0.08** | **0.12** | 0.12 | 0.14 | 0.09 |
| | logistic | 0.89 | 0.89 | 0.89 | **0.85** | 0.86 | **0.84** | **0.22** | **0.08** | **0.12** | 0.12 | 0.16 | 0.10 |
| | ridge | 0.89 | 0.89 | 0.89 | **0.85** | **0.87** | 0.83 | **0.22** | **0.08** | **0.12** | 0.12 | 0.14 | **0.07** |
| | SVM | 0.89 | 0.89 | 0.89 | **0.85** | 0.86 | 0.83 | **0.22** | **0.08** | **0.12** | 0.12 | 0.17 | 0.10 |
| | Pattern (ours) | 0.89 | 0.89 | 0.89 | **0.85** | 0.86 | 0.83 | **0.22** | **0.08** | **0.12** | **0.09*** | **0.09** | 0.09 |
| ViT | *Vanilla* | 0.89 | 0.89 | 0.89 | 0.78 | 0.83 | 0.82 | 0.22 | 0.11 | 0.10 | 0.18 | 0.14 | 0.05 |
| | lasso | 0.89 | 0.89 | 0.89 | **0.78** | 0.84 | **0.82** | **0.22** | 0.10 | **0.10** | 0.18 | 0.12 | 0.06 |
| | logistic | 0.89 | 0.89 | 0.89 | **0.78** | 0.85 | **0.82** | **0.22** | 0.09 | **0.10** | **0.15** | 0.07 | **0.04** |
| | ridge | 0.89 | 0.89 | 0.89 | **0.78** | 0.84 | **0.82** | **0.22** | 0.10 | **0.10** | 0.19 | 0.11 | 0.06 |
| | SVM | 0.89 | 0.89 | 0.89 | **0.78** | 0.83 | **0.82** | 0.25 | 0.11 | **0.10** | 0.25 | 0.15 | **0.04** |
| | Pattern (ours) | 0.89 | 0.89 | 0.89 | **0.78** | 0.85 | 0.82 | **0.22** | **0.08*** | 0.10 | 0.16 | **0.01*** | 0.11 |

## D.8 ADDITIONAL BONE AGE EXPERIMENTS

Complementing the experiments with the artificial brightness, we considered two additional artifacts in the Bone Age dataset. Specifically, we insert an artificial (grayscale) timestamp artifact into 20% of samples of exactly one class during training. Moreover, we consider a real-world artifact occurring in the Bone Age dataset: Images are scaled such that all hands are of similar size, leading to larger "L"-markers for hands of younger children, because the images needed a larger scaling factor due to smaller hands. In both settings, we train VGG16, ResNet18/50, EfficientNet-B0 and EfficientNet-V2 models. In Tab. 10, we report the accuracy on the clean test set and the artifact relevance in both settings, as well as the accuracy on the biased test set and $\Delta$TCAV$^{gt}$ in the controlled experiment. For the timestamp artifact, the Pattern-CAV outperforms filter-based CAVs both in terms of accuracy on the biased data and artifact relevance by a large margin across all architectures, while maintaining a high accuracy on the clean data. For the real-world artifact, the bias mitigation approach using Pattern-CAV successfully reduces the artifact relevance by a large margin for all architectures.

In addition, similar to the qualitative analysis in Section 4.2, we present RelMax visualizations for the most important neurons for different CAVs for a VGG16 model trained on Bone Age. We use CAVs representing the real-world "L"-marker artifact (Fig. 36) and the artificial timestamp artifact (Fig. 37). The same trends as with ISIC2019 (see Fig.27) cab be observed. Specifically, while Filter-CAVs have high values for irrelevant or noisy neurons, Pattern-CAVs have a less uniform distribution over neurons, with top neurons focusing on the concept of interest, *i.e.*, the timestamp and the "L"-marker.

Table 10: Results for VGG16, ResNet18/50, and EfficientNet-B0/V2 trained on Bone Age after model correction with RR-ClArC w.r.t. to the artificial timestamp (*left*) and the "L"-marker artifact (*right*). We report accuracy on clean and biased test set, the fraction of relevance put onto the artifact region, and the TCAV score (reported as $\Delta$TCAV$^{gt}$) using the sample-wise ground-truth concept direction $\mathbf{h}^{gt}$, measuring the models' sensitivity to the artifact. Stars indicate statistical significance according to z-tests (significance level. 0.05), and arrows whether low ($\downarrow$) or high ($\uparrow$) are better.

| model | CAV | Accuracy (clean) $\uparrow$ | Accuracy (biased) $\uparrow$ | Artifact relevance $\downarrow$ | $\Delta$TCAV$^{gt}$ $\downarrow$ |
|---|---|---|---|---|---|
| VGG16 | *Vanilla* | 0.79 \| 0.79 | 0.33 \| - | 0.72 \| 0.17 | 0.26 \| - |
| | lasso | 0.77 \| 0.79 | 0.34 \| - | 0.65 \| 0.17 | 0.33 \| - |
| | logistic | 0.77 \| 0.79 | 0.37 \| - | 0.62 \| 0.18 | 0.29 \| - |
| | ridge | 0.77 \| 0.79 | 0.34 \| - | 0.67 \| 0.17 | 0.33 \| - |
| | SVM | 0.79 \| 0.79 | 0.35 \| - | 0.59 \| 0.17 | 0.20 \| - |
| | Pattern (ours) | 0.78 \| 0.78 | **0.70**$^*$ \| - | **0.30**$^*$ \| **0.14**$^*$ | **0.11**$^*$ \| - |
| ResNet18 | *Vanilla* | 0.77 \| 0.76 | 0.44 \| - | 0.36 \| 0.19 | 0.03 \| - |
| | lasso | 0.78 \| 0.76 | 0.48 \| - | 0.32 \| 0.18 | 0.06 \| - |
| | logistic | 0.78 \| 0.77 | 0.57 \| - | 0.28 \| 0.18 | 0.06 \| - |
| | ridge | 0.77 \| 0.76 | 0.54 \| - | 0.30 \| 0.17 | 0.05 \| - |
| | SVM | 0.77 \| 0.76 | 0.59 \| - | 0.28 \| 0.18 | **0.02** \| - |
| | Pattern (ours) | 0.77 \| 0.75 | **0.62**$^*$ \| - | **0.25**$^*$ \| **0.15**$^*$ | 0.08 \| - |
| ResNet50 | *Vanilla* | 0.79 \| 0.78 | 0.49 \| - | 0.48 \| 0.24 | 0.03 \| - |
| | lasso | 0.80 \| 0.78 | 0.59 \| - | 0.41 \| 0.24 | 0.02 \| - |
| | logistic | 0.80 \| 0.79 | 0.70 \| - | 0.35 \| 0.23 | **0.01** \| - |
| | ridge | 0.80 \| 0.78 | 0.66 \| - | 0.38 \| 0.24 | 0.02 \| - |
| | SVM | 0.80 \| 0.79 | 0.68 \| - | 0.37 \| 0.24 | **0.01** \| - |
| | Pattern (ours) | 0.80 \| 0.78 | **0.72** \| - | **0.33**$^*$ \| **0.17**$^*$ | 0.02 \| - |
| Efficient Net-B0 | *Vanilla* | 0.78 \| 0.79 | 0.39 \| - | 0.61 \| 0.36 | 0.24 \| - |
| | lasso | 0.78 \| 0.79 | 0.40 \| - | 0.61 \| 0.36 | 0.24 \| - |
| | logistic | 0.78 \| 0.78 | 0.47 \| - | 0.58 \| 0.36 | 0.21 \| - |
| | ridge | 0.78 \| 0.79 | 0.40 \| - | 0.61 \| 0.36 | 0.24 \| - |
| | SVM | 0.78 \| 0.79 | 0.46 \| - | 0.59 \| 0.36 | 0.21 \| - |
| | Pattern (ours) | 0.78 \| 0.71 | **0.64**$^*$ \| - | **0.48**$^*$ \| **0.22**$^*$ | **0.05**$^*$ \| - |
| Efficient NetV2 | *Vanilla* | 0.80 \| 0.80 | 0.43 \| - | 0.44 \| 0.18 | 0.08 \| - |
| | lasso | 0.80 \| 0.80 | 0.43 \| - | 0.44 \| **0.18** | 0.08 \| - |
| | logistic | 0.80 \| 0.80 | 0.45 \| - | 0.44 \| **0.18** | **0.06** \| - |
| | ridge | 0.80 \| 0.80 | 0.43 \| - | 0.44 \| **0.18** | 0.08 \| - |
| | SVM | 0.80 \| 0.80 | 0.53 \| - | 0.38 \| **0.18** | 0.07 \| - |
| | Pattern (ours) | 0.81 \| 0.77 | **0.62**$^*$ \| - | **0.35**$^*$ \| 0.18 | 0.27 \| - |

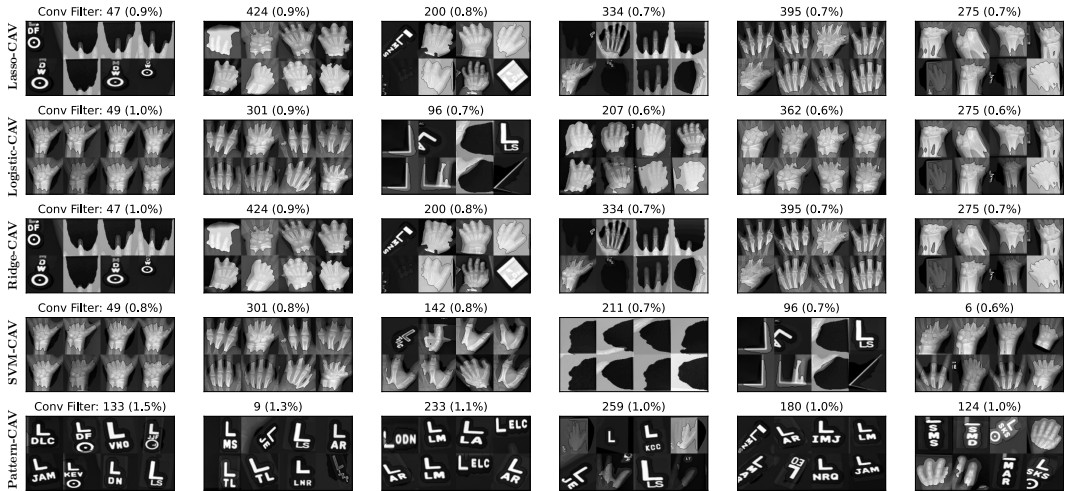

Figure 36: RelMax visualization for neurons corresponding to the largest absolute values in different CAVs, including 4 *filter-* (lasso, logistic, ridge, and SVM) and the *pattern*-CAVs, along with the Conv filter ID and the fraction of all (absolute) CAV values for the real-world "L"-marker artifact in the Bone Age dataset using a VGG16 model. While the filter-CAV picks up noisy neurons, the pattern-CAV uses neurons related to the relevant concept.

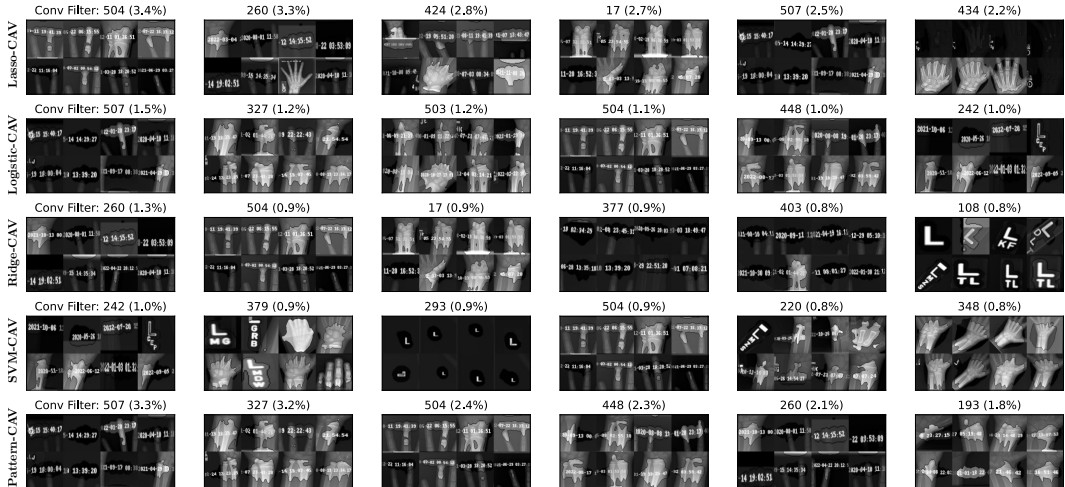

Figure 37: RelMax visualization for neurons corresponding to the largest absolute values in different CAVs, including 4 *filter-* (lasso, logistic, ridge, and SVM) and the *pattern*-CAVs, along with the Conv filter ID and the fraction of all (absolute) CAV values for the artificial timestamp artifact in the Bone Age dataset using a VGG16 model. While the filter-CAV picks up noisy neurons, the pattern-CAV uses neurons related to the relevant concept.

### D.9 ADDITIONAL IMAGENET AND CELEBA EXPERIMENTS

We conducted additional bias mitigation experiments with a natural spurious correlation in CelebA (Liu et al., 2015) and an artificial artifact in ImageNet (Deng et al., 2009). For the former, we study the negative correlation between the presence of ties and blonde hair for a hair color predictor, caused by the existence of many dark-haired men wearing suits (with ties) in the dataset. For the latter, we insert an artificial timestamp into $50\%$ of samples of class "tench" (`n01440764`) and finetune pre-trained models for 10 epochs. To amplify the impact of the artifact, we further insert the artifact into $0.5\%$ of samples from *other* classes as a backdoor by flipping the label to "tench". On both datasets, we train VGG16, ResNet18, ResNet50, EfficientNet-B0, and EfficientNet-V2 models and report bias mitigation results with RR-ClArC in Tab. 11. For all architectures, Pattern-CAVs outperform Filter-CAVs in terms of accuracy on the biased test set for ImageNet. Moreover, Pattern-CAVs achieves superior artifact relevance and $\Delta$TCAV$^{\mathrm{gt}}$ for all architectures except for ResNet18, where results are similar to those for Filter-CAVs.

Table 11: Results for VGG16, ResNet18/50, EfficientNet-B0 and EfficientNetV2 after model correction with RR-ClArC for ImageNet with the artificial timestamp artifact (*left*) and CelebA with the real-world "tie"-artifact (*right*). We report accuracy on clean and biased test set, the fraction of relevance put onto the data artifact region, and the TCAV score (reported as $\Delta$TCAV$^{\mathrm{gt}}$) using the sample-wise ground-truth concept direction $\mathbf{h}^{\mathrm{gt}}$, measuring the models' sensitivity towards the artifacts *after* model correction. Stars indicate statistical significance according to z-tests with a significance level of 0.05, and arrows whether low ($\downarrow$) or high ($\uparrow$) are better.

| model | CAV | Accuracy (clean) $\uparrow$ | Accuracy (biased) $\uparrow$ | Artifact relevance $\downarrow$ | $\Delta$TCAV$^{\mathrm{gt}}$ $\downarrow$ |
|---|---|---|---|---|---|
| VGG16 | *Vanilla* | 0.66 \| 0.92 | 0.53 \| - | 0.15 \| 0.30 | 0.15 \| - |
| | lasso | 0.66 \| 0.92 | 0.52 \| - | 0.15 \| 0.24 | 0.24 \| - |
| | logistic | 0.65 \| 0.92 | 0.58 \| - | 0.13 \| 0.24 | 0.13 \| - |
| | ridge | 0.67 \| 0.92 | 0.53 \| - | 0.13 \| 0.22 | 0.22 \| - |
| | SVM | 0.65 \| 0.91 | 0.58 \| - | 0.14 \| 0.23 | 0.16 \| - |
| | Pattern (ours) | 0.65 \| 0.91 | **0.63**$^*$ \| - | **0.09**$^*$ \| **0.14**$^*$ | **0.11**$^*$ \| - |
| ResNet18 | *Vanilla* | 0.66 \| 0.93 | 0.55 \| - | 0.12 \| 0.25 | 0.32 \| - |
| | lasso | 0.63 \| 0.92 | 0.60 \| - | 0.10 \| 0.26 | 0.02 \| - |
| | logistic | 0.64 \| 0.93 | 0.63 \| - | 0.08 \| 0.26 | 0.03 \| - |
| | ridge | 0.64 \| 0.92 | 0.62 \| - | 0.09 \| 0.26 | 0.02 \| - |
| | SVM | 0.64 \| 0.93 | 0.62 \| - | **0.08**$^*$ \| **0.26** | **0.01**$^*$ \| - |
| | Pattern (ours) | 0.67 \| 0.93 | **0.64**$^*$ \| - | 0.09 \| 0.26 | 0.05 \| - |
| ResNet50 | *Vanilla* | 0.77 \| 0.93 | 0.73 \| - | 0.10 \| 0.29 | 0.06 \| - |
| | lasso | 0.78 \| 0.93 | 0.77 \| - | 0.07 \| 0.29 | 0.05 \| - |
| | logistic | 0.79 \| 0.93 | **0.78** \| - | 0.07 \| 0.28 | 0.05 \| - |
| | ridge | 0.79 \| 0.93 | 0.77 \| - | 0.11 \| 0.29 | 0.09 \| - |
| | SVM | 0.78 \| 0.93 | **0.78** \| - | 0.07 \| 0.28 | 0.06 \| - |
| | Pattern (ours) | 0.78 \| 0.93 | **0.78** \| - | **0.05**$^*$ \| **0.27**$^*$ | **0.01**$^*$ \| - |
| EfficientNet-B0 | *Vanilla* | 0.74 \| 0.92 | 0.52 \| - | 0.18 \| 0.27 | 0.48 \| - |
| | lasso | 0.74 \| 0.92 | 0.51 \| - | 0.18 \| 0.27 | 0.47 \| - |
| | logistic | 0.75 \| 0.92 | 0.55 \| - | 0.17 \| 0.27 | 0.47 \| - |
| | ridge | 0.74 \| 0.92 | 0.51 \| - | 0.18 \| 0.27 | 0.48 \| - |
| | SVM | 0.75 \| 0.92 | 0.53 \| - | 0.18 \| 0.27 | 0.48 \| - |
| | Pattern (ours) | 0.75 \| 0.92 | **0.71**$^*$ \| - | **0.10**$^*$ \| **0.23**$^*$ | **0.42**$^*$ \| - |
| EfficientNetV2 | *Vanilla* | 0.81 \| 0.90 | 0.68 \| - | 0.12 \| 0.25 | 0.45 \| - |
| | lasso | 0.81 \| 0.90 | 0.69 \| - | 0.11 \| 0.25 | 0.44 \| - |
| | logistic | 0.81 \| 0.90 | 0.72 \| - | 0.11 \| 0.25 | 0.43 \| - |
| | ridge | 0.81 \| 0.90 | 0.69 \| - | 0.11 \| 0.25 | 0.44 \| - |
| | SVM | 0.81 \| 0.90 | 0.72 \| - | 0.11 \| 0.25 | 0.44 \| - |
| | Pattern (ours) | 0.81 \| 0.91 | **0.78**$^*$ \| - | **0.10**$^*$ \| **0.20**$^*$ | **0.23**$^*$ \| - |

