# OpenReview forum: "Navigating Neural Space: Revisiting Concept Activation Vectors to Overcome Directional Divergence"
_ICLR.cc/2025/Conference — ICLR 2025 Poster_

### Official Review · Reviewer_LfDf · 2024-10-26

**Soundness:** 3
**Presentation:** 4
**Contribution:** 3
**Rating:** 8
**Confidence:** 4

**Summary:**

The paper introduces pattern-based concept activation vectors (CAVs) that focus solely on concept signals to provide more accurate concept directions, addressing issues with distractor concept directions in filter-based CAVs. The proposed pattern-based CAVs are invariant to feature scaling and more robust to noise.

The method is evaluated across different datasets and model architectures through both controlled and uncontrolled real-world experiments.

**Strengths:**

This work proposes a novel computation approach for CAVs, known as pattern-based CAVs, which are less sensitive to distractors compared to existing filter-based CAV methods.

The paper is well-written and well-structured, presenting multiple experimental results to validate the method’s effectiveness.

The authors objectively analyze the advantages, limitations, and suitable contexts for pattern-based CAVs, highlighting that while pattern-based CAVs offer more accurate concept signals, filter-based CAVs emphasize concept separability, which is important for tasks such as concept-based image classification.

This paper offers valuable insights by highlighting issues with existing filter-based CAVs in the presence of noise and feature rescaling, as well as potential challenges when using filter-based CAVs for TCAV and model explanation.

**Weaknesses:**

However, I have the following concerns:
1. In Table 1, the performance improvements of pattern-based CAVs on the real-world ISIC2019 dataset are not significant, particularly in reducing artifact sensitivity. Given that artifacts (e.g., textual elements in radiology reports or image discrepancies caused by different imaging equipment across medical centers) are hard to avoid in medical contexts, I am concerned that pattern-CAVs may not outperform filter-based CAVs in practical applications.
2. Many medical imaging modalities produce grayscale images, where lesions may be small, and inter-class visual features are often very similar (unlike in ISIC, where subtypes generally have distinct attributes like color or shape). It might be beneficial to include experiments on grayscale datasets, such as Bone Age or similar medical datasets, to better assess the effectiveness of pattern-based CAVs.
3. In the RelMax visualization, samples with the highest neuron activation values are presented. Why do the neurons in the pattern-CAV method have higher activation values than in other CAV methods? Could the authors provide the distribution of neuron activation values across different methods and analyze it? Additionally, could they quantify each CAV method’s ability to identify samples with artifacts (e.g., showing that a certain percentage of neurons with the top activations can identify 95% of samples)?
4. The experiments in the main text should include transformer-based architectures to evaluate the generalizability of the method across different model types.

**Questions:**

My primary concern is the effectiveness of pattern-based CAVs in real-world applications, including tasks with real artifacts and grayscale image classification, please refer to the weaknesses for more details.

here are some minor questions:

1. A brief introduction to RelMax should be provided for better understanding.

---

> ### Author Response · Authors · 2024-11-14
>
> Thank you for your constructive and insightful comments, as well as the positive feedback regarding the writing quality and our experiments. Below, we address your concerns and questions.
>
> - **(W1) Table 1 / significance of results:** While we do see large improvements in artifact relevance for the real artifact (band-aid in ISIC2019) for VGG16 (31% vs 46%) and ResNet (22% vs 26%), it is true that the artifact relevance remains consistent (22%) for EfficientNetB0. However, $\Delta\text{TCAV}^{\text{gt}}$  largely improves (0.03 vs 0.11), indicating that the reliance on the abstract concept “band-aid” is highly reduced. In addition, the heatmaps in Fig. 6 confirm the reduction in artifact relevance through Pattern-CAVs for various real-world artifacts (band-aid, ruler, skin marker) for VGG16. We also want to stress that the improvement of bias mitigation approaches, such as RR-ClArC, is subject to future research to further decrease the artifact relevance. However, this is out of the scope of this paper, as our focus is the improvement of the precision of concept representations. Our findings indicate that Pattern-CAVs clearly address the shortcomings from Filter-CAVs, leading to precise concept directions. We would also like to stress that we do not regard the Pattern-CAV and Filter-CAV necessarily as competing methods, but rather as two complementing approaches providing two fundamentally different views on the classification problem. When looking at the problem through the lense of the “filter”, we aim to optimally separate the data (“discriminative” perspective), e.g., in neuroimaging separate patients from healthy controls. When looking at the problem through the lens of the “pattern”, we aim to identify, e.g., the brain areas where the two populations differ (“explanatory” perspective). The impactful insight of Haufe et al. (2014) was that these two views are not the same. Thus, when precise concept directions matter (e.g., concept erasure/addition, TCAV), we recommend the usage of Pattern-CAVs, due to the addressed shortcomings of filters. However, when a good decision boundary matters to predict the existence of a concept (e.g., as in concept-bottleneck models), Filter-CAVs are better suited. We strongly believe that these two perspectives are not known to the community of CAV users.
> - **(W2) Grayscale images:** We would like to kindly refer to our experiments with grayscale images, specifically the Bone Age dataset, in which we manipulate the brightness to insert a controlled artifact (see Fig. 9 in the appendix). In our experiments, (1) we measure a high alignment between Pattern-CAV and the ground truth concept direction (see Fig. 3, right) and (2) successfully decrease the reliance on the artifact by leveraging the CAV-based bias mitigation approach RR-ClArC (see Tab. 1, first value per cell). This confirms the superiority of Pattern-CAVs over Filter-CAVs also for grayscale images.

---

> > ### Author Response · Authors · 2024-11-14
> >
> > - **RelMax - Explanation and suggested experiment (W3 and Q1):**
> >     - **(Q1) Explanation:** Instead of searching samples with the highest activation for a certain neuron (ActMax), relevance maximization (RelMax) [1] searches for samples, for which the neuron of interest is most important for the prediction. These importance (or relevance) scores can be computed with backpropagation-based local attribution methods that assign relevance scores for a prediction to each activation and input value. Therefore, RelMax selects samples for which the neuron was really important for the prediction, in contrast to ActMax, which only selects based on activation. We will make this clearer in the paper.
> >     - **(W3) Clarification of Figure 2 (Distribution over neurons):** A CAV can be considered as a distribution over neurons and therefore, the percentages above the neurons do not represent activations, but the fraction of that neuron for the entire CAV. Hence, higher scores for top neurons (as for Pattern-CAV) indicate a less uniform distribution over neurons, and a larger focus on the top neurons. We want to refer to Fig. 27 in the appendix for visualizations of the distribution over concepts for additional filter-based CAVs. We observe similar trends: Compared to Pattern-CAV, Filter-CAVs have a more uniform distribution over neurons, and the most important neurons are not necessarily focusing on the concept of interest (timestamp).
> >   - **(W3) Quantification experiment (percentage of neurons to identify 95% of samples):** While the proposed experiment would be feasible, it does not address the core objective of the experiment presented in Fig. 2. The goal of ActMax/RelMax is not the detection of all concept samples, but the identification of most representative samples, in order to understand the role of a particular neuron (i.e. explaining the neuron by providing examples). Furthermore, many samples may not rank among the top-N representatives for any neuron and would therefore be undetectable. The suggested experiment can be considered as a data annotation experiment. Instead of using activations of single neurons to identify biased samples, we highly recommend the projection of activations onto a concept direction, i.e., a CAV. However, as noted in L501, this poses a scenario where the prediction of concept occurrence (i.e., class separability) matters, making filter-CAVs a more suitable option. While we consider data annotation as a very interesting research direction, it is out of the scope of this paper.
> > - **(W4) Transformer-based architectures:** We are happy to move our results for vision transformers from the appendix to the final main paper.
> >
> > Please let us know if anything remains unclear. If you are satisfied with the response, we would kindly ask you to reconsider your rating.
> >
> > Best regards,
> >
> > The authors
> >
> > [1] Achtibat, Reduan, et al. "From attribution maps to human-understandable explanations through concept relevance propagation." Nature Machine Intelligence 5.9 (2023): 1006-1019.

---

> > ### Comment · Reviewer_LfDf · 2024-11-17
> >
> > Thank you for your prompt response, particularly for the explanation of RelMax and the introduction to the two perspectives outlined by Haufe et al. (2014). However, I still have reservations regarding the experimental results on the bone age dataset and their potential applicability to real-world medical scenarios.
> >
> > First, the original images in the bone age dataset already exhibit a wide range of brightness levels. How did the authors ensure that the 20% brightness enhancement applied to a specific subset of images is meaningfully distinct from existing images and that this difference is learned by the model as a shortcut? Regarding the accuracy improvement on the biased dataset reported in Table 1, why do different model architectures exhibit varying trends and magnitudes of improvement? For instance, the accuracy gains of EfficientNet-B0 on the biased dataset and its corresponding TCAV results are both minimal. Additionally, existing methods already perform strongly on the bone age dataset. Could the authors provide a possible explanation for the influence of model architecture on these results and the observed trends in Table 1?
> >
> > Moreover, the paper does not include visualizations similar to Figures 2 or 27, which would provide valuable insights into the results and help address concerns regarding the applicability of pattern-CAV to medical problems involving grayscale images.
> >
> > The bone age dataset also contains inherent artifacts, such as the letters "L" and "JLS." Could the authors consider adopting an approach similar to that used in the ISIC dataset, by introducing similar artifacts into the bone age dataset (e.g., as a gray-scale timestamp or single letter within a specific class) for further testing and evaluation? Based on my experience, artifacts in grayscale images tend to be highly activated in model attribution heatmaps. If pattern-CAV demonstrates strong performance in mitigating this issue, it would represent a notable achievement.
> >
> > At this stage, I will maintain my current rating and look forward to further discussions with the authors regarding these points.

---

> ### Author Response · Authors · 2024-11-20
>
> Thank you again for your very insightful comments and the interesting discussion!
>
> - **How do we ensure the model picks up brightness artifact?**: It is true that the bone age dataset already exhibits a wide range of brightness levels. However, our experiment does not require manipulated images (i.e., images with artificially increased brightness) to be distinct from other, original images with a high brightness level. Assuming an approximately uniform distribution of brightness level over classes *before* our attack, the manipulation of a subset of images of exactly one class shifts that distribution. *After* the attack, there is a clear correlation between brightness level and class label, even if other classes contain a few images with an original brightness level that’s indistinguishable from our attacked images. This correlation can be picked up by the model as a shortcut. In addition, it is to note that *during* training, we do not explicitly force the model to use the shortcut. However, *after* training, we can measure that the model indeed picked up the brightness artifact as a shortcut. Specifically, the accuracy of the Vanilla model on the biased dataset is significantly lower than on the clean dataset. For example, for VGG16, the accuracy drops from 78% on clean data to 50% on attacked data, confirming that the model uses the brightness manipulation as a shortcut.
> - **Other grayscale artifacts in bone age?**: There are indeed other interesting artifacts in the bone age dataset. Based on your valuable suggestion, we ran two additional experiments using the bone age datasets with (1) an artificial grayscale timestamp artifact and (2) the “L”-marker as a naturally occurring artifact. For the latter, we follow [1], who found that models pick up the fact that “L” markings are larger for hands of younger children, as images are scaled such that all hands are of similar size. As a consequence, images of smaller hands (i.e., younger children) need to be enlarged more, leading to larger “L”-markers. Results for VGG16 are provided below and summarized in the following. We further updated the appendix with results for VGG16, ResNet18/50, EfficientNet-B0 and EfficientNet-V2 (see Sec. D.8, Tab. 10).
>     - **Grayscale timestamp**:  Pattern-CAV outperforms filter-based CAVs both in terms of accuracy on the biased test set and artifact relevance by a large margin across all model architectures, while maintaining a high accuracy on the clean test set. For example, for VGG16, the Vanilla model achieves an accuracy of 33% on the biased test set, the best filter-CAV 37% and pattern-CAV 70%. In terms of artifact relevance, the Vanilla model archives 72%, the best filter-CAV 59% and the Pattern-CAV 30% (lower is better).
>     - **Real artifacts (L-marker)**: For the real-world artifact, we report the relevance put onto the “L”-marker and the accuracy on the original (clean) dataset. The Pattern-CAV consistently reduces the artifact relevance the most, as shown below for VGG16. Here, the artifact relevance for the Vanilla model is 72% and can be reduced to 59% with a Filter-CAV and 30% with Pattern-CAV.
>
> These experiments confirm the applicability of Pattern-CAVs for bias mitigation approaches, even for grayscale images, such as the Bone Age dataset.
>
> ### VGG16 Results for (timestamp $\|$ “L”-marker)
> | CAV            | accuracy (clean) | accuracy (biased) | artifact relevance | $\Delta\text{TCAV}^{\text{gt}}$ |
> |----------------|:----------------:|:-----------------:|:------------------:|:-------------------------------:|
> | Vanilla        | 0.79 \| 0.79     |     0.33 \| -     |    0.72 \| 0.17    |            0.26 \| -            |
> | lasso          | 0.77 \| 0.79     |     0.34 \| -     |    0.65 \| 0.17    |            0.33 \| -            |
> | logistic       | 0.77 \| 0.79     |     0.37 \| -     |    0.62 \| 0.18    |            0.29 \| -            |
> | ridge          | 0.77 \| 0.79     | 0.34 \| -         | 0.67 \| 0.17       | 0.33 \| -                       |
> | SVM            | 0.79 \| 0.79     | 0.35 \| -         | 0.59 \| 0.17       | 0.20 \| -                       |
> | Pattern (ours) | 0.78 \| 0.78     | **0.70** \| -         | **0.30** \| **0.14**       | **0.11** \| -                       |
>
>
> [1] Pahde, Frederik, et al. "Reveal to revise: An explainable ai life cycle for iterative bias correction of deep models." International Conference on Medical Image Computing and Computer-Assisted Intervention. Cham: Springer Nature Switzerland, 2023.

---

> > ### Author Response · Authors · 2024-11-20
> >
> > - **Fig. 2/27 for Bone Age**: As per your suggestion, we re-created Fig. 27, visualizing the most important neurons for CAVs, for our new bone age experiments with grayscale images and the (a) timestamp and (b) “L”-marker artifacts. We added the figures to Sec. D.8 in the appendix of the paper (see Figs. 36 and 37). Similar trends as for the RGB-images in ISIC2019 can be observed: In comparison with Filter-CAVs, Pattern-CAVs result in a less uniform distribution with a larger fraction for relevant neurons, that mostly focus on the concept of interest.
> > - **Impact of model architecture**: The effectiveness of filter-based CAVs depends on the imperfections of the model’s latent space, i.e., the noise present in the latent activations. Our introduced Pattern-CAV is more robust towards noise and therefore less dependent on the architecture. As correctly pointed out, according to the experimental results reported in Tab. 1, filter-CAVs already work well for the EfficientNet-B0 experiments in terms of accuracy on the biased test set and $\Delta\text{TCAV}^{\text{gt}}$, possibly due to a noise pattern that filter-CAVs can handle, leaving not much room for improvement for Pattern-CAVs. An exception poses the experiment on ISIC (controlled), where the best filter-CAV achieves an accuracy of 67% on the biased test set and Pattern-CAV achieves 72%. In addition, we see large improvements for our new experiments with grayscale artifacts in bone age (timestamp, L-marker), as described above.
> > Another factor impacting the bias mitigation potential is the clarity of concepts within the model’s latent space. If concepts cannot be represented as directions in a model’s latent space (as assumed by CAVs), or are entangled with other concepts, this can limit the potential to mitigate biases independent of the choice of CAV (filter or pattern).
> >
> > We hope our new findings alleviate your concerns, leading to a reconsideration of your rating. We would also be happy to address any new emerging questions.

---

> > > ### Comment · Reviewer_LfDf · 2024-11-21
> > >
> > > Thanks for your detailed reply, I have revised my rating.

---

### Official Review · Reviewer_KXeP · 2024-11-02

**Soundness:** 2
**Presentation:** 2
**Contribution:** 2
**Rating:** 6
**Confidence:** 3

**Summary:**

The paper proposed pattern-based CAVs based on Concept Activation Vectors (CAVs). The authors assume that the weight vector of the linear classifier may fail at precisely identifying the concept. To amend this, the proposed method slightly modifies the optimization objective from Eq. (1) to Eq. (2) called pattern-based CAV.

**Strengths:**

The paper performed a large number of experiments on both toy datasets and controlled datasets.

**Weaknesses:**

The writing and presentation make the paper hard to follow. It could be better to reorganize the paper to make it easier to understand. For example, it seems better to introduce the concept of TCAV in the Method rather than Experiment section.

The novelty of the paper seems to be somewhat limited. The main contribution Eq. (2) was proposed by a previous work. The authors made a commendable effort to design the dataset, but the technical contribution seems to be the slight modification to the CAV based on linear models.

The datasets used here mainly emphasize hand-crafted concepts ("band-aid", "ruler", "bird", etc.). Is it possible to evaluate the performance on real-world and more realistic datasets like ImageNet?

**Questions:**

On page 5, "we generate pairs of samples with and without the concept". Was this generation the same or different for different datasets? It could be clearer how a pair for a timestamped image or a bird image is generated (e.g., with examples).

In Figure 5, it seems the performance is heavily dependent on the network architecture, and the results of Logistic, SVM, and Patter, are all very similar to GT if the model is ResNet. Does this mean that if we stick to ResNet as the model, then previous methods like SVM and Logistic are already good enough?

Is there any theoretical analysis of how close the vectors can be to the ground truth concepts?

**Details Of Ethics Concerns:**

No ethical issues.

---

> ### Author Response · Authors · 2024-11-14
>
> Thank you for your constructive and insightful comments, as well as the positive feedback regarding our experiments. Below, we address your concerns and questions.
>
> - **(W1) Organization of paper:** We decided to organize the paper as follows: After the introduction and related work, we discuss the filter-pattern problem, the resulting shortcomings of widely-used Filter-CAVs, and introduce our method, the Pattern-CAV as a robust alternative. Afterward, we demonstrate the effectiveness of our approach by (1) measuring the alignment of CAVs with true directions, and (2) evaluating their applicability in two applications, namely TCAV and ClArC. However, both applications are not part of our method, but means for evaluation. Therefore, in order to avoid confusion, we decided not to introduce TCAV in the methods-section.
> - **(W2) Novelty:** We can assure that our work is the first to analyze, evaluate, and mitigate the shortcomings of CAVs stemming from the pattern-filter problem. A previous, unpublished version of this paper is available on arXiv (due to double blindness we can not provide the reference here), but the concept of pattern-CAV has not been introduced in a published paper before.
> Since filter-based CAVs are highly popular in today’s research and are used in hundreds of papers, e.g., they are the main basis of methods such as TCAV [1], ClArC [2], or in mechanistic interpretability [3,4], we believe that making the community aware of their fundamental shortcomings and proposing an improved pattern-based CAV variant is a novel contribution with a large practical value. While previous work such as Haufe et al. (2014) have shown that weights of linear models are susceptible to distractor components in multivariate neuroimaging and Kindermans et al. (2018) use these findings to introduce a new local explainability method (PatternAttribution), we are the first to report these shortcomings in extensive experiments in the context of CAVs. After a decade has passed, one can say without doubts that the results of Haufe et al. have had a large impact on the neuroimaging community. For instance, no one uses the filter weights of linear classifiers for the interpretation of EEG signals anymore, e.g., in the context of Brain-Computer Interfacing. On the contrary, it is unanimously clear that the corresponding pattern is the quantity of interest. We strongly believe that also the steadily growing community of CAV users will benefit from learning about the potential issues of linear classifiers for concept modeling, as well as the availability of a robust alternative.
> - **(W3) Realistic datasets:** We want to stress that we also used real-world datasets (ISIC2019, Bone Age) with real-world, non-hand-crafted artifacts (ruler, band-aid, skin-marker) to solve highly relevant medical tasks. In addition to these real artifacts, we decided to insert controlled artifacts (timestamp, brightness) in separate experiments, to have full control over the artifacts for an improved evaluation. This allows (1) the computation of ground truth CAVs by the generation of pairs of images with and without concept (see next bullet point), (2) the computation of accuracies on clean and attacked test sets, and (3) the computation of artifact relevance via the existence of ground truth masks localizing the artifacts. In addition to evaluating our method on 3 different datasets with controlled and real-world artifacts, we will run experiments with additional datasets (ImageNet, CelebA) in the next few days and add the results to the paper.
> - **(Q1) Pairs of samples with/without concept:** The controlled concepts/artifacts (timestamp, brightness) are inserted by manipulating the original image. Therefore, we can obtain a pair of samples with/without concept by loading the exact same image twice and manipulating only one of them. Examples for ISIC2019 (timestamp) and Bone Age (brightness) are shown in Fig. 9 in the appendix. In the case of FunnyBirds, the removal of a concept is implemented by the replacement of the concept (e.g., green wing of shape “wing-01”) by another, random design of the same body part, as visualized in Fig. 10 in the appendix.

---

> > ### Author Response · Authors · 2024-11-14
> >
> > - **(Q2) Figure 5:** While we observe similar trends for VGG16 and EfficientNet-B0, interestingly, ResNet18 indeed poses an exception. As mentioned in L370, this can be explained by poorly localized concepts, which is visualized in Fig. 26 in the appendix. Instead of precisely localizing concepts, the sensitivity in the last Conv layer spreads over the entire sample (7 × 7 latent feature map). Therefore, TCAV scores for ResNet18 are less impacted by noisy concept sensitivity maps in irrelevant regions. Similar trends have been observed for ResNeXt50 and ReXNet100 models.
> > - **(Q3) Theoretical analysis: how close can CAVs be to ground truth?:** While the CAVs can theoretically perfectly align with ground truth concepts even when certain noise is present, in practice the level of alignment depends on the imperfections of the models and the shape of the noise in the data.  However, we provide theoretical proofs for certain kinds of noise: Namely, for different feature scaling (section B.6 in appendix), and rotated additive noise (B.7), where pattern-CAVs lead to exact ground truth concept directions (and linear classifier-based CAVs do not).
> >
> > Please let us know if anything remains unclear. If you are satisfied with the response, we would kindly ask you to reconsider your rating.
> >
> > Best regards,
> >
> > The authors
> >
> > [1] Kim, Been, et al. "Interpretability beyond feature attribution: Quantitative testing with concept activation vectors (tcav)." International conference on machine learning. PMLR, 2018.
> >
> > [2] Anders, Christopher J., et al. "Finding and removing clever hans: Using explanation methods to debug and improve deep models." Information Fusion 77 (2022): 261-295.
> >
> > [3] Nanda, Neel, at al. "Emergent linear representations in world models of self-supervised sequence models." arXiv preprint arXiv:2309.00941 (2023).
> >
> > [4] Wang, Zihao, et al. "Concept algebra for (score-based) text-controlled generative models."  NeurIPS (2024).

---

> > > ### Author Response · Authors · 2024-11-22
> > >
> > > Dear reviewer KXeP,
> > >
> > > please let us know if you have any further questions or require additional clarifications. If all your concerns have been addressed and alleviated, we would greatly appreciate your consideration in revising your review.
> > >
> > > Best regards,
> > >
> > > The authors

---

> > > > ### Comment · Reviewer_KXeP · 2024-11-27
> > > > **Thank you for the reply!**
> > > >
> > > > Dear Authors, thanks for the detailed reply and clarification. I'll modify my rating accordingly.

---

### Official Review · Reviewer_1Rm2 · 2024-11-04

**Soundness:** 2
**Presentation:** 3
**Contribution:** 2
**Rating:** 6
**Confidence:** 3

**Summary:**

Concept Activation Vectors (CAVs) is a tool in explainable AI for understanding neural network prediction strategies through human-understandable concepts in latent space. This paper argue that current CAV computation methods, which focus on class separability, often diverge from the actual goal of modeling the concept direction accurately. To address this, the authors propose "pattern-based CAVs," which emphasize concept signals while disregarding distractors. They evaluate their method against various model architectures and datasets, demonstrating that pattern-based CAVs align more closely with the true concept direction and yield improved results in applications like concept sensitivity testing and model correction.

**Strengths:**

1. The authors clearly identify the limitations of traditional CAV methods, specifically the influence of distractors in filter-based CAVs.
2. The authors conduct extensive experiments on multiple datasets and architectures, providing both quantitative and qualitative results that support their claims. The improved CAV method is demonstrated to benefit important applications, such as TCAV for concept sensitivity testing and ClArC for model correction.

**Weaknesses:**

1. My main concern is on the limited novelty, it seems that the key idea of pattern-CAV has been proposed, and the paper puts more efforts on evaluation.
2. The pattern-based CAV method, while precise in modeling concept directions, lacks the boundary optimization and transformation capabilities of traditional linear classifiers (e.g., SVM), which may limit its robustness in complex or noisy datasets. Specifically, the absence of separation-focused optimization could lead to inconsistencies in concept direction in cases with overlapping or ambiguous boundaries.
3. The authors focus on achieving high alignment with the true concept direction but acknowledge that in some applications (e.g., post-hoc concept bottleneck models), class-separability may be more critical. A clearer guideline on when to use pattern- vs. filter-based CAVs based on application needs would be helpful.

**Questions:**

1. how to address the unsupervised case? I see the descriptions in line 266 and Appendix, could you please provide more details, since the unsupervised is more important in real applications.
2. In Table 1, why the results of Efficient Net-B0 are quite similar?

---

> ### Author Response · Authors · 2024-11-14
>
> Thank you for your constructive and insightful comments, as well as the positive feedback regarding our experiments. Below, we address your concerns and questions.
>
> - **(W1) Novelty:** We can assure that our work is the first to analyze, evaluate, and mitigate the shortcomings of CAVs stemming from the pattern-filter problem. A previous, unpublished version of this paper is available on arXiv (due to double blindness we can not provide the reference here), but the concept of pattern-CAV has not been introduced in a published paper before.
> Since filter-based CAVs are highly popular in today’s research and are used in hundreds of papers, e.g., they are the main basis of methods such as TCAV [1], ClArC [2], or in mechanistic interpretability [3,4], we believe that making the community aware of their fundamental shortcomings and proposing an improved pattern-based CAV variant is a novel contribution with a large practical value. While previous work such as Haufe et al. (2014) have shown that weights of linear models are susceptible to distractor components in multivariate neuroimaging and Kindermans et al. (2018) use these findings to introduce a new local explainability method (PatternAttribution), we are the first to report these shortcomings in extensive experiments in the context of CAVs. After a decade has passed, one can say without doubts that the results of Haufe et al. have had a large impact on the neuroimaging community. For instance, no one uses the filter weights of linear classifiers for the interpretation of EEG signals anymore, e.g., in the context of Brain-Computer Interfacing. On the contrary, it is unanimously clear that the corresponding pattern is the quantity of interest. We strongly believe that also the steadily growing community of CAV users will benefit from learning about the potential issues of linear classifiers for concept modeling, as well as the availability of a robust alternative.
> - **(W2) Lacks boundary optimization:** We strongly believe that due to the aforementioned filter-pattern problem, optimizing for class-separability (or boundary optimization) is the incorrect optimization target to obtain precise concept directions. This is confirmed by two toy experiments simulating ambiguous boundaries: First, in a 2D experiment in Fig. 7 (bottom row) in the appendix, we sample with different random seeds from two distributions with different means and high standard deviation, leading to overlapping distributions. In this experiment, Pattern-CAVs (green) are more robust than Filter-CAVs (magenta). In a second experiment (Section D.3.2 in appendix), we gradually increase the mislabeling rate of known artifact samples to obscure the decision boundary, and observe consistently higher concept alignment scores for Pattern-CAV compared to Filter-CAVs (see Fig. 29 in the appendix).
> - **(W3) When to use pattern-vs-filter:** When precise concept directions matter (e.g., concept erasure/addition, TCAV), we recommend the usage of Pattern-CAVs, due to the addressed shortcomings of filters. However, when a good decision boundary matters to predict the existence of a concept (e.g., as in concept-bottleneck models), SVM-CAVs are better suited. Please note that the choice whether to consider the pattern or the filter is not an arbitrary choice, but a fundamentally different view on the classification problem. When looking at the problem through the lense of the “filter”, we aim to optimally separate the data (“discriminative” perspective), e.g., in neuroimaging separate patients from healthy controls. When looking at the problem through the lens of the “pattern”, we aim to identify, e.g., the brain areas where the two populations differ (“explanatory” perspective). The impactful insight of Haufe et al. (2014) was that these two views are not the same. We strongly believe that these two perspectives are not known to the community of CAV users. We will make this guidance clearer in the paper.

---

> > ### Author Response · Authors · 2024-11-14
> >
> > - **(Q1) Unsupervised case:** To find concept directions in unsupervised manner, recent approaches perform matrix factorization (e.g, SVD, PCA, or NMF) on latent activations. This leads to two matrices, with one matrix reinterpreted as the concept basis and the other matrix as the activations within that new basis. A nice overview is provided in [5]. It is up to the user to interpret the found set of concepts and there is no guarantee for the desired concept (e.g., ruler, band-aid, skin marker) to be discovered. Investigating the pros and cons of supervised vs. unsupervised concept modelling is an interesting topic. We believe that the former approach is more suitable for model correction tasks (e.g., we know that there is a “band-aid” concept which we want to precisely model and remove), while the latter is focusing more on concept exploration and discovery (e.g., we want to find out in the first place what concepts are represented in the model). However, we agree with the reviewer that the unsupervised case is highly relevant in real-world applications, as the collection of concept labels can be expensive (but can be supported with tools like spectral relevance analysis [6]). We believe that the filter-pattern problem may also affect (depending on the optimization criterion) the unsupervised case, but this has to be investigated in future work.
> > - **(Q2) EfficientNet-B0:** While we do see large and significant improvements for experiments with the timestamp artifact in ISIC2019 (biased accuracy 72% for Pattern-CAV vs 67% for logistic+ridge regression), and the band-aid artifact in ISIC2018 ($\Delta\text{TCAV}^{\text{gt}}$ 0.03 for Pattern-CAV vs 0.11 for SVM-CAV), we indeed only see small differences for Bone Age experiments. Here, despite the distractor component, Filter-CAVs are able to improve the accuracy on the biased test dataset to be almost on par with the accuracy on the clean dataset, as well as to reduce $\Delta\text{TCAV}^{\text{gt}}$ to 0. Therefore, in this case, there is not much room for improvement.
> >
> > Please let us know if anything remains unclear. If you are satisfied with the response, we would kindly ask you to reconsider your rating.
> >
> > Best regards,
> >
> > The authors
> >
> > [1] Kim, Been, et al. "Interpretability beyond feature attribution: Quantitative testing with concept activation vectors (tcav)." International conference on machine learning. PMLR, 2018.
> >
> > [2] Anders, Christopher J., et al. "Finding and removing clever hans: Using explanation methods to debug and improve deep models." Information Fusion 77 (2022): 261-295.
> >
> > [3] Nanda, Neel, Andrew Lee, and Martin Wattenberg. "Emergent linear representations in world models of self-supervised sequence models." arXiv preprint arXiv:2309.00941 (2023).
> >
> > [4] Wang, Zihao, et al. "Concept algebra for (score-based) text-controlled generative models."  NeurIPS (2024).
> >
> > [5] Fel, Thomas, et al. "A holistic approach to unifying automatic concept extraction and concept importance estimation." Advances in Neural Information Processing Systems 36 (2024).
> >
> > [6] Lapuschkin, Sebastian, et al. "Unmasking Clever Hans predictors and assessing what machines really learn." Nature communications 10.1 (2019): 1096.

---

> > > ### Author Response · Authors · 2024-11-22
> > >
> > > Dear reviewer 1Rm2,
> > >
> > > please let us know if you have any further questions or require additional clarifications. If all your concerns have been addressed and alleviated, we would greatly appreciate your consideration in revising your review.
> > >
> > > Best regards,
> > >
> > > The authors

---

> > > > ### Comment · Reviewer_1Rm2 · 2024-11-22
> > > >
> > > > Thanks for the response, my main concerns have been addressed.  I would suggest the authors to evaluate pattern-CAV on recent popular networks for larger impact. I have increased my score.

---

### Official Review · Reviewer_Tafu · 2024-11-04

**Soundness:** 3
**Presentation:** 3
**Contribution:** 3
**Rating:** 8
**Confidence:** 3

**Summary:**

In this paper, the authors propose a pattern-based concept activation vectors (CAV), which focuses on concept signal and provides accurate concept directions. Previous CAV methods are designed to compute by leveraging linear classiifers optimizing the separability of latent repsresentations of samples, which is harmful for accurately modeling the concept direction. The authors evaluate various CAV methods in terms of their alignment with the true concept direction and their impact on CAV applications.

**Strengths:**

- The paper is well written and easy to read. Related work includes several key papers.
- CAVs is widely explored to improve the interpretability of model and this paper tackles an important issue of the current CAV studies and propose a new CAV method named pattern-based CAV.
- The authors measure the alignment of CAVs with the true concept direction by setting controlled environment and the results show that the pattern-CAVs align with the true concept direction.
- The authors show the impact of directional alignment/shift in testing with CAV and model correction.
- Experimental design with real-world datasets look interesting.

**Weaknesses:**

- Some relevant papers are missing [R1]. It would be informative to clarify the contribution of the proposed method and discuss the pros and cons of the proposed method to better find better position of this paper.
- The paper looks many overlaps with [R2]. It would be great to clarify the novel contribution of this study and add discussion with [R2].
- The method is evaluated on BoneAge, ISIC2019 datasets. Evaluation on ImageNet/CelebA will make the paper more strong as in [R2].


[R1] Fel, Thomas, Victor Boutin, Louis Béthune, Rémi Cadène, Mazda Moayeri, Léo Andéol, Mathieu Chalvidal, and Thomas Serre. "A holistic approach to unifying automatic concept extraction and concept importance estimation." Advances in Neural Information Processing Systems 36 (2023).
[R2] Dreyer, Maximilian, Frederik Pahde, Christopher J. Anders, Wojciech Samek, and Sebastian Lapuschkin. "From Hope to Safety: Unlearning Biases of Deep Models by Enforcing the Right Reasons in Latent Space." arXiv preprint arXiv:2308.09437 (2023).

**Questions:**

- Would it be possible to apply the proposed method on a large realworld dataset like ImageNet?
- Would it be possible to add statistical significance of the results in Table 1?

---

> ### Author Response · Authors · 2024-11-14
>
> Thank you for your constructive and insightful comments, as well as the positive feedback regarding the writing quality and our experimental design. Below, we address your concerns and questions.
>
> - **(W1) Related Work (Fel at al., 2023):** We agree that [R1] is a highly relevant and related work and will mention it in our paper. In particular, we think it makes sense to discuss this work in the context of our experiments with unsupervised concept directions described in L266. While [R1] provides a holistic approach for the unsupervised detection of concept directions, our work leverages CAVs computed in supervised manner.
> - **(W2) Overlap with Dreyer et al. (2024):** We can assure this is the first and original work to discuss and demonstrate the limitations of filter-based CAVs in the presence of noise and introduce Pattern-CAVs as robust alternative. Dreyer at al.’s contribution and main focus is the right reason gradient penalization method “RR-ClArC”, not the filter-pattern CAV problem. They do not claim to introduce Pattern-CAV, but rather refer to a previous, unpublished version of this work. Such an overlap in time, where a method is applied before the original paper introducing the method is published, is not uncommon in today’s research.
> - **(W3/Q1) Evaluation on additional datasets:** We evaluate our method on 3 different datasets (ISIC2019, Bone Age, FunnyBirds) with controlled and real-world artifacts. We strongly believe these experiments are sufficiently demonstrating the benefits of our approach. However, we will run experiments with additional datasets (ImageNet, CelebA) in the next few days and add the results to the paper.
> - **(Q2) Statistical significance for results in Tab. 1:** In addition to the standard errors already reported in Tab. 8, we will also report statistical significance using z-tests, both in Tab. 1 (main paper) and Tab. 8 (appendix). Thank you for the suggestion!
>
> We hope our responses are satisfactory. Please let us know if anything remains unclear.
>
> Best regards,
>
> The authors

---

> > ### Comment · Reviewer_Tafu · 2024-11-26
> >
> > Thank you for the response! It will be great to update the manuscript according to the response. I see the other reviews and feedback. I think this paper has a merit for providing new perspective of CAVs for explainable AI. I will keep my original rating.

---

### Meta-Review · Area_Chair_8AUY · 2024-12-19

**Metareview:**

This paper presents a pattern-based concept activation vectors (CAV), which focuses on concept signal and provides accurate concept directions. The paper evaluates their method against various model architectures and datasets, demonstrating that pattern-based CAVs align more closely with the true concept direction and yield improved results in applications like concept sensitivity testing and model correction.

**Additional Comments On Reviewer Discussion:**

All the reviewers agree to accept this paper.

---

### Decision · Program_Chairs · 2025-01-22

Accept (Poster)